# Substantial somatic genomic variation and selection for *BCOR* mutations in human induced pluripotent stem cells

Foad J. Rouhani [1,2,7], Xueqing Zou [3,4,7], Petr Danecek[1,7], Cherif Badja [3,4], Tauanne Dias Amarante[3],
Gene Koh[3,4], Qianxin Wu[1], Yasin Memari[3], Richard Durbin[5], Inigo Martincorena [1],
Andrew R. Bassett [1], Daniel Gaffney [1,6] and Serena Nik-Zainal [3,4 ✉]

We explored human induced pluripotent stem cells (hiPSCs) derived from different tissues to gain insights into genomic integrity at single-nucleotide resolution. We used genome sequencing data from two large hiPSC repositories involving 696 hiPSCs and daughter subclones. We find ultraviolet light (UV)-related damage in ~72% of skin fibroblast-derived hiPSCs (F-hiPSCs), occasionally resulting in substantial mutagenesis (up to 15 mutations per megabase). We demonstrate remarkable genomic heterogeneity between independent F-hiPSC clones derived during the same round of reprogramming due to oligoclonal fibroblast populations. In contrast, blood-derived hiPSCs (B-hiPSCs) had fewer mutations and no UV damage but a high prevalence of acquired *BCOR* mutations (26.9% of lines). We reveal strong selection pressure for *BCOR* mutations in F-hiPSCs and B-hiPSCs and provide evidence that they arise in vitro. Directed differentiation of hiPSCs and RNA sequencing showed that *BCOR* mutations have functional consequences. Our work strongly suggests that detailed nucleotide-resolution characterization is essential before using hiPSCs.

In regenerative medicine, human induced pluripotent stem cells (hiPSCs) and latterly organoids have become attractive model systems because they can be propagated and differentiated into many cell types. Specifically, hiPSCs have been adopted as a cellular model of choice for in vitro disease modeling as well as being considered for cell-based therapies[1–3].

The genomic integrity and tumorigenic potential of human pluripotent stem cells have been explored previously, but systematic large-scale, whole-genome assessments of mutagenesis at single-nucleotide resolution have been limited[4–8]. Human embryonic stem cells (hESCs) cultured in vitro have been reported to harbor *TP53* mutations and recurrent chromosomal-scale genomic abnormalities ascribed to selection pressure[9–14]. However, in contrast, a recent study showed a low mutation burden in clinical-grade hESCs, and no cancer driver mutations were detected[15].

The mutational burden in any given hiPSC comprises mutations that were preexisting in the parental somatic cells from which it was derived and mutations that have accumulated over the course of reprogramming, cell culture and passaging[7,16–22]. Several small-scale genomic studies have shown that in some cell lines, preexisting somatic mutations make up a substantial proportion of the total burden[22–28]. With the advent of clinical trials using hiPSCs (e.g., NCT04339764) comes the need to gain in-depth understanding of the mutational landscape and potential risks of using these cells[29,30]. Here we contrast skin-derived (F-hiPSCs) and blood-derived (B-hiPSCs) from one individual. We then comprehensively assess hiPSCs from one of the world's largest stem cell banks, HipSci, and an alternative cohort called Insignia. All lines

had been karyotypically prescreened and deemed as chromosomally stable. We utilized combinations of whole-genome sequencing (WGS) and whole-exome sequencing (WES) of 555 hiPSC samples and 141 B-hiPSC-derived subclones (Supplementary Table 1) to understand the extent and origin of genomic damage and the possible implications.

## Results

**Genomic variations in skin and blood derived hiPSCs.** To first understand the extent to which the source of somatic cells used to make hiPSCs impacted on mutational load, we compared genomic variation in two independent F-hiPSCs and two independent B-hiPSCs from a 22-year-old healthy adult male (S2) (Fig. 1a). F-hiPSCs were derived from skin fibroblasts, and B-hiPSCs were derived from peripheral blood endothelial progenitor cells (EPCs). Additionally, we derived F-hiPSCs and B-hiPSCs from six healthy males (S7, oaqd, paab, yemz, qorq and quls) and four healthy females (iudw, laey, eipl and fawm) (Fig. 1a).

WGS analysis revealed a greater number of mutations in F-hiPSCs as compared to B-hiPSCs in the individual S2 (~4.4 increase), and in lines derived from the other ten donors (Fig. 1b and Supplementary Table 2). There were very few structural variants (SVs) observed; thus, chromosomal-scale aberrations were not distinguishing between F-hiPSCs and B-hiPSCs (Supplementary Table 2). We noted considerable heterogeneity in the total numbers of mutations between sister hiPSCs from the same donor, S2; one F-hiPSC line (S2_SF3_P2) had 8,171 single substitutions, 1,879 double substitutions and 226 indels, whereas the other F-hiPSC line

[1]Wellcome Sanger Institute, Wellcome Genome Campus, Hinxton, Cambridge, UK. [2]Department of Surgery, University of Cambridge, Cambridge, UK. [3]Early Cancer Institute, Hutchison/MRC Research Centre, Cambridge Biomedical Research Campus, Cambridge, UK. [4]Academic Department of Medical Genetics, Addenbrooke's Treatment Centre, Cambridge Biomedical Research Campus, Cambridge, UK. [5]Department of Genetics, University of Cambridge, Cambridge, UK. [6]Present address: Genomics plc, King Charles House, Oxford, UK. [7]These authors contributed equally: Foad J. Rouhani, Xueqing Zou, Petr Danecek. ✉e-mail: sn206@cam.ac.uk

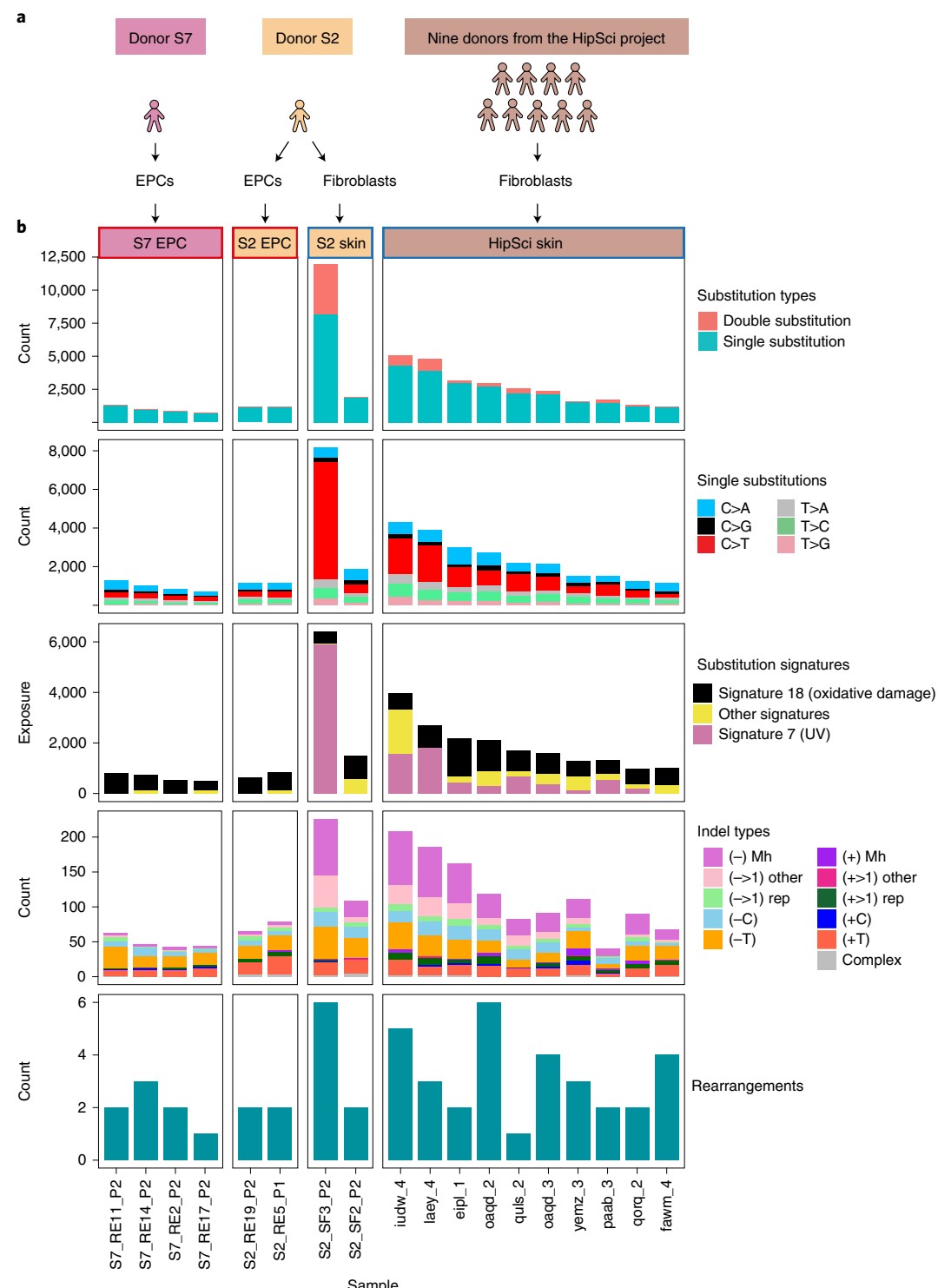

**Fig. 1 | Comparison of mutation burden in EPC-derived and F-hiPSCs. a**, Source of hiPSCs. Multiple hiPSC lines created from patient S2 contrasted to fibroblast- and EPC-derived hiPSCs created from ten other individuals. **b**, Mutation burden of substitutions, double substitutions (first row), substitution types (second row), skin-derived signatures (third row), indel types (fourth row) and rearrangements (lowest row). Supplementary Table 2 provides source information. UV-specific features, such as elevated CC>TT double substitutions and UV mutational signatures, were enriched in F-hiPSCs.

(S2_SF2_P2) had 1,873 single substitutions, 17 double substitutions and 71 indels (Fig. 1b). Mutational signature analysis demonstrated striking predominance of UV-associated substitutions (Reference Signature/COSMIC) Signature 7 (ref. [31]) in the F-hiPSCs, characterized by C>T transitions at TCA, TCC and TCT (Fig. 1b and

Extended Data Fig. 1). This finding is consistent with previously published work that attributed UV signatures in hiPSCs to preexisting damage in parental skin fibroblasts[8,32]. In contrast, EPC-derived B-hiPSCs did not show any evidence of UV damage but showed patterns consistent with possible oxidative damage (signature 18,

characterized by C>A mutations at TCT, GCA and ACA; Fig. 1 and Extended Data Fig. 1). Consistent with in vitro studies[33,34], double substitutions were enriched in UV-damaged F-hiPSCs (Fig. 1b). In all, we concluded that F-hiPSCs carry UV-related genomic damage as a result of sunlight exposure in vivo that does not manifest in EPC-derived B-hiPSCs. Importantly, screening for copy-number aberrations underestimated the substantial substitution/indel-based variation that exists in hiPSCs.

**High prevalence of UV-associated DNA damage in F-hiPSCs.** We asked whether these findings were applicable across F-hiPSC lines generally. Therefore, we interrogated all lines in the HipSci stem cell bank, comprising 452 F-hiPSCs generated from 288 healthy individuals (Fig. 2a and Supplementary Table 3). These F-hiPSC lines were generated using Sendai virus, cultured on irradiated mouse embryonic fibroblast feeder cells, which have been reported to help reduce genetic instability[20] and were expanded before sequencing (range, 7–46 passages; median, 18 passages; Extended Data Fig. 2)[35]. WGS data were available for 324 F-hiPSC and matched fibroblast lines (Fig. 2a). We sought somatic mutations and identified 1,365,372 substitutions, 135,299 CC>TT double substitutions and 54,390 indels. Large variations in mutation distribution were noted, including 692–37,120 substitutions, 0–7,864 CC>TT double substitutions and 17–641 indels per sample, respectively (Fig. 2b and Supplementary Table 4).

Mutational signature analysis[31] revealed that 72% of F-hiPSCs carried detectable substitution signatures of UV damage (Fig. 2c). hiPSCs with greater burden of UV-associated substitution signatures showed strong positive correlations with UV-associated CC>TT double substitutions[36,37] (Fig. 2d and Extended Data Fig. 3) and demonstrated clear transcriptional strand bias with an excess of C>T and CC>TT on the nontranscribed strand, enriched more in early replication timing domains[38–41] than in late ones (Fig. 2e,f). These findings are consistent with previous reports of UV-related mutagenesis observed in fibroblasts[36–41]. Of note, similar to findings of UV damage in skin[42], there was no correlation between total mutation burden of F-hiPSCs and donor age or gender (Fig. 2g,h).

**Substantial genomic heterogeneity between F-hiPSCs clones.** F-hiPSCs comprise the majority of hiPSCs in stem cell banks globally and are a prime candidate for use in disease modeling and cell-based therapies. Yet, we and others observed substantial heterogeneity between hiPSC sister lines generated from one reprogramming experiment (Fig. 1b, subject S2)[5,6,23,32]. It has been postulated that this heterogeneity may result from the presence of genetically diverse clones within the fibroblasts. To explore this further, we compared mutational profiles of 118 pairs of F-hiPSCs present in HipSci, each pair having resulted from the same reprogramming experiment. In all, 54 pairs (46%) of hiPSCs shared more than ten mutations and had similar substitution numbers and profiles (cosine similarity >0.9; Fig. 3a). The remaining 64 hiPSC pairs (54%) shared ten or fewer substitutions and were dissimilar in burden and profile (cosine similarity <0.9; Fig. 3a). We found some striking differences; for example, the F-hiPSC line HPSI0314i-bubh_1, derived from donor HPSI0314i-bubh, had 900 substitutions with no UV signature, whereas HPSI0314i-bubh_3 had 11,000 substitutions with representation mostly from UV-associated damage (>90%). Analysis of the parental fibroblast line HPSI0314i-bubh showed some, albeit reduced, evidence of UV-associated mutagenesis. Hence, we postulate that F-hiPSCs from the same reprogramming experiment could show considerable variation in mutation burden because of different levels of sunlight exposure to each parental skin fibroblast.

To investigate further, we analyzed bulk-sequenced skin fibroblasts, which revealed high burdens of substitutions, CC>TT double substitutions and indels (Extended Data Fig. 4a) consistent with

UV exposure in the majority of fibroblasts (166/204) (Extended Data Fig. 4b). Mutation burdens of F-hiPSCs were positively correlated with their matched fibroblasts (Fig. 3b), directly implicating the fibroblast population as the root cause of F-hiPSC mutation burden and heterogeneity. Investigating variant allele frequency (VAF) distributions of somatic mutations in F-hiPSCs and fibroblasts demonstrated that most F-hiPSC populations were clonal (VAF=0.5), whereas most fibroblast populations showed oligoclonality (VAF<0.5) (Fig. 3c and Extended Data Fig. 5). At least two peaks were observed in VAF distributions of fibroblasts: a peak close to VAF=0 (representing the neutral tail due to accumulation of mutations, which follows the power law distribution) and a peak close to VAF=0.25 (representing the subclones in the fibroblast population). From the mean VAF of the cluster, we can estimate the relative size of the subclones (e.g., VAF=0.25, indicating that the subclone occupies half of the fibroblast population[43]). According to this principle, we found that ~68% of fibroblasts contain oligoclonal populations with a VAF<0.25. The mutational burden of F-hiPSCs is thus dependent on which specific cells they were derived from in that oligoclonal fibroblast population. F-hiPSCs derived from the same subclone will be more similar to each other, whereas hiPSCs derived from different subclones within the fibroblast population could have hugely different mutation burdens (Fig. 3d).

There are important implications that arise from subclonal heterogeneity observed in fibroblasts. First, when detecting somatic mutations, it is preferable to compare the F-hiPSC genome to a matched germline sample if possible; otherwise, F-hiPSC mutations that are also present in a prominent fibroblast subclone will be dismissed as germline variants, giving a false sense of low DNA damage in F-hiPSCs (Supplementary Fig. 1). Indeed, ~95% of HipSci F-hiPSCs had some shared mutations with matched fibroblasts (Extended Data Fig. 6) that demonstrated a strong UV signature (Extended Data Fig. 7). Second, some F-hiPSC mutations may be present in the parental fibroblast population but not detected through lack of sequencing depth. In comparing WES data of the originating fibroblasts, at standard and at high coverage (hcWES), we found that an increased sequencing depth uncovered additional coding mutations that had been acquired in vivo; WES data showed 47% of coding mutations detected in hiPSCs were shared with matched fibroblasts, compared to 64% using hcWES (Extended Data Fig. 8). The additional 17% of mutations identified only in hcWES exhibited a strong UV substitution signature (Extended Data Fig. 8). suggesting that they may have been acquired in vivo and have been present within the parental fibroblast population but undetected at standard sequence coverage. Given recent sequencing studies that have demonstrated a high level of cancer-associated mutations in normal cells[37,44–48], it is therefore probable that some mutations identified in hiPSCs but not detected in corresponding fibroblasts were still acquired in vivo and not during cell culture. Third, this work highlights the need for careful clone selection and comprehensive genomic characterization, as reprogramming can produce hiPSC clones with vastly different genetic landscapes.

**Strong selection for *BCOR* mutations in hiPSCs.** When we mapped hiPSC coding mutations to the COSMIC Cancer Gene Census (a proxy for genes that are selected for), we found a total of 272 mutations in 145 of these cancer genes, across 177 lines (177/452, 39%) from 137 donors. However, it is possible that some of these mutations are passenger mutations, given the heavy mutagenesis from UV damage in some hiPSCs. Hence, to determine potential selective advantage of coding mutations to F-hiPSCs, we conducted an agnostic analysis of selective pressure based on the ratio of divergence of non-synonymous to synonymous substitutions or dN/dS and hotspot analysis across the HipSci F-hiPSCs cohort by examining all genes in the genome[49]. Although several cancer genes were hit across multiple F-hiPSC lines (Supplementary Table 5),

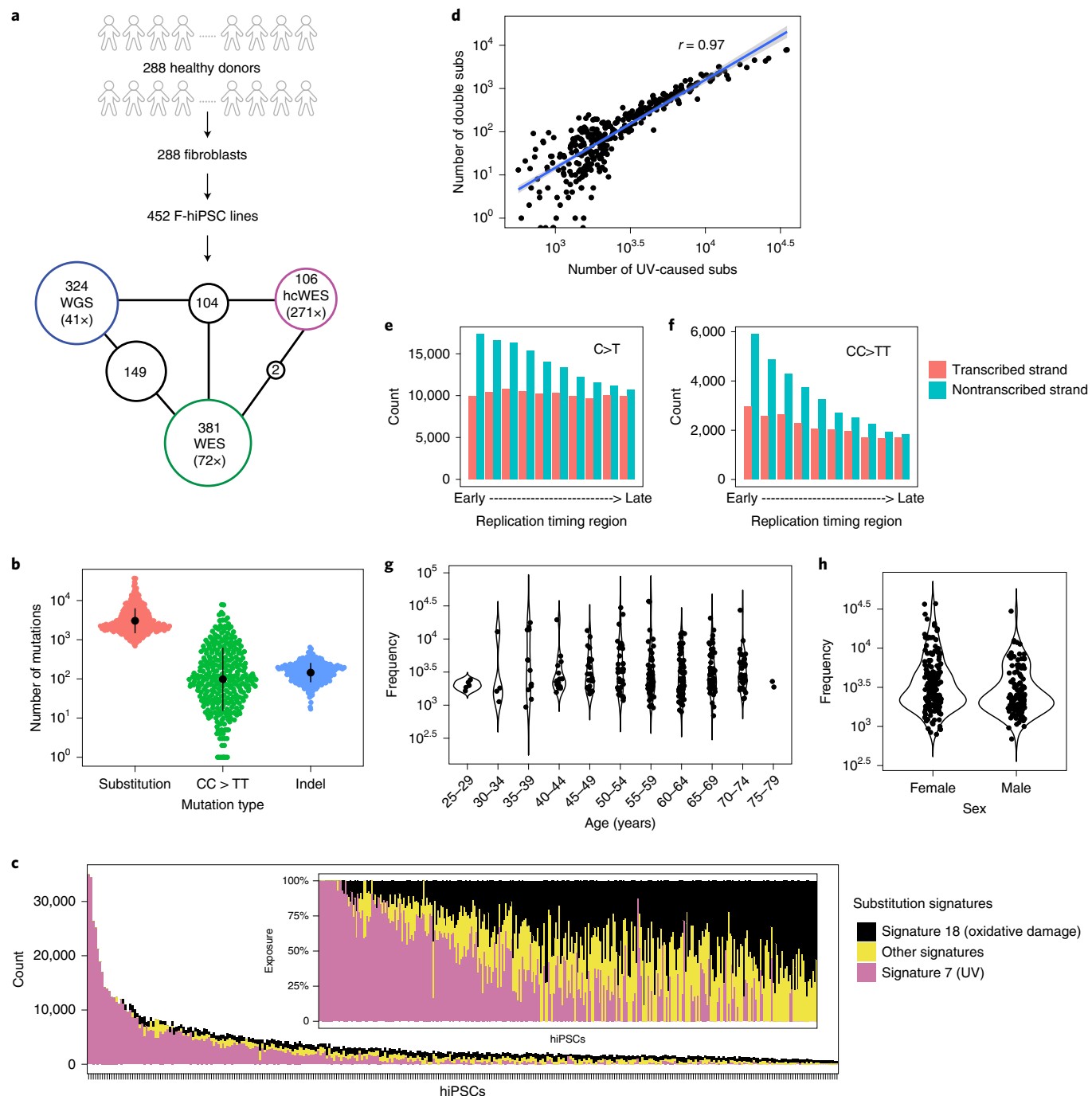

**Fig. 2 | Mutation burden and mutational signatures in F-hiPSCs. a**, Summary of HipSci F-hiPSC dataset. A total of 452 F-hiPSCs were generated from 288 healthy donors. A total of 324 hiPSCs were whole-genome sequenced (coverage, 41×), 381 were whole-exome sequenced (coverage, 72×), and 106 had their matched fibroblasts whole-exome sequenced with high-coverage (hcWES) (coverage, 271×). Supplementary Table 3 provides source information. Numbers within black circles denote the number of F-hiPSCs that had data from multiple sequencing experiments. **b**, Mutation burden of substitutions (subs), CC>TT double substitutions and indels in F-hiPSCs from WGS. Data summary is provided in Supplementary Table 4. Black dots and error represent mean ± standard deviation of hiPSC observations, $n = 324$ WGS of hiPSCs. **c**, Distribution of mutational signatures in 324 F-hiPSC lines. The inset figure shows the relative exposures of mutational signatures types. **d**, Relationships between mutation burdens of CC>TT double substitutions and UV-caused mutation burden of substitutions in F-hiPSCs, $n = 324$ WGS of hiPSCs. **e,f**, Histograms of aggregated mutation burden on transcribed (red) and nontranscribed (cyan) strands for C>T (**e**) and CC>TT (**f**), $n = 324$ WGS of hiPSCs. Transcriptional strand asymmetry across replication timing regions was observed. **g,h**, Distribution of substitution burden of F-hiPSCs with respect to donor's age (**g**) and gender (**h**).

only *BCOR* was found to demonstrate statistically significant positive selection ($q = 3.64 \times 10^{-8}$; Fig. 4a and Supplementary Table 6) using dNdScv[49]. To increase the statistical power for detection of selection, a restricted hypothesis test of known cancer genes[49] still

revealed only *BCOR* mutations as significant. Interestingly, all the *BCOR* mutations found in 11 F-hiPSCs were truncating mutations and predicted to be pathogenic (Fig. 4b). No reads containing these *BCOR* mutations were seen in any founding fibroblast, even when

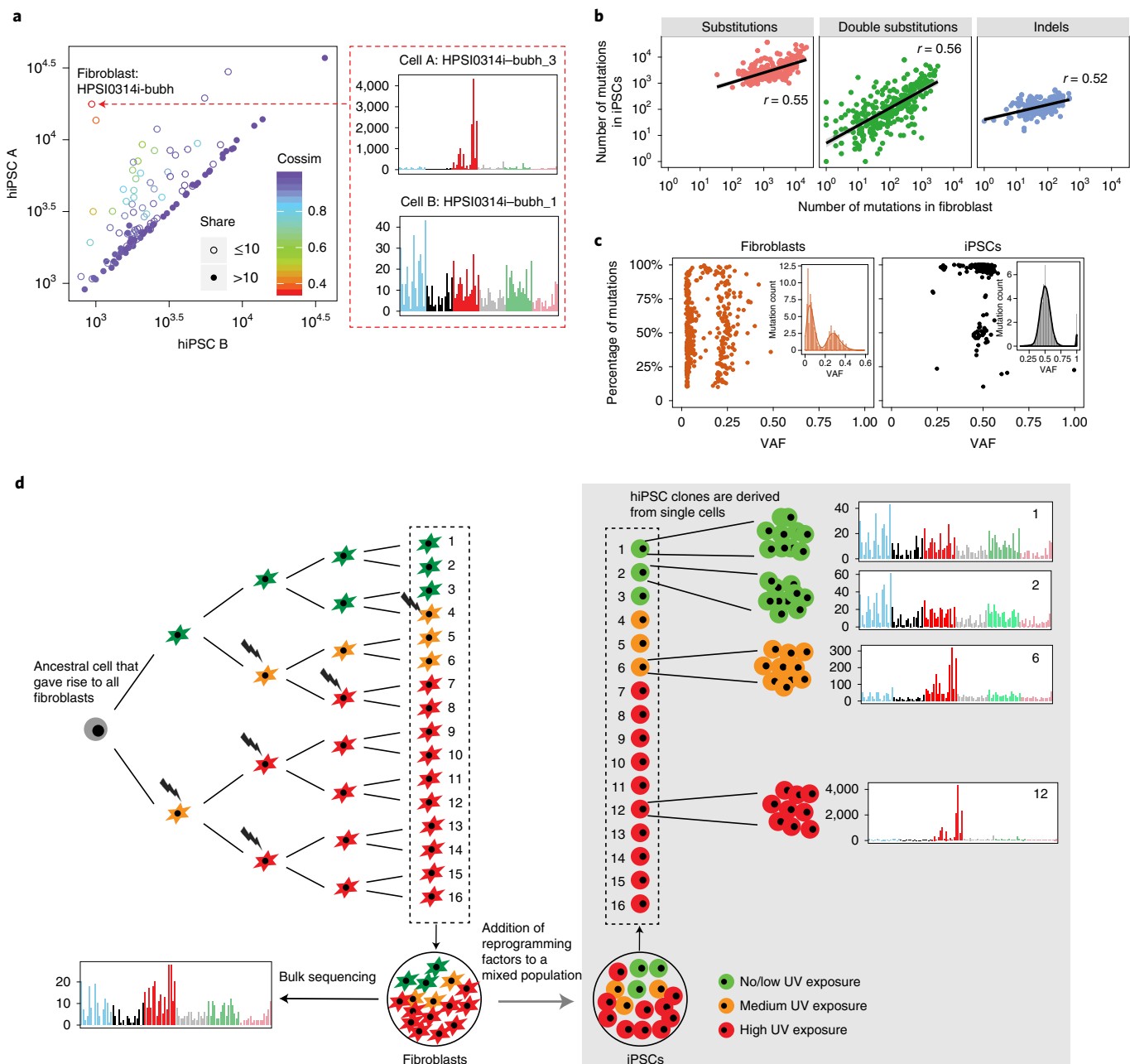

**Fig. 3 | Genomic heterogeneity in F-hiPSCs. a**, Comparison of substitutions carried by pairs of hiPSCs that were derived from fibroblasts of the same donor and in the same reprogramming experiment. Each circle represents a single donor. Hollow circles indicate that two hiPSCs from the same donor share fewer than ten mutations, whereas filled circles indicate that they share ten or more mutations. Colors denote cosine similarity values between mutation profiles of two hiPSCs. A very high score (purple hues) indicates a strong likeness. n=164 fibroblasts with two derived hiPSCs. **b**, Correlations between number of mutations in hiPSCs and their matched fibroblasts, n=324 WGS of hiPSCs. **c**, Summary of subclonal clusters in fibroblasts (n=204 WGS of fibroblasts) and hiPSCs (n=324 WGS of hiPSCs). Kernel density estimation was used to smooth the distribution. Local maximums and minimums were calculated to identify subclonal clusters. Each dot represents a cluster that has at least 10% of total mutations in the sample. Most fibroblasts are polyclonal with a cluster VAF near 0.25, whereas hiPSCs are mostly clonal, with VAF near 0.5. **d**, Schematic illustration of genomic heterogeneity in F-hiPSCs. Through bulk sequencing of fibroblasts, all of these cells will carry all the mutations that were present in the gray cell, their most recent common ancestor. The individual cells will also carry their unique mutations depending on the DNA damage received by each cell. Each hiPSC clone is derived from a single cell. Subclone 1 and subclone 2 cells are more closely related and could share a lot of mutations in common, because they share a more recent common ancestor. However, they are distinct cells and will create separate hiPSC clones. Subclone 6 (orange cells) and subclone 12 (red cells) are not closely related to the green cells and have received more DNA damage from UV, making them genomically divergent from the green cells. They could still share some mutations in common, because they shared a common ancestor at some early point, but will have many of their own unique mutations (largely due to UV damage).

some fibroblasts were sequenced to high coverage (>150×, Fig. 4b), indicating that *BCOR* mutations observed in F-hiPSCs were more likely to have arisen in vitro.

We extended our analysis to B-hiPSCs in the HipSci cohort. These B-hiPSCs were generated from erythroblasts derived from peripheral blood. Three of 17 sequenced B-hiPSCs carried mutations in

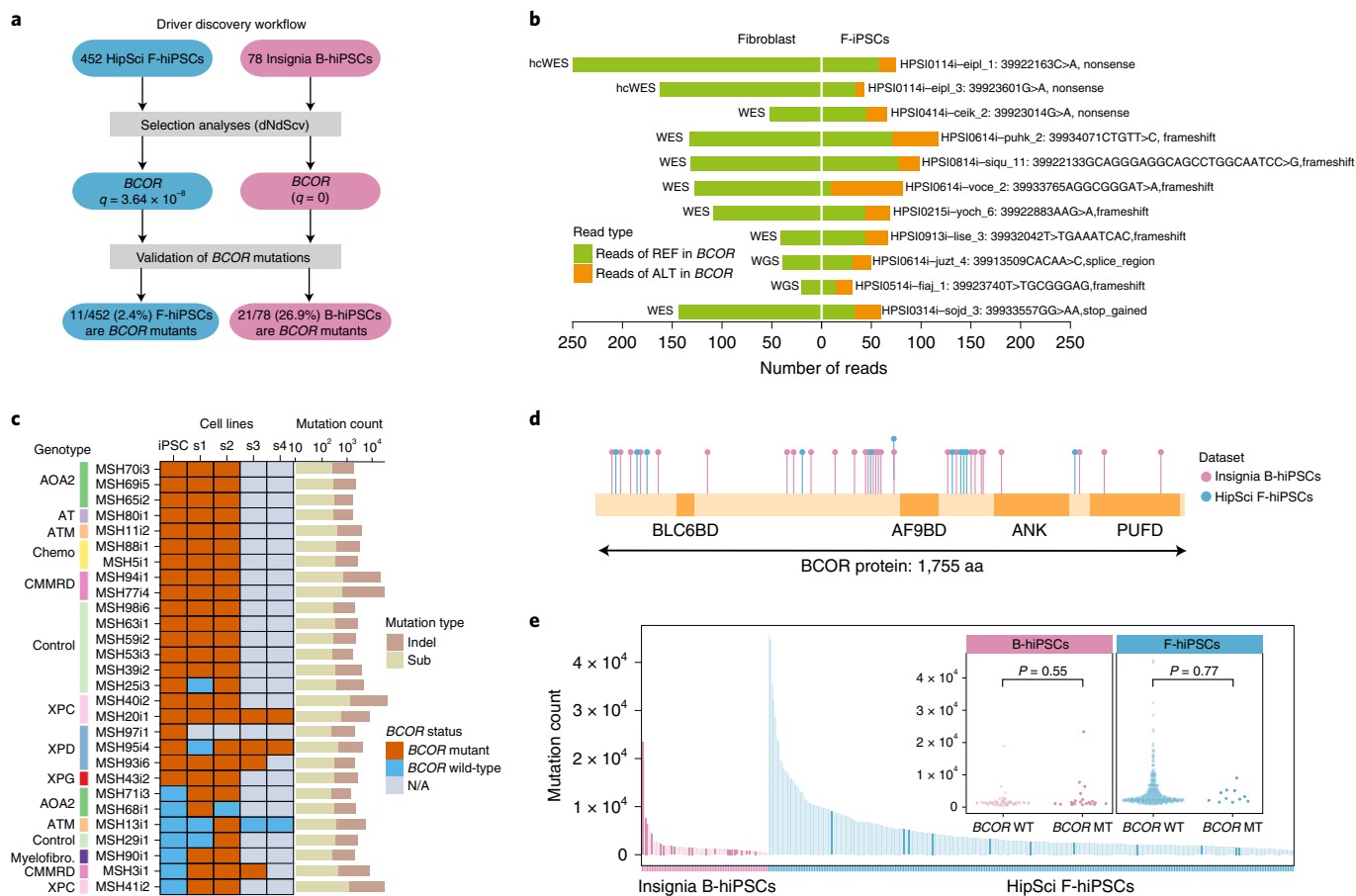

**Fig. 4 | Selection analysis revealed positive selective forces on *BCOR* mutations in F-hiPSCs and B-hiPSCs. a**, Driver discovery workflow. Each *BCOR* mutation was curated using a genome browser. **b**, Number of reads reporting *BCOR* mutations in 11 F-hiPSCs and their founding fibroblasts. Colors are indicative of whether the sequencing read contains a mutation or otherwise (green, reads with reference (REF) allele; orange, reads with alternate (ALT) allele). **c**, *BCOR* mutation status of Insignia B-hiPSCs. Not shown are 50 of 78 B-hiPSCs that do not carry any *BCOR* variants in parental hiPSCs or their corresponding subclones. Rows indicate B-hiPSCs derived from various donors, including patients with genetic defects and healthy controls. Columns indicate parental hiPSC (iPSC) and daughter subclones (s1–s4). *BCOR* mutation status is shown in brown if at least one read contains a *BCOR* mutation in the sample or blue if there are no reads containing *BCOR* mutations in that sample. Gray indicates that a sample was not available (N/A). Total count of substitutions and indels of each iPSC is shown in histogram on the right. **d**, Schematic illustration of positions of mutations in *BCOR* protein in all F-hiPSCs (blue) and B-hiPSCs (purple). **e**, Exploring relationship between hiPSC mutation burden and *BCOR* status (two-sided Mann–Whitney test). hiPSCs with *BCOR* hits are highlighted with a darker shade of purple (B-hiPSCs) and blue (F-hiPSCs). MT, mutant; WT, wild-type.

*BCOR* (Supplementary Table 7), a higher proportion than F-hiPSCs in HipSci (*P* = 0.0076, binomial test). However, many B-hiPSCs in the HipSci cohort did not have matched germline samples to perform subtraction of germline variation, rendering it possible (even if unlikely) for the *BCOR* mutations to be germline in origin.

Therefore, we sought alternative cohorts of hiPSCs. Blood derivation methods for generating hiPSCs have been gaining popularity due to the ease of sample collection but are much less common than F-hiPSCs, and large cohorts of genomically characterized B-hiPSCs do not exist. Nevertheless, we accessed erythroblast-derived B-hiPSCs created from 78 individuals who were part of the Insignia project[50], comprising 53 patients with inherited DNA repair defects, 5 patients with exposure to environmental agents and 20 healthy controls (Supplementary Table 8 and Supplementary Note); WGS was performed, and somatic mutations were identified. dN/dS analysis on all coding variants of the 78 B-hiPSCs showed that only *BCOR* was under significant positive selection[49] (*q* = 0; Fig. 4a and Supplementary Tables 9 and 10), consistent with F-hiPSCs, but present at a much higher prevalence in 21 (26.9%) B-hiPSC lines. Hotspot analysis did not find any recurrently mutated sites in *BCOR* in B-hiPSC lines.

**Source of recurrent *BCOR* mutations.** *BCOR* encodes for the BCL6-corepressor protein and is a member of the ankyrin repeat domain containing gene family. The corepressor expressed by *BCOR* binds to BCL6, a DNA-binding protein that acts as a transcription repressor for genes involved in regulation of B cells. Somatic mutations in *BCOR* have been reported in hematological malignancies, including acute myeloid leukemia and myelodysplastic syndromes[51–53], and has also been reported at a low prevalence in other cancers, including lung, endometrial, breast and colon cancers[54]. The high prevalence of *BCOR* mutations in B-hiPSCs, but not F-hiPSCs, led us to ask whether these could have been derived from hematopoietic stem cell clones that were present in the donors. Clonal hematopoiesis (CH) has been reported in older individuals[55,56]. Some of the most common genes that are mutated in CH include *DNMT3A, TET2, ASXL1, JAK2* and *TP53*[57–59]. *BCOR* is not one of the frequently mutated CH genes but has been reported in rare cases of aplastic anemia[53]. However, *DNMT3A*, reported once in our cohort, could represent a B-hiPSC derived from a CH clone, as the mutation p.G543V has also been reported as a CH variant[60]. Another possibility is that *BCOR* dysregulation is selected for in the culture process particularly in erythroblast-derived hiPSCs.

To examine these possibilities, we asked whether the *BCOR* variants could be detected earlier in the B-hiPSC derivation process, either at the erythroblast population stage or in the germline DNA sample. We did not observe any of these *BCOR* variants in any of the sequencing reads in either erythroblast or germline DNA samples. It is nevertheless possible that CH clones escaped detection through lack of very deep sequencing depth.

Notably, 6 of 21 *BCOR* mutations were present at lower VAFs (VAF < 0.3) in the B-hiPSCs and could represent subclonal *BCOR* variants within the parental B-hiPSC clone. This would suggest that *BCOR* mutations were arising because of ongoing selection pressure in culture following erythroblast derivation or after reprogramming. To investigate further, we cultured parental B-hiPSCs for 12–15 days. A minimum of two (and up to four) subclones were derived from the parental lines (Fig. 4c and Supplementary Table 11). A total of 141 subclones were assessed by WGS (Supplementary Table 12). In all parental B-hiPSC clones where *BCOR* mutations were identified and where daughter subclones were successfully generated, the *BCOR* variants were recapitulated in related daughter subclones, serving as an internal validation of those *BCOR* variants (Fig. 4c and Supplementary Table 10). Interestingly, 7 additional B-hiPSCs that did not have *BCOR* mutations in the parental B-hiPSC population developed new *BCOR* mutations in daughter subclones; MSH71 had p.P1229fs*5 *BCOR* mutations in both subclones but not the parent, MSH68 had p.D1118fs*22 *BCOR* mutations in one of two subclones, MSH13 had p.P1115fs*45 *BCOR* mutation in one of four subclones, MSH29 had a p.S1122fs*37 *BCOR* mutation in one of two subclones, MSH90 had a p.D1118fs*44 in *BCOR* in two subclones, MSH3 had a p.S158fs*28 *BCOR* mutation in all three subclones and MSH41 had p.R1398fs*4 in *BCOR* in two subclones (Fig. 4c and Supplementary Table 10). Given that these *BCOR* mutations are present in some, but not all, subclones and are not present at a detectable frequency in the parental population, this finding suggests that they have arisen late in culture of the parental B-hiPSC. All *BCOR* mutations in B-hiPSCs were predicted to be truncating variants distributed throughout the gene (Fig. 4d). No correlation was found between mutation burden in parental line and *BCOR* status in either F-hiPSCs ($P = 0.77$, Mann–Whitney test) or B-hiPSCs ($P = 0.55$, Mann–Whitney test), indicating that *BCOR* mutations were not simply mutations in hypermutated samples (Fig. 4e).

Therefore, in this analysis, we found a high prevalence of recurrent *BCOR* mutations in two independent cohorts of erythroblast-derived B-hiPSCs (18% of HipSci and 27% of Insignia (~25% overall, across all B-hiPSCs)). We did not find *BCOR* mutations in originating erythroblasts or in germline DNA samples. Instead, single-cell-derived subclones variably carried new *BCOR* mutations, suggesting that there may be selection for *BCOR* dysfunction post-reprogramming in B-hiPSCs. We cannot definitively exclude CH as the source of *BCOR* mutations, as extremely high sequencing depths may be required to detect CH clones. However, *BCOR* is not frequently mutated in CH. The majority of donors were young (<45 yr), and some were healthy controls. All arguments against *BCOR* being due to CH, and more likely due to selection pressure in B-hiPSCs in culture. That *BCOR* mutations are seen at a lower frequency in F-hiPSCs and never in fibroblasts hints at a culture-related selection pressure (Fig. 4b). Why *BCOR* mutations are more enriched in B-hiPSCs remains unclear but may be related to the process of transforming peripheral blood mononuclear cells (PMBCs) toward the myeloid lineage.

**Global transcriptional changes in *BCOR*-mutated B-hiPSCs.** To understand the functional impact of recurrent *BCOR* variants in B-hiPSCs, we performed RNA sequencing on B-hiPSCs from all 78 Insignia donors. Global transcriptomic analysis revealed two principal components (PCs) driving variance in the dataset (Fig. 5a). The first PC distinguished two groups, with almost all the *BCOR*-mutated

B-hiPSCs (VAF > 0, highlighted in yellow or orange in Fig. 5a) restricted to one group, herewith termed *BCOR*-mut. The other group comprised B-hiPSCs with no *BCOR* mutations, *BCOR*-wt (VAF = 0, highlighted in gray or blue in Fig. 5a). The second PC distinguished the donors by gender; *BCOR* is on the X chromosome. This result suggests that *BCOR* mutations are associated with important global transcriptional changes in B-hiPSCs. Differential gene expression analysis showed that 10,486 genes were differently regulated in the *BCOR*-mut lines (Fig. 5b and Supplementary Table 13). Furthermore, we found that *UTF1* (implicated in maintenance of pluripotency through chromatin regulation), *VENTX*, *IRX4*, *PITX2* and *MIXL1* (all homeobox-related proteins with various roles in embryonic patterning) and *FOXC1* (important in the development of organs derived from the mesodermal-lineage) were strongly upregulated in BCOR-mut lines, whereas *RAX* (involved in development of the hypothalamus and retina) was strongly downregulated (Fig. 5b).

**BCOR-mutant hiPSCs have impaired differentiation capacity.** *BCOR* is a member of the polycomb group of proteins (PRC1.1) that regulates self-renewal and differentiation of stem cells and is critical for maintaining pluripotency[61]. *BCOR* depletion has been associated with decreased polycomb repressive activity, initiating differentiation toward the endodermal and mesodermal lineages[61]. Thus, we next asked whether the differentiation capacity was compromised in *BCOR*-mut lines, particularly for ectodermal lineages. We performed directed differentiation toward a neuronal lineage, contrasting two independent *BCOR*-wt B-hiPSC lines, MSH34i2 and MSH30i3, and two independent *BCOR*-mut B-hiPSC lines, MSH40i2 and MSH93i6 (Fig. 5c,d), each with three biological replicates. To capture dynamic shifts through directed differentiation, we took samples during a time course, characterizing these lines morphologically and transcriptionally on day 0 as hiPSCs, day 6 and day 12 representative of early and late neural stem cell (NSC) induction stages respectively and on day 27 as neurons.

There were no overt morphological disparities between *BCOR*-mut and *BCOR*-wt colonies at the hiPSC stage, with near-identical brightfield images and immunofluorescence characterization of pluripotent markers SSEA4 and OCT4 (day 0; Fig. 5d, Extended Data Figs. 9 and 10a). However, upon NSC induction (days 6 and 12), *BCOR*-mut replicates showed inefficiency in differentiation confirmed by patchy expression of NSC marker PAX6 (Fig. 5d and Extended Data Fig. 10b). At day 27, neuronal generation in *BCOR*-mut lines was markedly affected, as seen in depletion of neuronal marker TUBB3 (Fig. 5d and Extended Data Fig. 10c).

In keeping with the morphological characterization, global transcriptomics of each hiPSC line at each stage of differentiation revealed differences between *BCOR*-mut and *BCOR*-wt B-hiPSCs (Supplementary Fig. 2). At the pluripotent stage, *BCOR*-mut hiPSCs exhibited a modest reduction in *BCOR* expression but elevated levels of *NANOG*, *KLF4* and *NODAL* compared to *BCOR*-wt (Fig. 5c), similar to reports in other pluripotent models with *BCOR*-PRC1.1 defects[61].

At the NSC induction stage (days 6 and 12), transcriptional dynamics evolved to show that mesodermal markers such as *PAX7*, *TBX1* and *PAX3* were substantially elevated in the *BCOR*-mut compared to *BCOR*-wt B-hiPSCs, implicating a drive toward mesodermal lineages in *BCOR* compromised lines (Fig. 5c). This difference was maintained at late neuronal differentiation (day 27), where neuronal markers such as *TUBB3*, *DCX* and *FOXG1* were upregulated in *BCOR*-wt but not *BCOR*-mut B-hiPSCs, underscoring the failure of neuronal differentiation in the latter (Fig. 5c).

In all, these results demonstrate that B-hiPSC lines that acquire *BCOR* mutations may have compromised differentiation potential. *BCOR*-mutated lines seem to be less efficient at differentiation toward neuronal lineages and transcriptionally appear primed to differentiate toward mesodermal lineages.

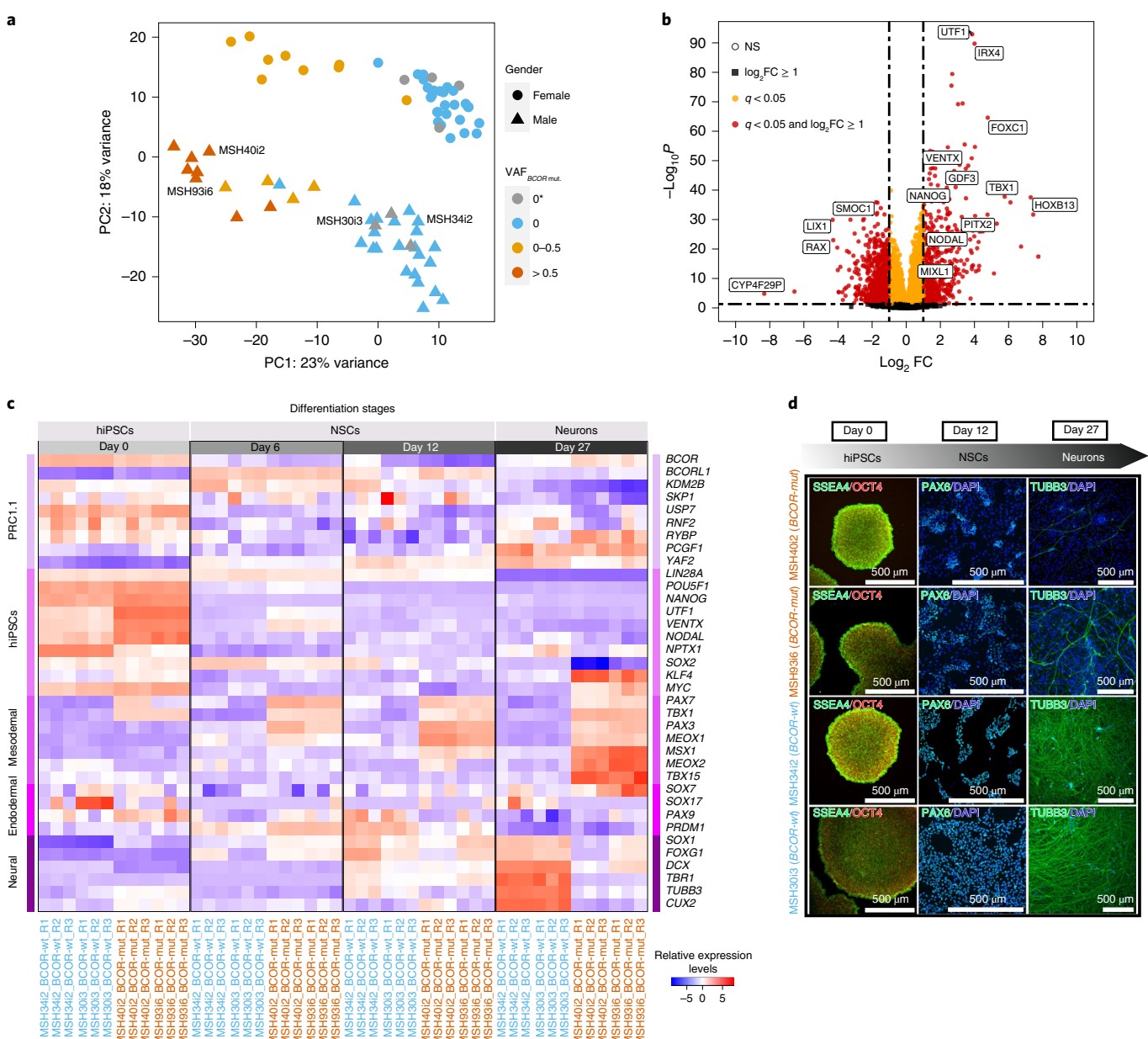

**Fig. 5 | Functional validation of the impact of *BCOR* mutations in hiPSCs and their differentiation potential. a**, PC analysis showing hiPSCs distribution based on their transcriptome expression. Transcriptomics changes are correlated with *BCOR* mutation VAF values. 0* denotes that no *BCOR* mutations were found in hiPSC parental lines but were seen in their subclones. **b**, Volcano plot of differential gene expression analysis in hiPSCs with *BCOR* mutations (*BCOR*-mut) and no *BCOR* mutations (*BCOR*-wt) lines. FC, fold change. **c**, Heatmap of relative expression levels of members of noncanonical polycomb repressive complex 1.1 (PRC1.1), hiPSCs pluripotency and the three germ layers markers for two *BCOR*-wt (blue: MSH34i2, MSH30i3) and two *BCOR*-mut (orange: MSH40i2, MSH93i6) hiPSC lines. **d**, Immunofluorescence characterization of hiPSCs, neural stem cells (NSCs) and neurons at day 0, day 12 and day 27, respectively. Both *BCOR*-mut and *BCOR*-wt hiPSCs have similar undifferentiated morphology and express pluripotency markers OCT4/POU5F1 (red) and SSEA4 (green). *BCOR*-mut lines undergo inefficient neural induction, as highlighted by a reduced number of NSCs expressing PAX6 (green) at day 12 and a reduced number of neurons marked by TUBB3 (green) at day 27.

**Long-term culture increases 8-oxo-dG DNA damage.** Finally, we examined the genomic effects of in vitro hiPSC culture. Mutational signature analyses revealed that all hiPSCs, regardless of primary cell of origin, have imprints of substitution signature 18, previously hypothesized to be due to oxidative damage in culture[22] (Figs. 1b and 2c). Recently, knockouts of *OGG1*, a gene encoding a glycosylase in base excision repair that specifically removes 8-oxoguanine (8-oxo-dG), have been shown to result in mutational signatures characterized by C>A at A̲C̲A̲>A̲A̲A̲, G̲C̲A̲>G̲A̲A̲, G̲C̲T̲>G̲A̲T̲, identical to signature 18 (ref. [62]). Of note, signature 18 has also been

reported in other cell culture systems such as in ES cells[15], near haploid cell lines[63] and human tissue organoids in which the contribution to the overall mutation burden was reported to increase with *in vitro* culture[64,65].

We examined mutations shared between hiPSCs and their matched fibroblasts, representing in vivo and/or early in vitro mutations, and private mutations that are only present in hiPSCs, most (but not all) of which are likely representative of mutations acquired in vitro. We observed that signature 18 is enriched among private mutations of nearly all hiPSCs (313 or 97%). By contrast, the

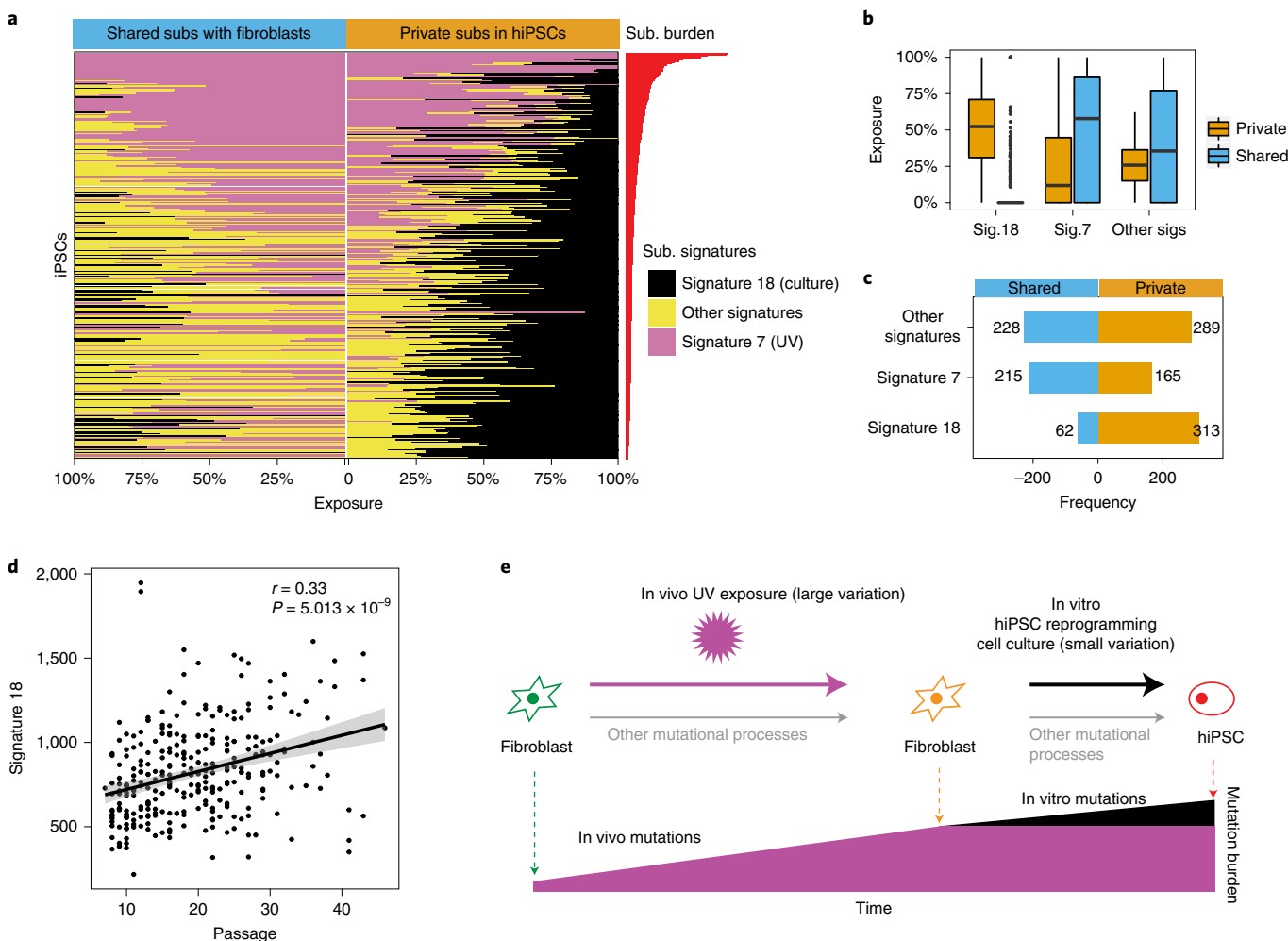

**Fig. 6 | Culture-associated mutagenesis in hiPSCs. a**, Mutations in F-hiPSCs were separated into two groups: (1) mutations shared between F-hiPSCs and their founding fibroblasts (left) and (2) mutations that are private to hiPSCs (right). Proportional graphs show substitution signatures of shared and private mutations. Culture-related signature (signature 18) is enriched in private mutations. **b**, Box plot of exposures of substitution signatures of shared (blue) and private (orange) mutations. Culture signatures (signature 18) account for ~50% of private substitutions, in contrast to nearly zero among shared mutations. Box plots denote median (horizontal line) and 25th to 75th percentiles (boxes). The lower (minima) and upper (maxima) whiskers extend to 1.5× the interquartile range. $n = 324$ WGS of hiPSCs. **c**, Number of samples carrying each signature within shared or private mutations. Most samples acquired the culture signature late in the hiPSC cloning process. **d**, Relationship between exposure of culture-related signature (signature 18) and passage number (Pearson's correlation test, two sided). **e**, Schematic illustration of the mutational processes in fibroblasts and F-hiPSCs.

majority of hiPSCs (262 or 80%) showed no evidence of signature 18 among shared variants (Fig. 6a–c and Supplementary Fig. 3). We then investigated the relationship between signature 18 and passage number in the HipSci cohort. We found that there was a positive correlation between signature 18 and passage number (correlation = 0.327; $P = 5.013 \times 10^{-9}$; Fig. 6d), reinforcing the notion that prolonged time in culture is likely to be associated with increased acquisition of somatic mutations through elevated levels of DNA damage from 8-oxo-dG (Fig. 6e).

## Discussion

hiPSCs are on the verge of entering clinical practice. Therefore, there is a need to better understand the breadth and source of mutations in hiPSCs to minimize the risk of harm. Crucially, our work shows that the choice of a starting material for hiPSC derivation is complex. We demonstrate copious UV-associated genomic damage and substantial variation in genomic integrity between different clones from the same reprogramming experiment. In all, 39% of F-hiPSCs carried at least one mutation in a cancer driver gene. However,

only *BCOR* mutations showed evidence of statistically significant positive selection, both in F-hiPSCs and in B-hiPSCs (prevalence of 25%), despite lower rates of genome-wide mutagenesis in the latter. The lack of *BCOR* mutations in the founder somatic cell populations (fibroblasts and erythroblasts) and the finding of new *BCOR* mutations arising in subclones during propagation experiments provides support for strong selection for *BCOR* variants in hiPSC culture. Furthermore, our work demonstrates that these mutations are associated with transcriptomic changes and influence differentiation of hiPSCs. Finally, both F-hiPSCs and B-hiPSCs exhibit oxidative damage that is increased by long-term culture. Critically, all of these lines had been previously screened for large-scale aberrations, and therefore, our work shows the value of WGS to fully capture variation at the resolution of nucleotides in hiPSCs. Further work will be necessary to investigate the functional significance of these mutations; for example, do they predispose cells toward malignant transformation or alter the fate of differentiated progeny? Our work highlights best practice points to consider when establishing hiPSC cellular models: an originating somatic cell type with low levels of

preexisting genomic damage and minimizing duration of cell culture. Ultimately, however, comprehensive genomic characterization is indispensable to fully understand the magnitude and significance of mutagenesis in all cellular models.

## Online content

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

## Methods

**Samples.** The research project complied with all relevant ethical regulations, and the protocols were approved by research ethics committees (details below). All participants were recruited voluntarily; provided written, informed consent; and were not financially compensated. The procedures used to derive fibroblasts and EPCs from two donors (S2 and S7) were previously described[22]. Two F-hiPSCs were obtained from S2. Four and two EPC B-hiPSCs were obtained from S7 and S2, respectively. A total of 452 F-hiPSCs were obtained from 288 donors and 17 erythroblast B-hiPSCs were obtained from 9 donors from the HipSci project. A total of 78 erythroblast B-hiPSCs and 141 subclones were obtained from 78 donors from the Insignia project. The details of HipSci and Insignia iPSC lines are described below.

**HipSci hiPSC line generation and growth.** All lines were incubated at 37 °C and 5% $CO_2$ (ref. [35]). Primary fibroblasts were derived from 2-mm punch biopsies collected from organ donors or healthy research volunteers recruited from the NIHR Cambridge BioResource under ethics for hiPSC derivation[35] (Cambridgeshire and East of England Research Ethics Committee REC 09/H306/73, REC 09/H0304/77-V2 04/01/2013, REC 09/H0304/77-V3 15/03/2013). Biopsy fragments were mechanically dissociated and cultured with fibroblast growth medium (knockout DMEM with 20% fetal bovine serum; 10829018, ThermoFisher Scientific) until outgrowths appeared (within 14 days, on average). Approximately 30 days following dissection, when fibroblasts cultures had reached confluence, the cells were washed with phosphate-buffered saline (PBS), passaged using trypsin into a 25-cm² tissue-culture flask and then again to a 75-cm² flask upon reaching confluence. These cultures were then split into vials for cryopreservation and those seeded for reprogramming, with one frozen vial later used for DNA extraction for WES or WGS.

For erythroblast derivation, two blood samples were obtained for each donor. All samples were scanned and checked for ethical approval. Peripheral blood mononuclear cell (PBMC) isolation, erythroblast expansion and hiPSC derivation were done by the Cellular Generation and Phenotyping facility at the Wellcome Sanger Institute, Hinxton. Briefly, whole-blood samples collected from consented patients were diluted with PBS, and PBMCs were separated using standard Ficoll Paque density gradient centrifugation method. Following the PBMC separation, cells were cultured in expansion media containing StemSpan H3000, stem cell factor, interleukin-3, erythropoietin IGF-1 and dexamethasone for a total of 9 days[66].

The EPCs were isolated using Ficoll separation of 100 ml peripheral blood from organ donors (REC 09/H306/73) and the buffy coat transferred onto a 5 μg/cm² collagen (BD Biosciences, 402326)-coated T-75 flask. The EPCs were grown using EPC media (EGM-2MV supplemented with growth factors, ascorbic acid plus 20% Hyclone serum; CC-3202, Lonza and HYC-001-331G; ThermoFisher Scientific Hyclone respectively)[67]. EPC colonies appeared after 10 days, and these were passaged using trypsin in a 1 in 3 ratio and eventually frozen down using 90% EPC media and 10% dimethyl sulfoxide.

Fibroblasts and erythroblasts were transduced using nonintegrating Sendai viral vectors expressing human OCT3/4, SOX2, KLF4 and MYC51 (CytoTune, Life Technologies, A1377801) according to the manufacturer's instructions and cultured on irradiated mouse embryonic fibroblasts (MEFs; CF1). The EPCs were transduced using four Moloney murine leukemia retroviruses containing the coding sequences of human OCT4, SOX2, KLF4 and C-MYC and also cultured on irradiated MEFs.

Following all reprogramming experiments, the medium was changed to hiPSC culture medium[35] containing advanced DMEM (Life Technologies), 10% knockout serum replacement (Life Technologies), 2 mM L-glutamine (Life Technologies), 0.007% 2-mercaptoethanol (Sigma-Aldrich) and 4 ng ml⁻¹ recombinant zebrafish fibroblast growth factor 2 (CSCR, University of Cambridge) and 1% penicillin/streptomycin (Life Technologies). Cells with an iPSC morphology first appeared approximately 14 to 28 days after transduction, and undifferentiated colonies (six per donor) were picked between days 28 and 40, transferred onto 12-well MEF-CF1 feeder plates and cultured in hiPSC medium with daily medium changes until ready to passage.

Successful reprogramming was confirmed via genotyping array and expression array[35]. Pluripotency quality control (QC) was performed based on the HipSci QC steps, including the PluriTest using expression microarray data from the Illumina HT12v4 platform and copy number variation and loss of heterozygosity (CNV/LOH) detection using the HumanExome BeadChip Kit platform.

Pluripotent hiPSC lines were transferred onto feeder-free culture conditions, using 10 μg ml⁻¹ Vitronectin XF (Stemcell Technologies)-coated plates and Essential 8 (E8) medium (DMEM/F12 (HAM), E8 supplement (50×) and 1% penicillin/streptomycin; Life Technologies)[35]. The media was changed daily, and cells were passaged every 5–7 days, depending on the confluence and morphology of the cells, at a maximum 1:3 split ratio until established, usually at passage five or six. The passaging method involved washing the confluent plate with PBS and incubating with PBS-EDTA (0.5 mM) for 5–8 min. After removing the PBS-EDTA, cells were resuspended in E8 media and replated onto Vitronectin-coated plates[35]. Once the hiPSCs were established in culture, lines were selected based on morphological qualities (undifferentiated, roundness and compactness of colonies) and expanded for banking and characterization. DNA from fibroblasts and hiPSCs was extracted using Qiagen Chemistry on a QIAcube automated extraction platform.

**HipSci hiPSC line whole-exome genome library preparation and sequencing.** A 96-well plate containing 500 ng genomic DNA in 120 μl was cherry-picked and an Agilent Bravo robot used to transfer the gDNA into a Covaris plate with glass wells and adaptive focused acoustics (AFA) fibers. This plate was then loaded into the LE220 for the shearing process. The sheared DNA was then transferred out of this plate and into an Eppendorf TwinTec 96 plate using the Agilent Bravo robot. Samples were then purified ready for library prep. In this step, the Agilent NGS Workstation transferred AMPure-XP beads and the sheared DNA to a Nunc deep-well plate, then collected, and washed the bead-bound DNA. The DNA was eluted and transferred along with the AMPure-XP beads to a fresh Eppendorf TwinTec plate. Library construction comprised end repair, A-tailing and adapter ligation reactions, performed by a liquid handling robot.

In this step, the Agilent NGS Workstation transferred PEG/NaCl solution and the adapter ligated libraries containing AMPure-XP beads to a Nunc deep-well plate and size-selected the bead-bound DNA. The DNA was eluted and transferred to a fresh Eppendorf TwinTec plate. Agilent Bravo and Labtech Mosquito robotics were used to set up a 384-well quantitative polymerase chain reaction (qPCR) plate, which was then ready to be assayed on the Roche Lightcycler. The Bravo was used to create a qPCR assay plate. This 384-well qPCR plate was then placed in the Roche Lightcycler. A Beckman NX08-02 was used to create an equimolar pool of the indexed adapter ligated libraries. The final pool was then assayed against a known set of standards on the ABI StepOne Plus. The data from the qPCR assay was used to determine the concentration of the equimolar pool. The pool was normalized using Beckman NX08-02. All paired-end sequencing was performed using a range of Illumina HiSeq platforms as the lines were generated over many years (HiSeq 2000 onwards). The sequencing coverage of WGS, WES and hcWES in hiPSC lines are 41×, 72× and 271×, respectively.

**HipSci hiPSC sequence alignment, QC and variant calling.** Reads were aligned to the human genome assembly GRCh37d5 using bwa version 0.5.10 (ref. [68]) ('bwa aln -q 15' and 'bwa sampe') followed by quality score recalibration and indel realignment using GATK version 1.5-9 (ref. [69]) and duplicate marking using biobambam2 version 0.0.147. VerifyBamID version 1.1.3 was used to check for possible contamination of the cell lines, and all but one passed (Supplementary Fig. 4).

Variable sites were called jointly in each fibroblast and hiPSC sample using BCFtools/mpileup and BCFtools/call version 1.4.25. The initial call set was then prefiltered to exclude germline variants that were above 0.1% minor allele frequency in 1000 Genomes phase 3 (ref. [70]) or ExAC 0.3.1 (ref. [71]). For efficiency we also excluded low coverage sites that cannot reach statistical significance and for subsequent analyses considered only sites that had a minimum sequencing depth of 20 or more reads in both the fibroblast and hiPSC and at least 3 reads with a nonreference allele in either the fibroblast or hiPSC sample. At each variable site a Fisher's exact test was performed on a two-by-two contingency table, with rows representing the number of reference and alternate reads and the columns the fibroblast or hiPSC sample. This approach for mutation calling is implemented through BCFtools/ad-bias, and we have adopted it preferentially instead of existing tumor-normal somatic-variant calling tools because, by definition, tools developed for the analysis of tumor-normal data assume that mutations of interest are absent from the normal tissue. However, in our experiment, many mutations were present, albeit at low frequency, in the source tissue fibroblasts.

More information on bcftools ad-bias can be found on the online bcftools-man page at http://samtools.github.io/bcftools/bcftools-man.html. The ad-bias protocol is distributed as a plugin in the main bcftools package, which can be downloaded from http://www.htslib.org/download/. Bcftools ad-bias implements a Fisher test on a $2 \times 2$ contingency table that contains read counts of reference/alternate alleles found in either the iPSC or fibroblast sample. We ran bcftools ad-bias with default settings as follows:

$$bcftools + ad - bias\ exome.bcf - - - t1 - s\ sample.pairs.txt - f$$

$$'\%REF \backslash t \%ALT \backslash t \%CSQ \backslash t \%INFO/ExAC \backslash t \%INFO/UK1KG'$$

where 'exome.bcf' was the BCF file created by our variant calling pipeline, described in Methods, and sample.pairs.txt was a file that contained matched pairs of the iPSC and corresponding fibroblast sample, one per line, as follows:

$$HPSI0213i - koun\_2 \quad HPSI0213pf - koun$$

$$HPSI0213i - nawk\_55 \quad HPSI0213pf - nawk.$$

$$HPSI0313i - airc\_2 \quad HPSI0313pf - airc$$

We corrected for the total number of tests (84.8 M) using the Benjamini–Hochberg procedure at a false discovery rate of 5%, equivalent to a $P$ value threshold of $9.9 \times 10^{-4}$, to call a mutation as a significant change in allele frequency between the fibroblast and iPSC samples. Furthermore, we annotated sites from regions of low mappability and sites that overlapped with copy-number alterations previously called from array genotypes[35] and removed sites that had greater than 0.6 alternate allele frequency in either the fibroblast or hiPSC, as these sites are likely to be enriched for false positives. Dinucleotide mutations were called by sorting mutations occurring in the same iPSC line by genomic position and marking mutations that were immediately adjacent as dinucleotides.

**Mutation calling of hiPSCs derived from S2, S7 and ten HipSci lines by using fibroblasts as germline controls.** Single substitutions were called using CaVEMan (Cancer Variants Through Expectation Maximization; http://cancerit.github.io/CaVEMan/) algorithm[72]. To avoid mapping artefacts, we removed variants with a median alignment score <90 and those with a clipping index >0. Indels were called using cgpPindel (http://cancerit.github.io/cgpPindel/). We discarded indels that occurred in repeat regions with repeat count >10 and variant call format (VCF) quality <250. Double substitutions were identified as two adjacent single substitutions called by CaVEMan. The ten HipSci lines are HPSI0714i-iudw_4, HPSI0914i-laey_4, HPSI0114i-eipl_1, HPSI0414i-oaqd_2, HPSI0414i-oaqd_3, HPSI1014i-quls_2, HPSI1013i-yemz_3, HPSI0614i-paab_3, HPSI1113i-qorq_2 and HPSI0215i-fawm_4.

**Mutational signature analysis.** Mutational signature analysis was performed on S7 EPC-hiPSCs, S2 F-hiPSCs, S2 EPC-hiPSCs and the HipSci F-hiPSC WGS dataset. All dinucleotide mutations were excluded from this analysis. We generated 96-channel single substitution profiles for 324 hiPSCs and 204 fibroblasts. We fitted previously discovered skin-specific substitutions to each sample using an R package (signature.tools.lib)[31]. Function SignatureFit_withBootstrap() was used with default parameters. In downstream analysis, the exposure of two UV-caused signatures Skin_D and Skin_J were summed up to represent the total signature exposure caused by UV (signature 7). A de novo signature extraction was performed on 324 WGS HipSci F-hiPSCs to confirm that the UV-associated skin signatures (Skin_D and Skin_J, signature 7) and culture-associated one (Skin_A, signature 18) are also the most prominent signatures identified in de novo signature extraction (Supplementary Fig. 5).

**Analysis of C>T/CC»TT transcriptional strand bias in replication timing regions.** Reference information of replication timing regions were obtained from Repli-seq data of the ENCODE project (https://www.encodeproject.org/)[73]. The transcriptional strand coordinates were inferred from the known footprints and transcriptional direction of protein coding genes. In our dataset, we first orientated all G>A and GG>AA to C>T and CC>TT (using pyrimidine as the mutated base). Then, we mapped C>T and CC>TT to the genomic coordinates of all gene footprints and replication timing regions. Lastly, we counted the number of C>T/CC>TT mutations on transcribed and nontranscribed gene regions in different replication timing regions.

**Identification of fibroblast-shared mutations and private mutations in HipSci F-hiPSCs.** We classified mutations (substitutions and indels) in HipSci F-hiPSCs into fibroblast-shared mutations and private mutations. Fibroblast-shared mutations in hiPSCs are the ones that have at least one read from the mutant allele found in the corresponding fibroblast. Private mutations are the ones that have no reads from the mutant allele in the fibroblast. Mutational signature fitting was performed separately for fibroblast-shared substitutions and private substitutions in hiPSCs. For indels, only the percentage of different indel types was compared between fibroblast-shared indels and private indels.

**Clonality of samples.** We inspected the distribution of VAFs of substitutions in HipSci fibroblasts and HipSci F-hiPSCs. Almost all hiPSCs had VAFs distributed around 50%, indicating that they were clonal. In contrast, all fibroblasts had lower VAFs, which distributed around 25% or lower, indicating that they were oligoclonal. We computed kernel density estimates for VAF distributions of each sample. Based on the kernel density estimation, the number of clusters in a VAF distribution was determined by identification of the local maximum. Accordingly, the size of each cluster was estimated by summing up mutations having VAF between two local minimums.

**Variant consequence annotation.** Variant consequences were calculated using the Variant Effect Predictor[74] and BCFtools/csq[75]. For dinucleotide mutations, we recorded only the most impactful consequence of either of the two members of the dinucleotide, where the scale from least to most impactful was intergenic, intronic, synonymous, 3′ untranslated region, 5′ untranslated region, splice region, missense, splice donor, splice acceptor, start lost, stop lost and stop gained. We identified overlaps with putative cancer driver mutations using the COSMIC 'All Mutations in Census Genes' mutation list (CosmicMutantExportCensus.tsv.gz) version 92, 27 August 2020.

**dNdScv analysis.** To detect genes under positive selection, we used dN/dS ratios as implemented in the dNdScv R package (https://github.com/im3sanger/dndscv)[49]. dNdScv uses maximum likelihood models to calculate the ratio of nonsynonymous to synonymous mutations per gene, normalized by sequence composition, trinucleotide substitution rates and the local mutability of each gene based on epigenetic covariates.

Three analyses were run for 452 F-hiPSCs and 78 B-hiPSCs sequencing data:

(1) default dNdScv (exome-wide, looking at all genes in the genome or exome for selection);

(2) restricted hypothesis testing of known cancer genes (to increase the statistical power on known drivers, using the gene list from Martincorena et al.[49]); and

(3) detection of mutational hotspots (using the sitednds function in dNdScv on hotspots detected in The Cancer Genome Atlas).

**Insignia B-hiPSC line generation, growth, QC and sequencing.** Erythroblasts were derived from PBMCs, following appropriate ethics committee approvals (REC 13/EE/0302), and reprogrammed using the nonintegrating CytoTune Sendai virus reprogramming kit (OCT3/4, SOX2, KLF4 and C-MYC) by the Cellular Generation and Phenotyping facility at the Wellcome Sanger Institute in the same way as for the HipSci lines (described above). After establishment of B-hiPSCs lines that had passed all QC steps (described above) and at cell passage equivalent to about 30 doublings, expanded clones were single-cell subcloned to generate two to four daughter subclones for each B-hiPSC line. WGS was run on germline, erythroblasts, B-hiPSC parental clones and B-hiPSC subclones. The average sequencing coverage of WGS was 38× (Supplementary Table 14). Single-nucleotide polymorphism genotyping was performed as a QC measure to ensure matches between all hiPSCs and respective original starting sample. RNA sequencing was run on 78 iPSC parental clones.

**Insignia B-hiPSC mutation calling by using blood as germline controls.** Single substitutions were called using CaVEMan (Cancer Variants Through Expectation Maximization; http://cancerit.github.io/CaVEMan/) algorithm[72]. To avoid mapping artefacts, we removed variants with a median alignment score <140 and those with a clipping index >0. Indels were called using cgpPindel (http://cancerit.github.io/cgpPindel/). We discarded indels that occurred in repeat regions with repeat count >10 and VCF quality <250. Double substitutions were identified as two adjacent single substitutions called by CaVEMan. Mutation calls were obtained for erythroblasts, iPSC parental clones and subclones.

**Differentiation of Insignia B-hiPSCs (*BCOR* mutant and *BCOR* wild-type) and RNA sequencing.** The *BCOR*-mutant B-hiPSCs (MSH40i2, MSH93i6) and the *BCOR*-wild-type B-hiPSCs (MSH34i2, MSH30i3) were maintained in feeder-free conditions cultured in Essential E8 medium (ThermoFisher Scientific, A1517001) on Vitronectin FX (Stemcell Technologies, 07180)-coated plates. hiPSC medium was changed daily, and the cells were monitored to ensure there were no signs of spontaneous differentiation. hiPSCs were expanded every 3 or 4 days as small clumps using 0.5 mM UltraPure EDTA (ThermoFisher Scientific, 1557020) diluted in Dulbecco's phosphate buffered saline (DPBS) (ThermoFisher Scientific, 14190342).

Before neural induction, three independent replicates of hiPSC from each donor line were generated and cultured for 1 week as described above. Healthy, nondifferentiating hiPSCs colonies were dissociated into single-cell suspension using TrypLE Express Enzyme (ThermoFisher Scientific, 12605010) and plated on Vitronectin FX-coated plates at 50,000 cells/cm² density in the presence of RevitalCell Supplement (ThermoFisher Scientific, A26445-01, lot 2170092). The cells were cultured for another 2 days until they reached 60–75% confluence. At day 0, the culture medium was switched to neural induction medium (NIM) containing V/V DMEMF12 HEPES (ThermoFisher Scientific, 11330032, lot 2186798) and neurobasal medium (ThermoFisher Scientific, 21103-049, lot 2161553), 1× B-27 Supplement (ThermoFisher Scientific, 17504-044, lot 2188886), 1× N2 Supplement (ThermoFisher Scientific, 17502-048; Lot: 2193551), MEM NEAA (ThermoFisher Scientific, 11140-035, lot 2202923), 1× Glutamax-I (ThermoFisher Scientific, 35050-061, lot 2085268), 1× penicillin/streptomycin in the presence of 10 μM SB431542 (Tocris, 1414/10) and 200 nM LDN193189 (Tocris, 6053/10) with an addition of 1× RevitaCell. Starting from day 1, NIM without RevitaCell was changed every day until day 12.

At the end of the neural induction process (day 12), the cells were dissociated into single-cell suspension using TrypLE Express Enzyme and plated at high cell density (200,000 cells/cm²) in double-coated plates of PDL (ThermoFisher Scientific, A3890401, lot 881772E) and 15 μg ml⁻¹ Cultrex mLaminin I Pathclear (Biotechne, 3400-010-02; Lot: 1594368). The NIM was switched to neuron differentiation medium (NDM) containing BrainPhys Neural Medium (Stemcell Technologies, 05790, batch 1000031535), 1× B-27 Supplement (ThermoFisher Scientific, 17504-044, lot 2188886), 1× N2 Supplement (ThermoFisher Scientific, 17502-048, lot 2193551), 50 μM dibutyryl-cAMP, sodium salt (Tocris, 1141/50), 200 nM L-ascorbic acid (Tocris, 4055), 20 ng ml⁻¹ BDNF (Cambridge Bioscience, GFH1-100), 20 ng ml⁻¹ GDNF (Cambridge Bioscience, GFM37-100) in presence of 10 μM Y-27632 (Tocris, 1254/10). On day 13, the medium was changed to NDM without Y-27632, and the cells were allowed to differentiate for another 14 days. Two thirds of the medium was changed three times a week.

During the cell differentiation process, cell pellets from all culture replicates were harvested at days 0, 6, 12 and 27 (endpoint) for an RNA-sequencing serial time study. Immunostaining characterization was performed at days 0, 12 and 27 of differentiation to assess the differentiation efficiency.

Total RNA was extracted using PureLink RNA Mini Kit (ThermoFisher Scientific, 12183018 A) following the manufacturer's recommendations. The RNA was quality controlled; the cDNA libraries were prepared and sequenced using Illumina NovaSeq 6000 technology. Each sequenced sample had ≥20 million read pairs of 150-bp paired-end reads.

**Processing RNA-sequencing data.** Splice-aware STAR v2.5.0a[76] was used to map RNA-sequencing data to the reference genome. For the human decoy reference

genome hs37d5.fa.gz, a genome index was first generated. Then, using the splice junction information from Gencode GTF annotation file v19, fastq files were mapped. The fragments of reads linked with the gene features were then counted using featureCounts v2.0.1 (ref. [77]). The samples' raw counts matrices were then analyzed in R version 4.0.4. Differential gene expression was performed using the DESeq2 R package.

**Immunofluorescence staining.** The expression of pluripotency markers at day 0 (hiPSC) of differentiation was assessed using a commercially available PSC (OCT4, SSEA4) Immunocytochemistry Kit (ThermoFisher Scientific, A25526, lot 2194558).

NSCs at day 12 and neurons at day 27 of differentiation were stained as already described before with minor modifications[78]. Briefly, the medium was discarded from the plates, and the cells were rinsed gently with DPBS. A 4% solution of paraformaldehyde was used to fix the cells for 20 min at room temperature. The cells were rinsed twice with DPBS and permeabilized for 20 min with 0.1% Triton X-100 (Sigma-Aldrich, T8787-50ML). Nonspecific epitopes were blocked with 0.5% BSA solution for 1 h at room temperature. Cells were incubated overnight at 4 °C with the primary antibodies as follows: on day 12, cells were incubated with an anti-PAX6 antibody (ThermoFisher Scientific, 14-9914-82, dilution 1:100); on day 27, cells were incubated with an anti-Tubulin beta III (TUBB3) (Millipore, MAB1637, dilution 1:400). The cells were then rinsed three times with 1× DPBS and incubated with the secondary antibody Alexa Fluor 488 donkey anti-mouse (ThermoFisher Scientific, A21202, dilution 1:500). The cells were rinsed three times with DPBS and the nuclei counterstained with NucBlue Fixed Cell Stain ReadyProbes (ThermoFisher Scientific, R37606). The images were acquired within 48 h using EVOS FL Auto 2 microscope (ThermoFisher Scientific, AMAFD2000), and the figures were made using the FigureJ plugin in ImageJ software.

**Statistics and reproducibility.** All statistical analyses were performed in R[79]. The effects of age and sex on mutation burden of F-hiPSCs were estimated using Mann–Whitney test, 'wilcox.test()' in R. Tests for correlation in the study were performed using 'cor.test()' in R.

For cancer driver mutations identified in HipSci F-hiPSCs, a two-sided Fisher test was used to call a mutation as a significant change in allele frequency between the fibroblast and iPSC samples (Supplementary Table 5). A Benjamini–Hochberg procedure for multiple hypothesis testing was used.

For the differentiation and immunostaining experiments, each *BCOR*-mut and *BCOR*-wt cell line had three independent biological replicates differentiated, and for each of these replicates, three wells were stained and imaged using immunofluorescence. At every stage of the neural differentiation (day 0, day 12 and day 27), a total of 36 images were analyzed for both *BCOR*-mut and *BCOR*-wt cell lines.

Differential gene expression analysis of Insignia B-hiPSCs was performed using DESeq2, which fits each gene's negative binomial generalized linear model. The default DESeq2 Wald test was used for significance testing, and a threshold of <0.05 of the adjusted *P* value was applied (Supplementary Table 13).

**Reporting summary.** Further information on research design is available in the Nature Research Reporting Summary linked to this article.

## Data availability

All datasets generated as part of this study are included in this data availability statement with no omissions. Links to raw sequencing data produced in this study are available from the HipSci project website (WGS data: https://www.hipsci.org/lines/#/assays/wgs; WES data https://www.hipsci.org/lines/#/assays/exomeseq). Raw data have been deposited in the European Nucleotide Archive under accession numbers ERP006946 (WES, open access samples) and ERP017015 (WGS, open access samples) and the European Genotype-phenotype Archive under accession number EGAS00001000592 (WES, managed access samples involving healthy donors, following completion and approval of a data Access Agreement via eDAM at the Welcome Sanger Institute; https://www.hipsci.org/data#faq). The raw sequence files of Insignia samples are deposited at the European Genome-phenome Archive with accession number EGAD00001007029 (WGS, open access samples, https://ega-archive.org/datasets/EGAD00001007029). The open access data samples are freely available to download, whereas the managed access data are available following a request and a data access agreement (via Wellcome Sanger Institute electronic Data Access Mechanism). The variant call sets are deposited at Mendeley (https://data.mendeley.com/datasets/6rfc2xrnyd/1)[80].

## Code availability

All code used for analysis is detailed in Methods and the Reporting summary. The code of bespoke software pertaining to data processing and analysis is on GitHub (https://github.com/dg13/ips-seq). The code of statistical analysis and figures is on GitHub (https://github.com/Nik-Zainal-Group/hiPSCs_BCOR.git). Differential gene expression analysis was performed using the DESeq2 R package (https://github.com/mikelove/DESeq2)[81]. Signature fitting was conducted using the signature.tools.lib R package (https://github.com/Nik-Zainal-Group/signature.tools.lib)[31]. dN/dS ratios were calculated using the dNdScv R package (https://github.com/im3sanger/dndscv)[49].

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

## Acknowledgements

This work was funded by Cancer Research UK Advanced Clinician Scientist Award (C60100/A23916), Dr Josef Steiner Cancer Research Award 2019, Medical Research Council Grant-in-Aid to the Medical Research Council Cancer Unit, Cancer Research UK Pioneer Award and Wellcome Intermediate Clinical Fellowship (WT100183), all awarded to S.N.-Z. and supported by NIHR-BRC Cambridge core grant (BRC-125-20014) and UK Regenerative Medicine Platform (MR/R015724/1). The funders had no role in study design, data collection and analysis, decision to publish or preparation of the article, and the views expressed are those of the author(s) and not necessarily those of the NIHR or the Department of Health and Social Care. We thank the individuals who participated in the studies. We are grateful to R. Barker, P. Andrews and Z. Hewitt from the UK Regenerative Medicine Platform for helpful perspectives and comments on the article.

## Author contributions

F.J.R., X.Z., P.D., C.B., D.G. and S.N.-Z. conceived this study, designed the experiments and wrote the article. R.D. contributed to the methodology of the study. X.Z., C.B., T.D.A., G.K., Q.W., Y.M. and I.M. performed bioinformatic analysis of the sequencing data. C.B. conducted experiments for *BCOR* functional validation. A.R.B. reviewed and edited the article.

## Competing interests

S.N.-Z. holds patents on mutational signature based clinical algorithms although these are not relevant to the research in this paper. The remaining authors declare no competing interests.

## Additional information

**Extended data** is available for this paper at https://doi.org/10.1038/s41588-022-01147-3.

**Correspondence and requests for materials** should be addressed to Serena Nik-Zainal.

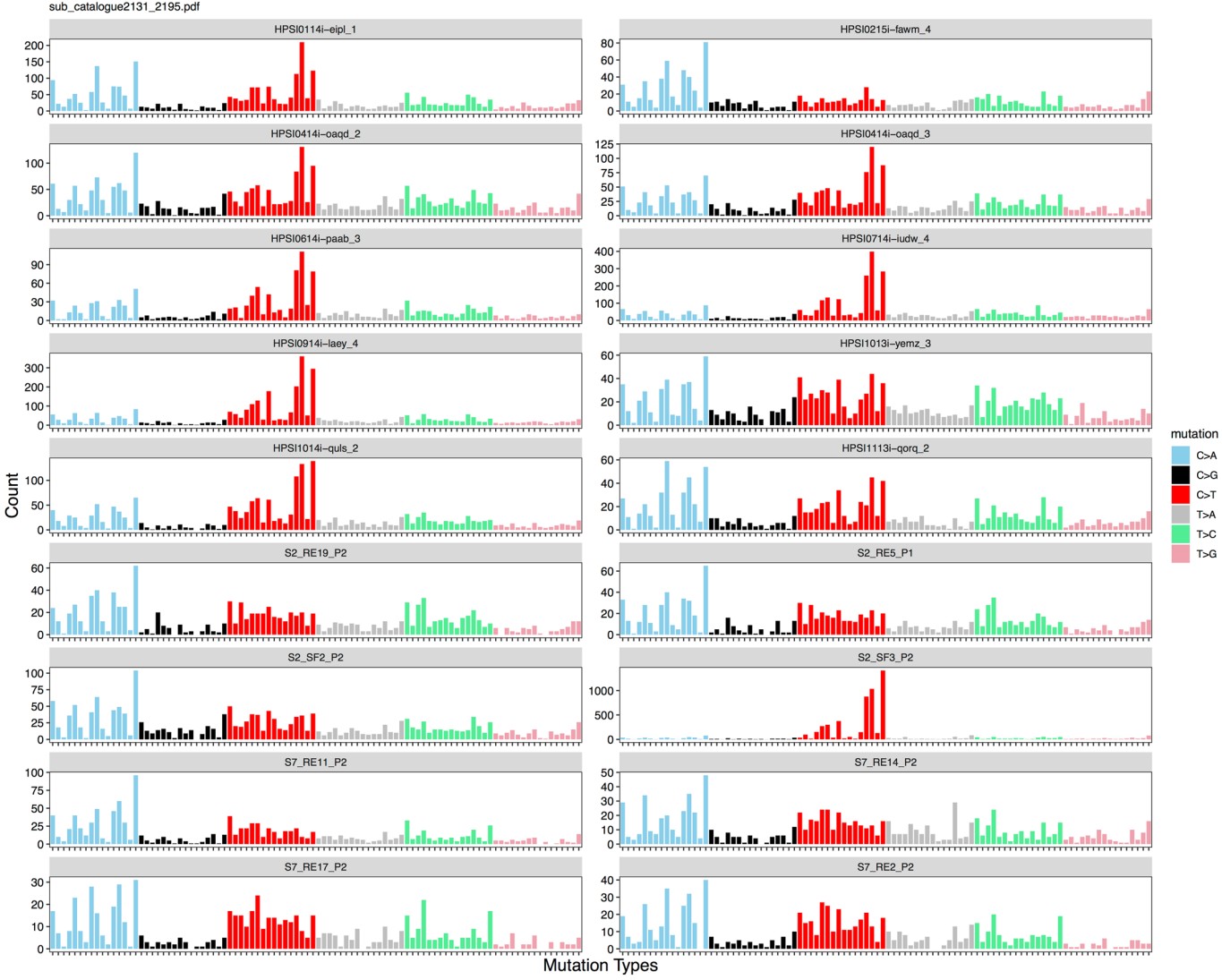

**Extended Data Fig. 1 | Mutational signatures of hiPSCs.** Mutational profiles of 18 blood-derived human induced pluripotent stem cells (hiPSCs) and skin-derived hiPSCs featured in Fig. 1.

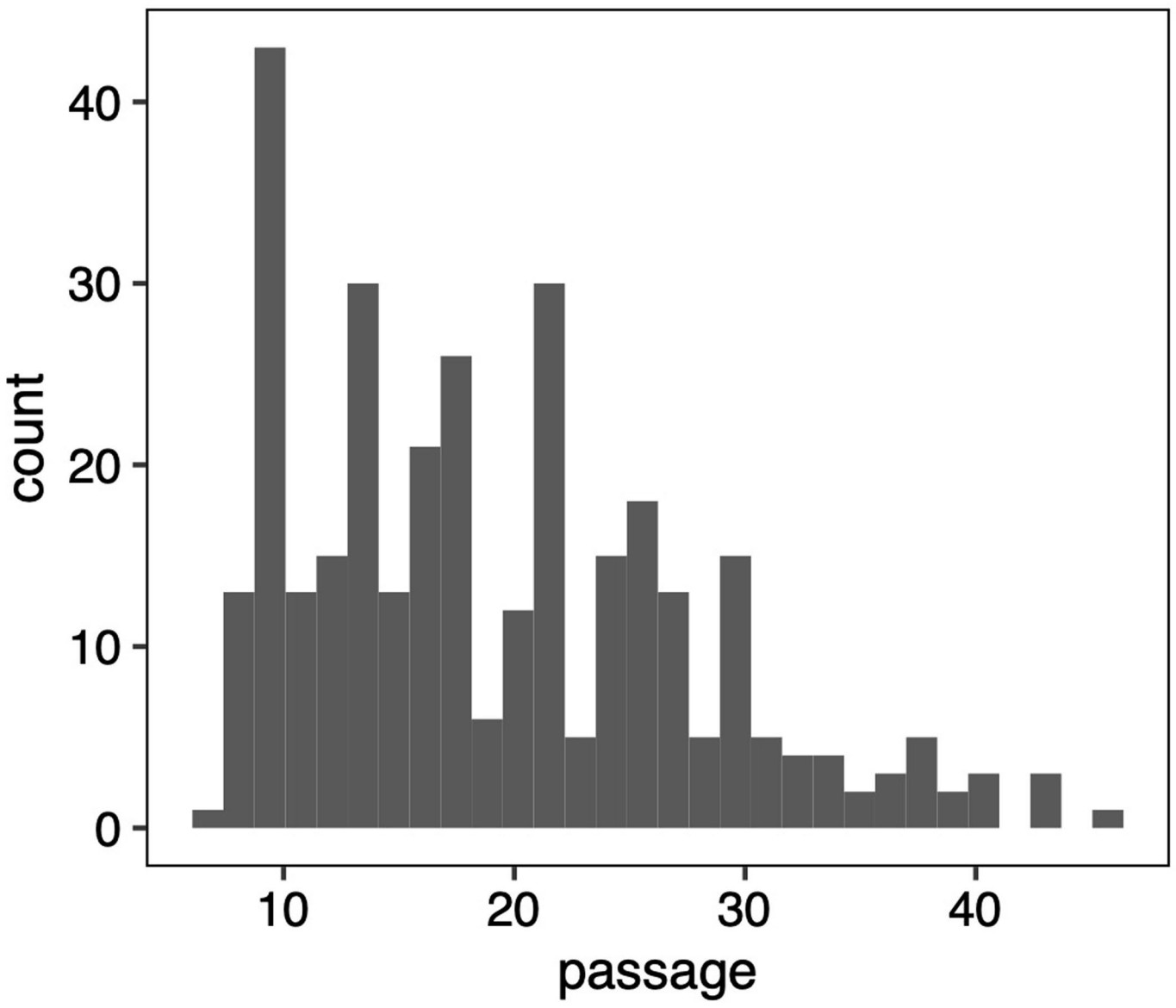

**Extended Data Fig. 2 | Distribution of passage number of HipSci F-hiPSCs.** The median is 18. n = 288 fibroblasts. Passage (P) numbers of eight non-HipSci hiPSCs are: P2 for: S7_RE11, S7_RE14, S7_RE2, S7_RE17, S2_RE19, S2_SF3 and S2_SF2; P1 for S2_RE5.

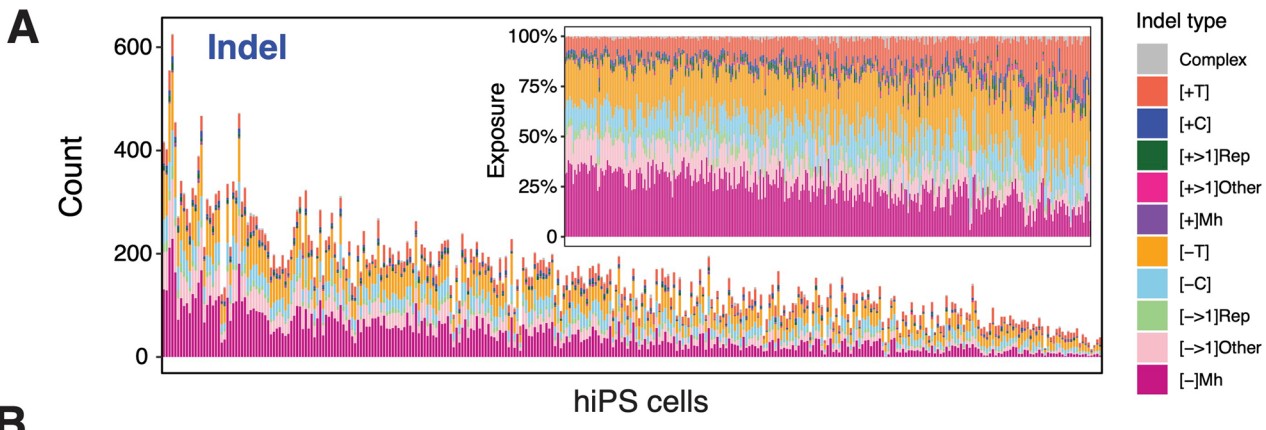

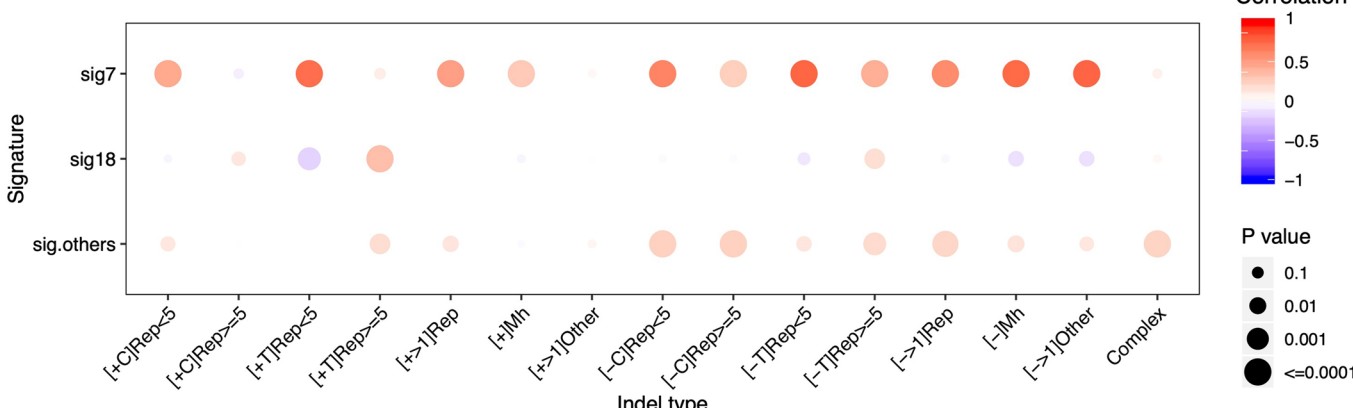

**Extended Data Fig. 3 | HipSci F-hiPSC mutational signatures. A**, Distribution of mutational signatures in 324 fibroblast-derived iPSC lines. The inset figure shows the relative exposures of mutational signatures/indel types. **B**, Correlation between substitution signatures and indel types (Pearson's correlation test, two-sided).

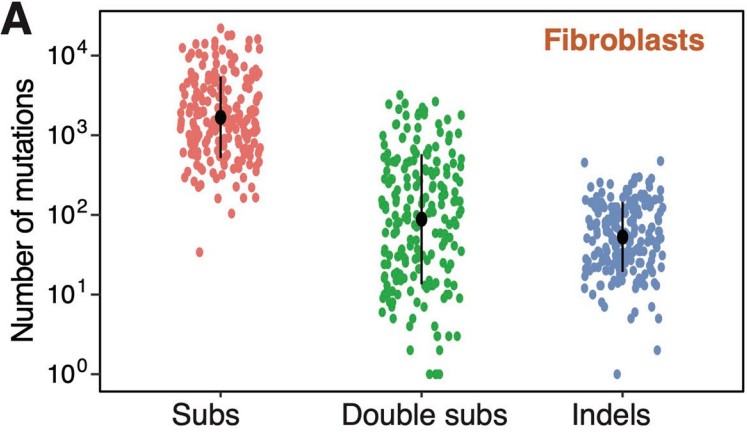

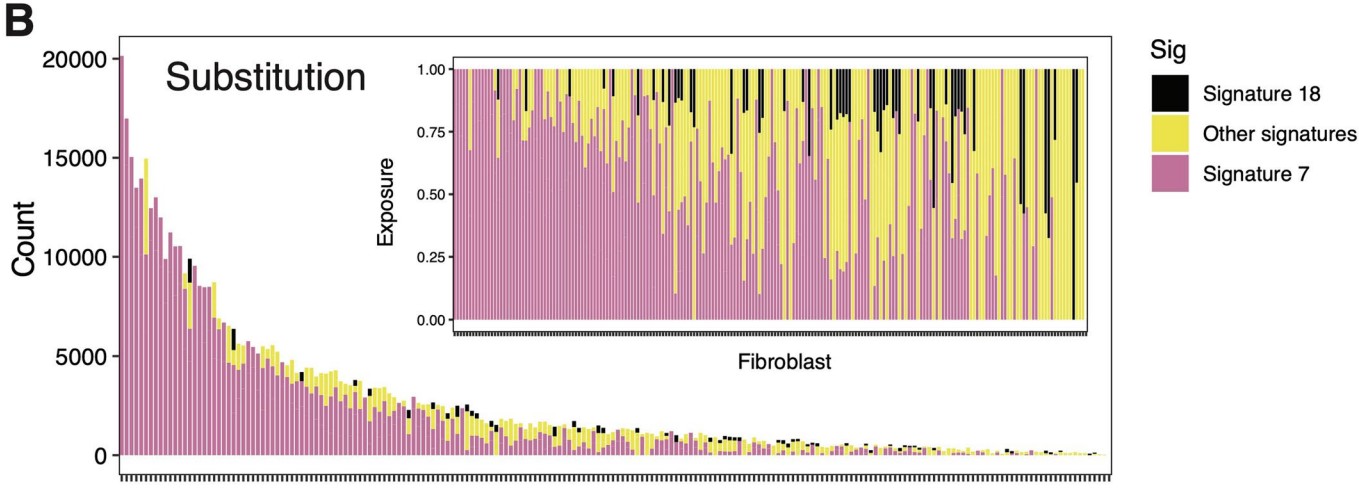

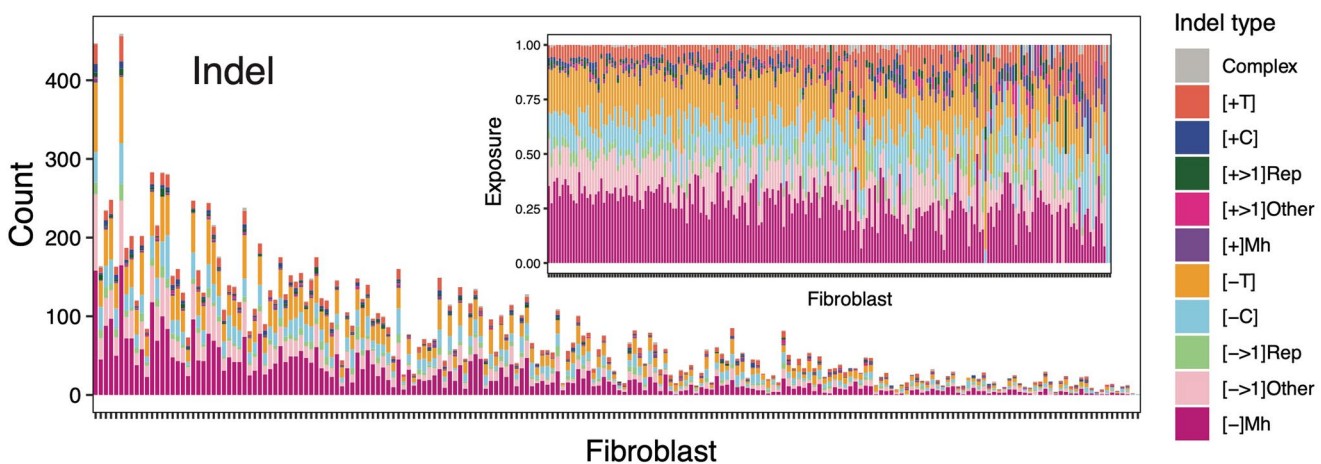

**Extended Data Fig. 4 | Mutation burden and mutational signatures in fibroblasts. A**, Mutation burden of substitutions, CC>TT double substitutions and indels in fibroblasts. Black dots and error bars represent mean ± SD of fibroblast observations, n = 204 Whole Genome Sequencing (WGS) of fibroblasts. **B**, The amount of each mutational signature and indel type (exposure) in fibroblasts.

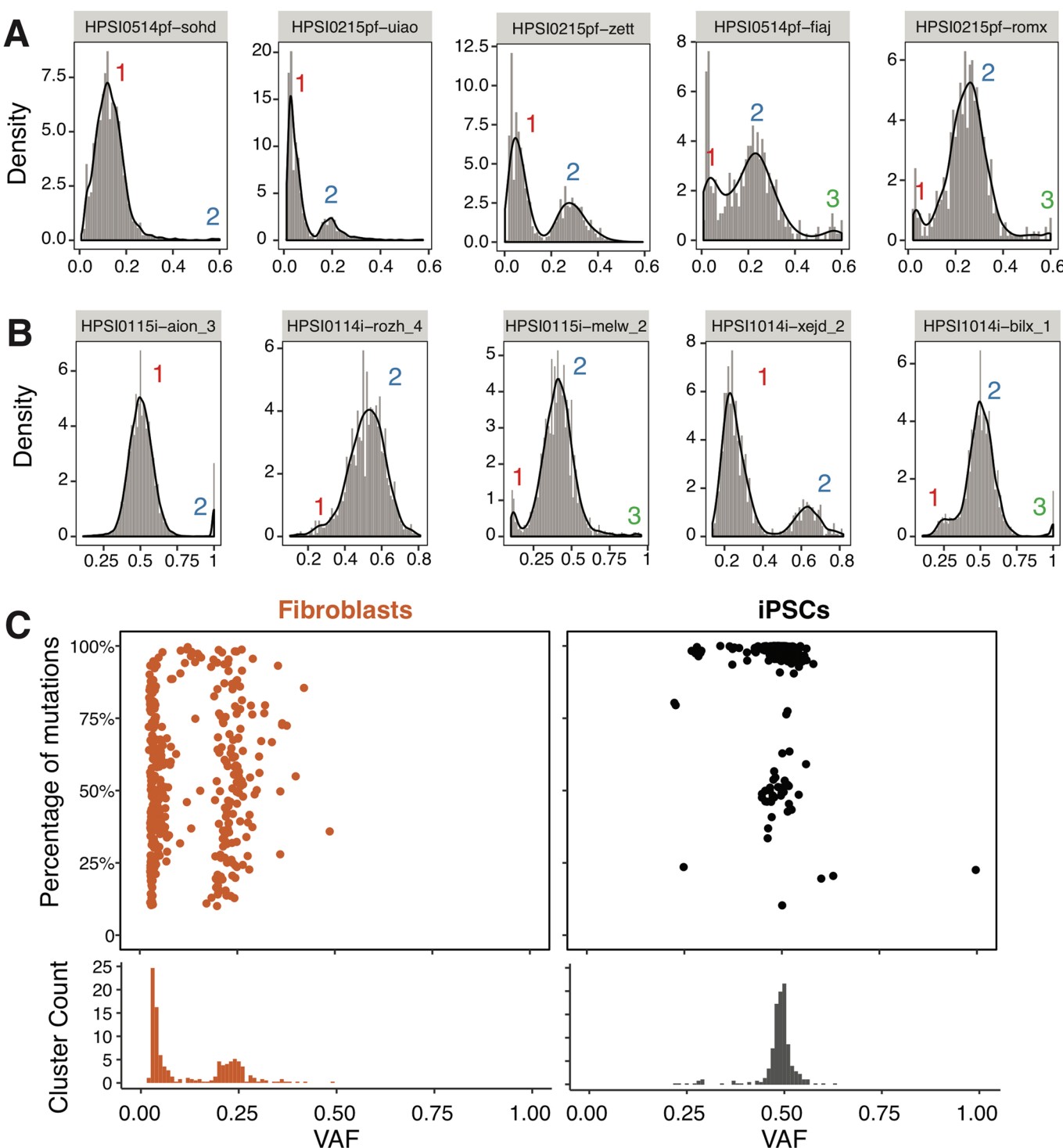

**Extended Data Fig. 5 | Analysis of Variant Allele Frequency in fibroblasts and iPSCs.** Distribution of Variant Allele Frequency (VAF) distribution of five fibroblasts and five hiPSCs are shown in **(A)** and **(B)**, respectively. Kernel density estimation was used to smooth the distribution. Local maximums and minimums were calculated to identify subclonal clusters. **C**, Summary of subclonal clusters in fibroblasts (n = 204) and hiPSCs (n = 324). Each dot represents a cluster which has at least 10% of total mutations in the sample. Most of the fibroblasts are polyclonal with VAF of a cluster of nearly 0.25, whereas hiPSCs are mostly clonal with VAF of nearly 0.5.

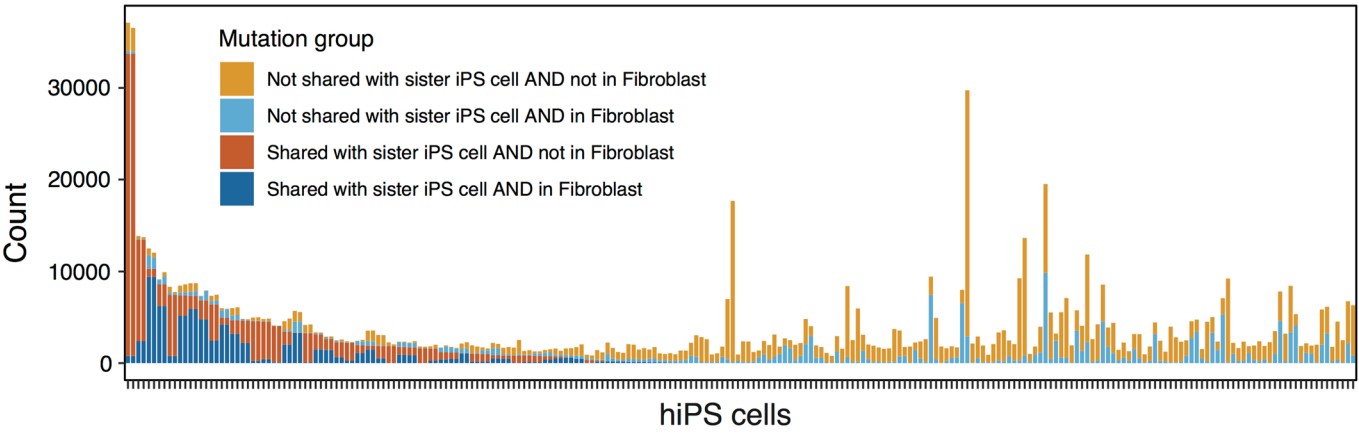

**Extended Data Fig. 6 | Shared mutations.** Histogram showing shared mutations between hiPSCs and the matched fibroblasts.

muts_NotInCGPSet_catalogue.pdf

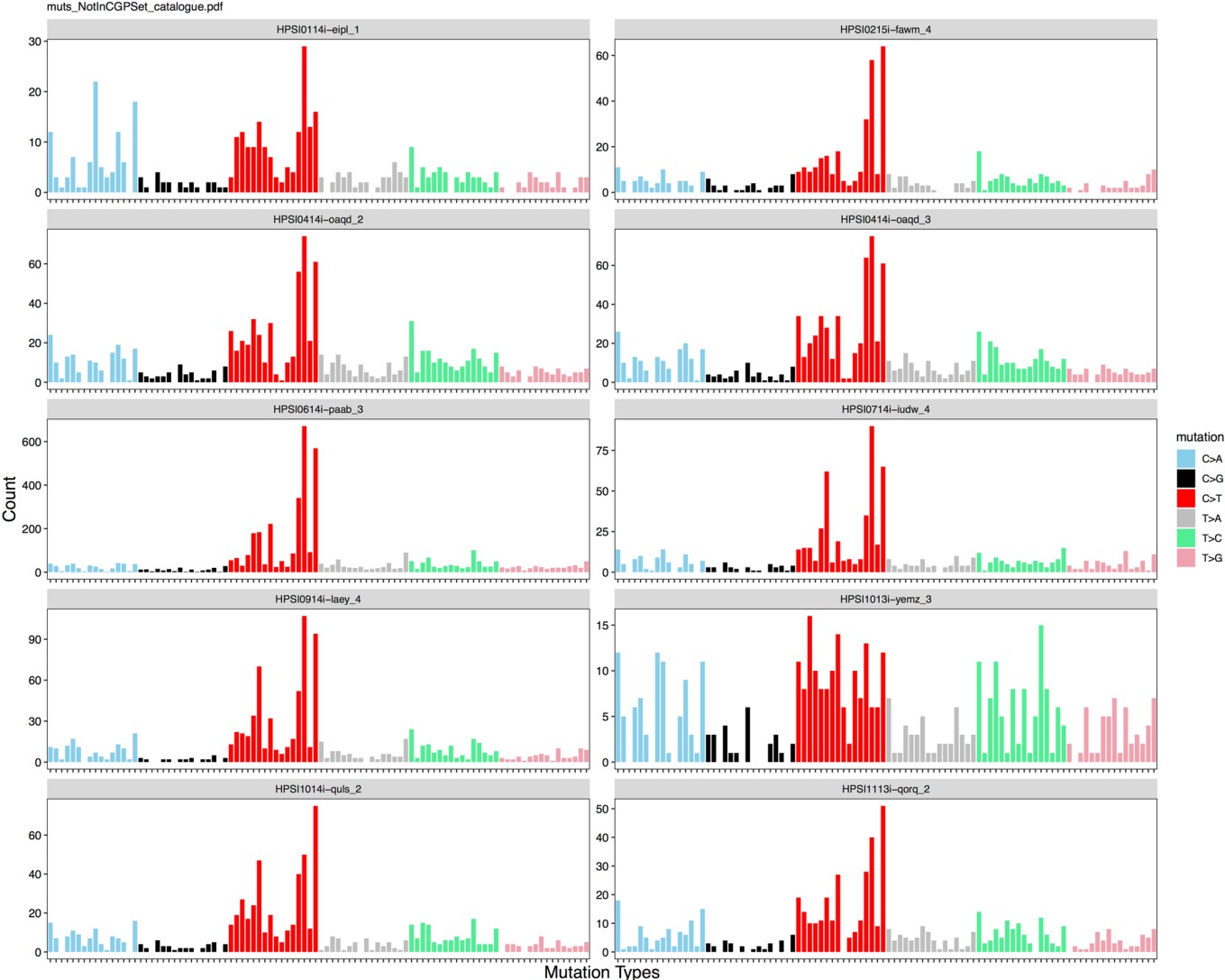

**Extended Data Fig. 7 | Mutational signatures of shared mutations.** Mutation profile of substitutions that were removed from hiPSCs by using fibroblast as 'normal' for ten HipSci samples from Fig. 1. These removed mutations are not germline Single Nucleotide Polymorphisms (SNPs), and are mostly composed of mutations that are typical of ultraviolet (UV) light exposure.

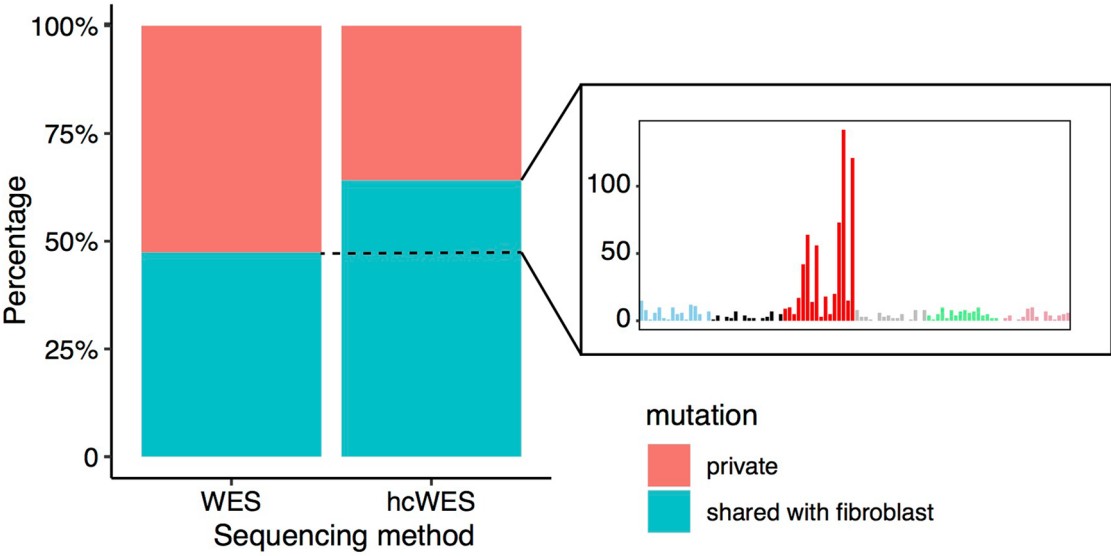

**Extended Data Fig. 8 | Comparison of WES (72X) and high-coverage WES (hcWES, 271X) of fibroblasts.** More mutations in hiPSCs were discovered in fibroblasts (shared mutations) through high-coverage Whole Exome Sequencing (hcWES) than through WES, resulting in the percentage of shared mutations increased in hcWES data. Interestingly, the mutational profile of these increased shared mutations is very similar to the UV signature, indicating that increasing sequencing depth enables more UV-caused somatic mutations that were found in hiPSCs to also be detected in the corresponding fibroblasts.

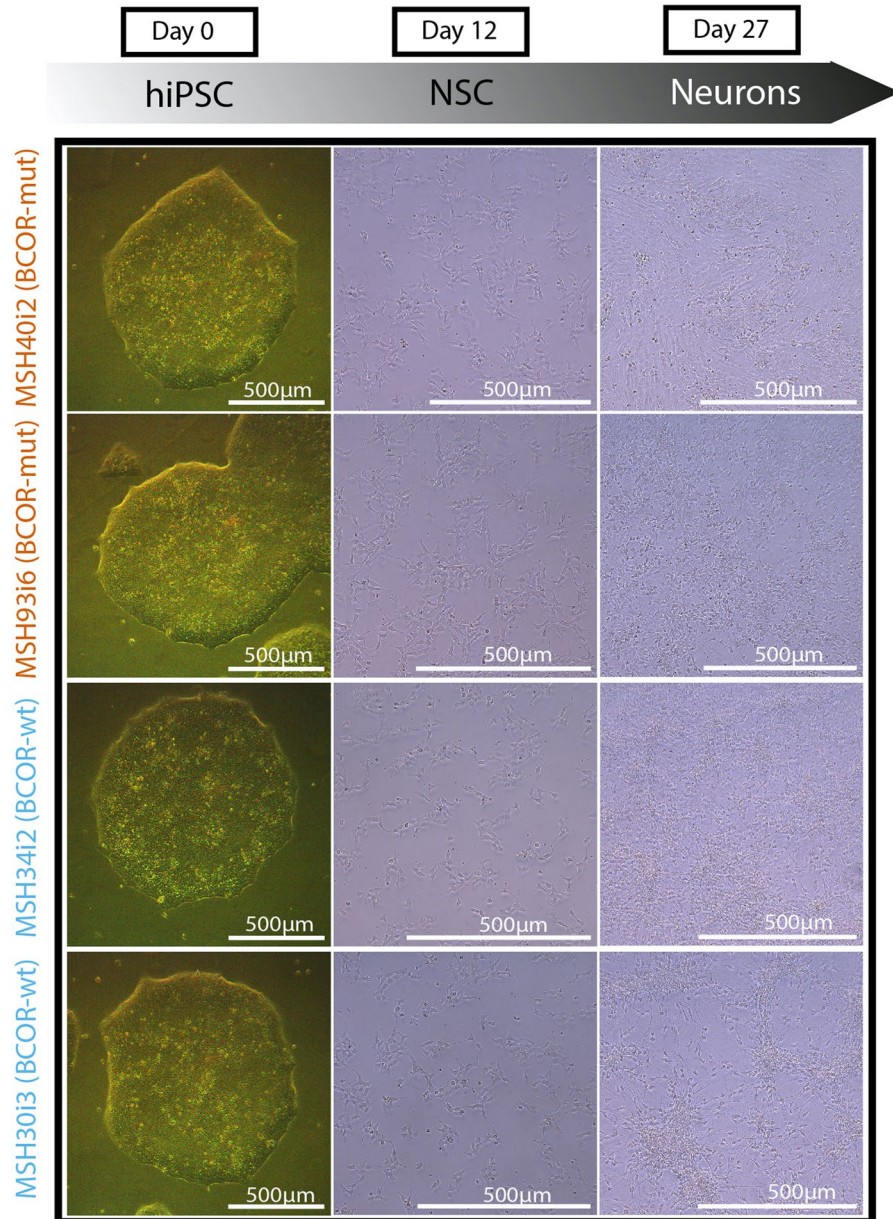

**Extended Data Fig. 9 | Brightfield images showing cell morphology changes during neural differentiation stages.** HiPSCs with *BCOR* mutations (*BCOR*-mut) samples have normal hiPSC colony morphology (Day 0). The differentiation of cells into neurons (Day 27) showed fewer differentiated cells in *BCOR*-mut compared to hiPSCs with no *BCOR* mutations (*BCOR*-wt).

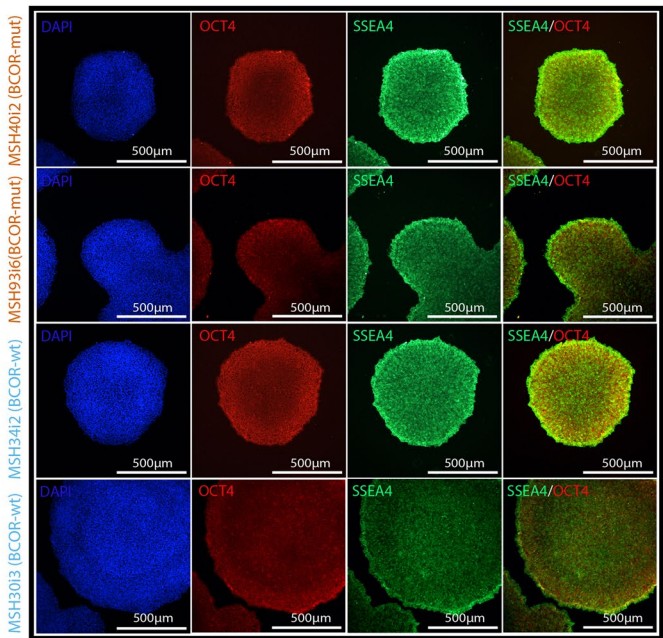

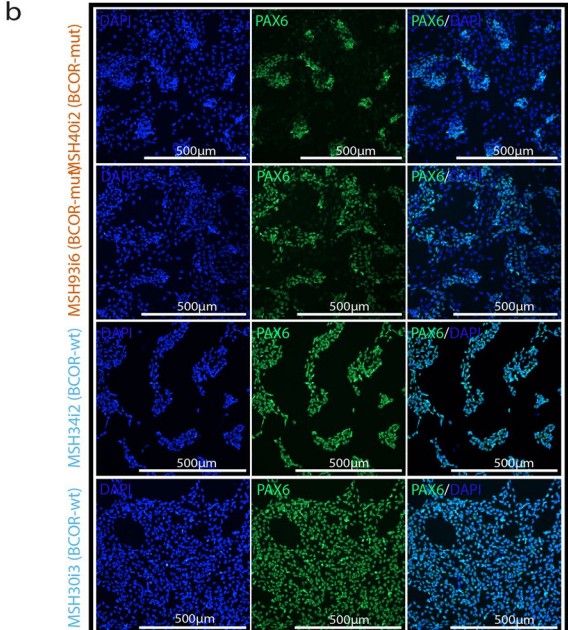

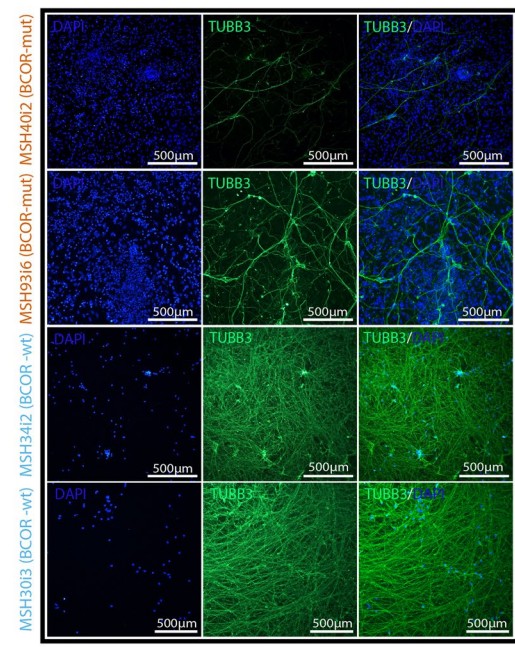

**Extended Data Fig. 10 | Immunofluorescence characterization of *BCOR*-mut and *BCOR*-wt at different neural differentiation stages. a**, At the hiPSC stage, *BCOR*-mut colonies are indistinguishable from *BCOR*-mut, both expressing SSEA4 (green) and OCT4/POU5F1 (red). **b,c**, *BCOR*-mut cells have impaired neural differentiation resulting in fewer Pax 6 positive cells at Day 12 (**b**) and TUBB3 positive cells at Day 27 (**c**).

# Reporting Summary

## Statistics

For all statistical analyses, confirm that the following items are present in the figure legend, table legend, main text, or Methods section.

| n/a | Confirmed | |
|---|---|---|
| ☐ | ☒ | The exact sample size (*n*) for each experimental group/condition, given as a discrete number and unit of measurement |
| ☐ | ☒ | A statement on whether measurements were taken from distinct samples or whether the same sample was measured repeatedly |
| ☐ | ☒ | The statistical test(s) used AND whether they are one- or two-sided |
| | | *Only common tests should be described solely by name; describe more complex techniques in the Methods section.* |
| ☒ | ☐ | A description of all covariates tested |
| ☒ | ☐ | A description of any assumptions or corrections, such as tests of normality and adjustment for multiple comparisons |
| ☐ | ☒ | A full description of the statistical parameters including central tendency (e.g. means) or other basic estimates (e.g. regression coefficient) AND variation (e.g. standard deviation) or associated estimates of uncertainty (e.g. confidence intervals) |
| ☐ | ☒ | For null hypothesis testing, the test statistic (e.g. *F*, *t*, *r*) with confidence intervals, effect sizes, degrees of freedom and *P* value noted |
| | | *Give P values as exact values whenever suitable.* |
| ☒ | ☐ | For Bayesian analysis, information on the choice of priors and Markov chain Monte Carlo settings |
| ☒ | ☐ | For hierarchical and complex designs, identification of the appropriate level for tests and full reporting of outcomes |
| ☐ | ☒ | Estimates of effect sizes (e.g. Cohen's *d*, Pearson's *r*), indicating how they were calculated |

*Our web collection on statistics for biologists contains articles on many of the points above.*

## Software and code

Policy information about availability of computer code

| | |
|---|---|
| Data collection | CaVEMan (http://cancerit.github.io/CaVEMan/) (Jones et al., 2016), Pindel (http://cancerit.github.io/cgpPindel) (Raine et al., 2015). The code of bespoke software is on github: https://github.com/dg13/ips-seq. |
| Data analysis | All code used for analysis are detailed in the Methods section. The code of statistical analysis and figures is on github: https://github.com/Nik-Zainal-Group/hiPSCs_BCOR.git. dN/dS ratios were calculated using dNdScv R package (https://github.com/im3sanger/dndscv) |

For manuscripts utilizing custom algorithms or software that are central to the research but not yet described in published literature, software must be made available to editors and reviewers. We strongly encourage code deposition in a community repository (e.g. GitHub). See the Nature Portfolio guidelines for submitting code & software for further information.

## Data

Policy information about availability of data

All manuscripts must include a data availability statement. This statement should provide the following information, where applicable:
- Accession codes, unique identifiers, or web links for publicly available datasets
- A description of any restrictions on data availability
- For clinical datasets or third party data, please ensure that the statement adheres to our policy

Links to raw sequencing data produced in this study are available from the HipSci project website (www.hipsci.org). Raw data are deposited in the European Nucleotide Archive (ENA) under the following accessions: ERP006946 (WES, open access samples) and ERP017015 (WGS, open access samples), and in the European Genotype-Phenotype Archive under accession EGAS00001000592 (WES, managed access samples). The open access data samples are freely available to download, while the managed access data are available following a request and a data access agreement. The variant call sets are deposited at Mendeley: https://data.mendeley.com/datasets/6rfc2xrnyd/1

# Field-specific reporting

Please select the one below that is the best fit for your research. If you are not sure, read the appropriate sections before making your selection.

☒ Life sciences ☐ Behavioural & social sciences ☐ Ecological, evolutionary & environmental sciences

For a reference copy of the document with all sections, see nature.com/documents/nr-reporting-summary-flat.pdf

# Life sciences study design

All studies must disclose on these points even when the disclosure is negative.

| | |
|---|---|
| Sample size | The sample size was 555 human iPS cells lines from multiple cohorts including HipSci, one of the largest iPS cell repositories in the world. This was a systematic study on the mutational landscape of hiPSC WGS. We included as many iPSCs as possible and therefore maximized the sample size as much as possible. |
| Data exclusions | No data or cell lines were excluded. |
| Replication | The data analysis was replicated independently by co-authors. Some patients had multiple iPS cells lines generated and sequenced. Some B-hiPSCs were single-cell sub-cloned (outlined in Table S14). The differentiation and immunostaining experiments were performed in triplicates. |
| Randomization | There was no randomization as it was an observational and descriptive study on human iPS cells. |
| Blinding | This was not a clinical trial. The investigators were not blinded to allocation during experiments and outcome assessment. |

# Reporting for specific materials, systems and methods

We require information from authors about some types of materials, experimental systems and methods used in many studies. Here, indicate whether each material, system or method listed is relevant to your study. If you are not sure if a list item applies to your research, read the appropriate section before selecting a response.

## Materials & experimental systems

| n/a | Involved in the study |
|---|---|
| ☒ | ☐ Antibodies |
| ☐ | ☒ Eukaryotic cell lines |
| ☒ | ☐ Palaeontology and archaeology |
| ☒ | ☐ Animals and other organisms |
| ☐ | ☒ Human research participants |
| ☒ | ☐ Clinical data |
| ☒ | ☐ Dual use research of concern |

## Methods

| n/a | Involved in the study |
|---|---|
| ☒ | ☐ ChIP-seq |
| ☒ | ☐ Flow cytometry |
| ☒ | ☐ MRI-based neuroimaging |

# Eukaryotic cell lines

Policy information about cell lines

| | |
|---|---|
| Cell line source(s) | The cell lines were from HipSci and the Insignia projects. Some hiPSCs from S2 and S7 have been previously published (https://doi.org/10.1371/journal.pgen.1004432) |
| Authentication | The cell lines have been authenticated by HipSci and Insignia . |
| Mycoplasma contamination | All cell lines tested negative for mycoplasma. |
| Commonly misidentified lines (See ICLAC register) | There were no commonly misidentified lines. |

# Human research participants

Policy information about studies involving human research participants

| | |
|---|---|
| Population characteristics | All participants were organ donors, volunteers via the NIHR Cambridge Bioresource or recruited for the Insignia project (doi: https://doi.org/10.1101/2020.08.04.234245) and appropriately consented. The exact age information of the human patient samples is not publicly available as the information could compromise privacy and lead to identification of individuals. The gender and genotype of the Insignia participants are as follows:<br><br>Samples Genotype Gender |

MSH65 AOA2 Male
MSH69 AOA2 Male
MSH70 AOA2 Male
MSH80 AT Female
MSH11 ATM Female
MSH5 Chemo Female
MSH88 Chemo Female
MSH77 CMMRD Male
MSH94 CMMRD Female
MSH25 Control Female
MSH39 Control Female
MSH53 Control Female
MSH59 Control Male
MSH63 Control Male
MSH98 Control Male
MSH20 XPC Male
MSH40 XPC Male
MSH93 XPD Male
MSH95 XPD Female
MSH97 XPD Male
MSH43 XPG Male
MSH71 AOA2 Female
MSH68 AOA2 Male
MSH13 ATM Female
MSH29 Control Female
MSH90 Myelofibrosis Male
MSH3 PMS2 Female
MSH41 XPC Male
MSH67 AOA2 Male
MSH72 AOA2 Male
MSH14 ATM Female
MSH28 ATM Female
MSH35 ATM Female
MSH44 ATM Male
MSH48 ATM Male
MSH46 BLM Male
MSH73 BRCA1 Male
MSH75 BRCA1 Male
MSH74 BRCA2 Male
MSH76 BRCA2 Male
MSH89 CMMRD Female
MSH54 Cockaynes Female
MSH55 Cockaynes Female
MSH22 Control Male
MSH23 Control Female
MSH26 Control Female
MSH31 Control Male
MSH33 Control Female
MSH36 Control Female
MSH45 Control Female
MSH52 Control Female
MSH58 Control Female
MSH61 Control Male
MSH62 Control Female
MSH64 Control Female
MSH87 Control Female
MSH51 FVS Female
MSH56 FVS Male
MSH79 FVS Female
MSH47 Lynch Female
MSH60 Lynch Male
MSH78 Lynch Female
MSH57 RTS Female
MSH6 SECISBP2 Male
MSH8 SECISBP2 Male
MSH2 XPA Male
MSH86 XPB Female
MSH21 XPC Male
MSH42 XPC Female
MSH30 XPD Male
MSH37 XPD Male
MSH91 XPE Female
MSH27 XPG Female
MSH32 XPG Female
MSH34 XPG Male
MSH92 XPV Male

MSH19 XPV Male
MSH85 XPV Female

Recruitment | Recruitment was voluntary for all research participants. We aimed to recruit as many patients with DNA repair deficiency as possible.

Ethics oversight | HipSci: REC 09/H0304/77, V2 04/01/2013, REC 09/H0304/77, V3 15/03/2013; Insignia: REC 13/EE/0302; organ donors: REC 09/H306/73)

Note that full information on the approval of the study protocol must also be provided in the manuscript.

