## [Peer Review File · Nature Genetics]

Peer Review Information

Manuscript Title: Substantial somatic genomic variation and selection for BCOR mutations in human induced pluripotent stem cells

Corresponding author name(s): Dr. Serena Nik-Zainal

Reviewer Comments & Decisions:

Decision Letter, initial version:
--

2nd Jul 2021

Dear Dr Nik-Zainal,

I'm sorry that it's taken so long to get this decision to you - thanks so much for bearing with me.

Your Article, "Substantial somatic genomic variation and selection for BCOR mutations in human induced pluripotent stem cells" has now been seen by 4 referees. Please note that Reviewers #1 and #2 reviewed the work together and have uploaded the same report. You will see from their comments copied below that while they find your work of considerable potential interest, they have raised quite substantial concerns that must be addressed. In light of these comments, we cannot accept the manuscript for publication, but would be very interested in considering a revised version that addresses these serious concerns.

We hope you will find the referees' comments useful as you decide how to proceed. If you wish to submit a substantially revised manuscript, please bear in mind that we will be reluctant to approach the referees again in the absence of major revisions.

To guide the scope of the revisions, the editors discuss the referee reports in detail within the team, including with the chief editor, with a view to identifying key priorities that should be addressed in revision and sometimes overruling referee requests that are deemed beyond the scope of the current study. In this case, we noted that the reviewers provided relatively concordant reports and we'd expect all their comments to be addressed in full, either through further analysis or experimentation (both preferable, where appropriate) or through textual edits. They consider the scale of the work to be impressive and they are enthusiastic about the value of the resource that you have generated. However, there is also agreement that the overall biological advance afforded by the work is modest.

Given the dataset you have at your disposal, they have provided suggestions as to how this aspect of the work might be fortified; for example, through further in silico analysis, and a deeper dive into the relevance of the BCOR mutations, which they all agree is an interesting and potentially important finding. You'll also see that Reviewer #3 has asked for variant call sets, summary files, and custom code to be made available, we also consider this to be a priority.

Please do not hesitate to get in touch if you would like to discuss these issues further.

If you choose to revise your manuscript taking into account all reviewer and editor comments, please highlight all changes in the manuscript text file. At this stage we will need you to upload a copy of the manuscript in MS Word .docx or similar editable format.

*2) If you have not done so already please begin to revise your manuscript so that it conforms to our Article format instructions, available [here](http://www.nature.com/ng/authors/article_types/index.html). Refer also to any guidelines provided in this letter.

[REDACTED]

If you wish to submit a suitably revised manuscript we would hope to receive it within 6 months. If you cannot send it within this time, please let us know. We will be happy to consider your revision so long as nothing similar has been accepted for publication at Nature Genetics or published elsewhere. Should your manuscript be substantially delayed without notifying us in advance and your article is

eventually published, the received date would be that of the revised, not the original, version.

Thank you for the opportunity to review your work.

Sincerely,

Safia Danovi
Editor
Nature Genetics

Referee expertise:

Referee #1: iPSCs (including translational/clinical applications, reviewed with Reviewer #2)

Referee #2: iPSCs (including translational/clinical applications, reviewed with Reviewer #1)

Referee #3: Mutational signatures (this reviewer has signed their report)

Referee #4: iPSCs

Reviewers' Comments:

Reviewer #1:

Remarks to the Author:

The authors analyze large cohorts of dermal fibroblast-derived "F-hiPSCs" as well as blood (erythroblast or EPC)-derived "B-hiPSCs" by WGS and WES to catalogue differences in mutational frequencies and types. They show that many F-hiPSC contain more mutations than B-hiPSC and in particular, more mutations that are consistent with UV exposure. They also show that F-hiPSCs made from the same individual can be highly heterogeneous due to the oligoclonal nature of the skin biopsy fibroblasts. B-hiPSCs, in contrast, show lower levels of mutations but show BCOR mutations at rates that are much higher than those of F-hiPSCs. The BCOR mutations, detected in multiple independent cohorts of hiPSCs, were not detected in somatic cells and therefore were likely selected for during reprogramming. Finally, the authors describe increased levels of oxidative stress mutational pattern

after long-term culture.

Several of the main findings are not particularly novel: prior studies have shown that skin biopsy fibroblasts are polyclonal and contain numerous mutations (many of which are consistent with UV exposure, e.g. Refs #8,32), resulting in genetically heterogeneous F-hiPSC clones [e.g., Refs #8,23]. Similarly, it has already been shown that blood-derived B-hiPSCs tend to contain few mutations compared to F-hiPSCs (e.g., PMID 23573220, 30840883) and hiPSC cultured under normoxic conditions accumulate mutations linked to oxidative stress (Ref #15), as is the case with many other long-term cultured cells. Nevertheless, the catalogued mutations and mutational patterns constitute a valuable resource, and the finding that BCOR mutations may be very common in otherwise mutation-sparse erythroblast-hiPSCs is, if validated, novel and potentially significant. Overall the study is quite descriptive and would benefit from some further experimental validation.

UV exposure as the main cause of the high frequency of apparently UV-related mutational signatures in fibroblasts and F-hiPSCs:

Many of the mutational signatures identified specifically in fibroblast and F-hiPSC are consistent with 'COSMIC Signature 7' (UV-damage). It is reasonable to assume that they are indeed at least partially if not predominantly caused by prior UV-exposure of the parental skin fibroblast. However, several other mutational signatures show similarities to Signature 7 but are linked to other processes such as DNA replication (10B) and repair (6,30) or aging and epigenetic remodeling (1,2). To what degree do these or other processes contribute to the occurrence of the observed mutational spectra? If a history of more intense UV exposure was indeed the predominant reason for the observed patterns then one would perhaps also predict a correlation with age, but the authors did not see that (Fig 2G). Analysis of hiPSCs derived from UV-unexposed fetal HDFs before and after controlled in vitro exposure to UV light might clarify this point. Furthermore, the amount of UV damage and accumulation of mutations in vivo depends on the location of the skin. Do the authors know which part of the skin was used in each donor? If they do and if the locations vary: does the burden of Signature 7-like mutations correlate with a more exposed location of the biopsied skin? Analysis of hiPSCs derived from exposed and unexposed fibroblast obtained from the same individual would be quite informative.

Frequent BCOR mutations in erythroblast hiPSCs:

Erythroblasts are among the most frequent donor cell types used for hiPSC production, and indeed peripheral blood is increasingly the predominant source of tissue for reprogramming given the relative ease of access over skin biopsy. The very high frequency of BCOR mutations in erythroblast-derived hiPSCs derived from multiple individuals and independent cohorts is therefore potentially quite alarming. However, the cohort of erythroblast donors and hiPSCs used in this study was very small compared to the size of the F-hiPSC cohort: BCOR mutations were found in 3 out of 17 erythroblast hiPSCs (HipSci cohort, 9 donors) and 12/21 erythroblast hiPSCs (Insignia patient cohort, 21 donors). It is not clear how the BCOR mutations might affect the function and properties of differentiated cells derived from mutant hiPSCs. A table showing all of the identified BCOR mutations should be included. BCOR is highly expressed by pluripotent stem cell derived neural precursors and linked to neuroepithelial cell tumors and AML. Do the observed BCOR mutations affect neuroectodermal lineage cells or myeloid cell differentiation and function? Can these phenotypes (if present) be rescued by targeted gene repair? Given that BCOR is X-linked, the gender and allele state of the affected lines should also be disclosed in the main text. Given that BCOR function intersects with apoptosis and TP53 pathways it would be interesting to know if episomal reprogramming of PBMCs (which often employs BCL-XL overexpression or TP53 pathway inhibition) results in lower BCOR mutation frequencies

compared to Sendai viral reprogramming. The hiPSC derivation, expansion, and subcloning methods should be described in much more detail: differences (e.g., in media composition or addition of small molecule inhibitors) could easily alter the selective pressure landscapes during somatic cell derivation/expansion, reprogramming, and hiPSC expansion.

Minor issues:

That claim that “hiPSCs reprogrammed from erythroblasts show ... BCOR mutations in ~57% of lines” appears to correct for the cohort showing 12/21 (57%) but the authors also observed BCOR mutations in 2/9 (22%) in the other cohort; i.e., in total, in 14/30 (47%) of such lines.

Suppl. Figure 4B, bottom: the legend is shown in reverse order.

The authors should separate the description and analysis of PBMC-erythroblast derived “B-hiPSCs” from that of PBMC-EPC derived “B-hiPSCs”: EPCs and erythroblasts are different cell types derived from different lineages and their expansion and reprogramming involves different medias and culture conditions that likely result in significantly different selection pressure landscapes.

Another note of interest is the frequent occurrence of clonal hematopoiesis, especially in older individuals. One DNMT3A mutation was found in the B-hiPSC, but it was not mentioned whether the nature of the mutation was similar to those found in individuals with clonal hematopoiesis.

The presence of mutations should be confirmed by PCR/Sanger or ddPCR.

Reviewer #2:

Remarks to the Author:

The authors analyze large cohorts of dermal fibroblast-derived “F-hiPSCs” as well as blood (erythroblast or EPC)-derived “B-hiPSCs” by WGS and WES to catalogue differences in mutational frequencies and types. They show that many F-hiPSC contain more mutations than B-hiPSC and in particular, more mutations that are consistent with UV exposure. They also show that F-hiPSCs made from the same individual can be highly heterogeneous due to the oligoclonal nature of the skin biopsy fibroblasts. B-hiPSCs, in contrast, show lower levels of mutations but show BCOR mutations at rates that are much higher than those of F-hiPSCs. The BCOR mutations, detected in multiple independent cohorts of hiPSCs, were not detected in somatic cells and therefore were likely selected for during reprogramming. Finally, the authors describe increased levels of oxidative stress mutational pattern after long-term culture.

Several of the main findings are not particularly novel: prior studies have shown that skin biopsy fibroblasts are polyclonal and contain numerous mutations (many of which are consistent with UV exposure, e.g. Refs #8,32), resulting in genetically heterogeneous F-hiPSC clones [e.g., Refs #8,23]. Similarly, it has already been shown that blood-derived B-hiPSCs tend to contain few mutations compared to F-hiPSCs (e.g., PMID 23573220, 30840883) and hiPSC cultured under normoxic conditions accumulate mutations linked to oxidative stress (Ref #15), as is the case with many other long-term cultured cells. Nevertheless, the catalogued mutations and mutational patterns constitute a valuable resource, and the finding that BCOR mutations may be very common in otherwise mutation-

sparse erythroblast-hiPSCs is, if validated, novel and potentially significant. Overall the study is quite descriptive and would benefit from some further experimental validation.

UV exposure as the main cause of the high frequency of apparently UV-related mutational signatures in fibroblasts and F-hiPSCs:

Many of the mutational signatures identified specifically in fibroblast and F-hiPSC are consistent with 'COSMIC Signature 7' (UV-damage). It is reasonable to assume that they are indeed at least partially if not predominantly caused by prior UV-exposure of the parental skin fibroblast. However, several other mutational signatures show similarities to Signature 7 but are linked to other processes such as DNA replication (10B) and repair (6,30) or aging and epigenetic remodeling (1,2). To what degree do these or other processes contribute to the occurrence of the observed mutational spectra? If a history of more intense UV exposure was indeed the predominant reason for the observed patterns then one would perhaps also predict a correlation with age, but the authors did not see that (Fig 2G). Analysis of hiPSCs derived from UV-unexposed fetal HDFs before and after controlled in vitro exposure to UV light might clarify this point. Furthermore, the amount of UV damage and accumulation of mutations in vivo depends on the location of the skin. Do the authors know which part of the skin was used in each donor? If they do and if the locations vary: does the burden of Signature 7-like mutations correlate with a more exposed location of the biopsied skin? Analysis of hiPSCs derived from exposed and unexposed fibroblast obtained from the same individual would be quite informative.

Frequent BCOR mutations in erythroblast hiPSCs:

Erythroblasts are among the most frequent donor cell types used for hiPSC production, and indeed peripheral blood is increasingly the predominant source of tissue for reprogramming given the relative ease of access over skin biopsy. The very high frequency of BCOR mutations in erythroblast-derived hiPSCs derived from multiple individuals and independent cohorts is therefore potentially quite alarming. However, the cohort of erythroblast donors and hiPSCs used in this study was very small compared to the size of the F-hiPSC cohort: BCOR mutations were found in 3 out of 17 erythroblast hiPSCs (HipSci cohort, 9 donors) and 12/21 erythroblast hiPSCs (Insignia patient cohort, 21 donors). It is not clear how the BCOR mutations might affect the function and properties of differentiated cells derived from mutant hiPSCs. A table showing all of the identified BCOR mutations should be included. BCOR is highly expressed by pluripotent stem cell derived neural precursors and linked to neuroepithelial cell tumors and AML. Do the observed BCOR mutations affect neuroectodermal lineage cells or myeloid cell differentiation and function? Can these phenotypes (if present) be rescued by targeted gene repair? Given that BCOR is X-linked, the gender and allele state of the affected lines should also be disclosed in the main text. Given that BCOR function intersects with apoptosis and TP53 pathways it would be interesting to know if episomal reprogramming of PBMCs (which often employs BCL-XL overexpression or TP53 pathway inhibition) results in lower BCOR mutation frequencies compared to Sendai viral reprogramming. The hiPSC derivation, expansion, and subcloning methods should be described in much more detail: differences (e.g., in media composition or addition of small molecule inhibitors) could easily alter the selective pressure landscapes during somatic cell derivation/expansion, reprogramming, and hiPSC expansion.

Minor issues:

That claim that "hiPSCs reprogrammed from erythroblasts show ... BCOR mutations in ~57% of lines" appears to correct for the cohort showing 12/21 (57%) but the authors also observed BCOR mutations in 2/9 (22%) in the other cohort; i.e., in total, in 14/30 (47%) of such lines.

Suppl. Figure 4B, bottom: the legend is shown in reverse order.

The authors should separate the description and analysis of PBMC-erythroblast derived "B-hiPSCs" from that of PBMC-EPC derived 'B-hiPSCs': EPCs and erythroblasts are different cell types derived from different lineages and their expansion and reprogramming involves different medias and culture conditions that likely result in significantly different selection pressure landscapes.

Another note of interest is the frequent occurrence of clonal hematopoiesis, especially in older individuals. One DNMT3A mutation was found in the B-hiPSC, but it was not mentioned whether the nature of the mutation was similar to those found in individuals with clonal hematopoiesis.

The presence of mutations should be confirmed by PCR/Sanger or ddPCR.

Reviewer #3:

Remarks to the Author:

Main Findings

The manuscript by Rouhani and colleagues reports substantial mutagenesis in blood and skin derived human induced pluripotent stem cells (hiPSCs) from one individual, especially in hiPSCs derived from skin fibroblasts, which have many ultraviolet-radiation (UV) induced mutations.

The manuscript follows up on this finding with an analysis of whole genome sequence from 324 fibroblast derived hiPSCs from the Human Induced Pluripotent Stem Cell Initiative, "HipSci", and finds UV mutations in 72%, with mutation burdens ranging from 0.25 to 15 single base mutations per megabase.

After adding some additional exome sequence data from HipSci fibroblast derived hiPSCs, the manuscript reports 272 predicted pathogenic mutations in 77 out of 452 hiPSC lines.

The manuscript also analysed mutations in 44 erythroblast-derived hiPSCs (of which 17 were HipSci), which show much less mutagenesis: 0.28 to 1.4 single base mutations per megabase. However, over half of these hiPSCs have mutations in the BCOR (BCL6 corepressor) gene, which the manuscript proposes is due to selection pressure.

The manuscript also notes that the examined hiPSC lines were karyotypically stable, indicating that use of hiPSCs for cell-based therapy would require sequencing in addition to karyotypic screening.

I do not have extensive background in the hiPSC area, but a quick literature search indicates that these sorts of high mutational burdens have not been reported in hiPSCs. In the context of literature on mutagenesis in normal skin fibroblasts, the results regarding the fibroblast derived hiPSCs are not surprising but well worth reporting. Below, I have some further comments on missed opportunities regarding this theme.

Major issues:

1. It was not possible to fully assess this manuscript given that variant call sets, summary files, and

code (bespoke software) were not available for review (page 26 of the manuscript). These should be provided to allow for a full review.

2. Results and interpretation regarding UV mutations in fibroblast derived hiPSCs. There are two substantial publications regarding UV mutagenesis in normal skin; one is reference 41 in the current manuscript and the other is Saini et al., <https://doi.org/10.1371/journal.pgen.1006385>, which should be referenced. One would assume that there is minimal UV exposure during cell culture, and that all UV mutagenesis observed in the hiPSC lines was inherited from the founding fibroblasts. Therefore, the analysis on page 8 showing positive correlation between mutation burdens in founding fibroblasts and derived hiPSC lines seems confirmatory and could be in supplementary information. There is a surprisingly weak statement on page 9: "It is therefore possible that some driver mutations identified in hiPSCs were acquired in vivo and not during cell culture". The mutational spectra are completely dominated by UV (Figure 2C). Does the manuscript really propose that UV mutations were acquired during cell culture? The manuscript should computationally attribute (assign) the driver mutations to likely causal mutational processes and report how many are likely due to UV and therefore likely acquired prior to cell culture. The manuscript misses the opportunity to compare mutation burdens in the founder fibroblast lines and in the derived hiPSCs to those reported in reference 41 and in Saini et al. The manuscript also misses the opportunity to compare the driver mutation profile in fibroblast derived hiPSCs to that reported in reference 41. The Discussion, page 15, states "Whilst 17% of lines have variants in cancer genes that are recurrently mutated in hiPSCs, nearly half of these variants are also present in the starting fibroblast material and represent clonal mutations acquired in vivo.". Does this seem to imply that half were acquired during hiPSC generation and culture as opposed to the UV mutations pre-existing but not being detected? The same issue arises in the sentence on page 10 "[i]n summary, nearly half of cancer-related variants in hiPSC lines that were seen recurrently, had been acquired in vivo." The question of how UV mutations could be acquired during cell culture is not touched on, so these sentences, absent further interpretation, seem fundamentally misleading. In fact, the null hypothesis would be that almost all cancer gene mutations were present in the founding fibroblast population because the overwhelming proportion of mutations were UV mutations (Figure 2C), but these mutations were not detected in the cultured fibroblasts because they were not present in a large-enough proportion of the cells to be detected by sequencing or were present when the hiPSCs were derived but were subsequently lost during culture of the founding fibroblasts.

3. The BCOR findings are intriguing but there seem to be many loose ends. First, 3 / 17 BCOR mutated hiPSC lines is not very impressive after considering the implicit multiple hypothesis testing across many candidate driver genes. That statistics regarding which driver mutations occur in which lines and which lines are from the same founding population are not clear in the top partial paragraph on page 10 or in the second paragraph on that page. We need a table and more details on how the data were used in the statistical analysis. The follow on analysis of the Insignia project data also does not provide enough information. Many of the hiPSC lines were from patients with DNA repair defects, and some may have had high somatic mutation burdens in the founding populations. Of the 9 donors with BCOR mutations, how many were in patients with DNA repair defects and high mutation burdens? I could not locate Tables S7 and S8. Overall, I think the BCOR finding may well be correct, but the supporting data and analysis needs to be provided in detail. I also think that the discussion on pages 12 and 13 should be clarified. It seems that the key question is whether there is selection for the BCOR mutations, for which there seems reasonable evidence. The question of whether there is selection is somewhat independent of whether the mutations were inherited from the founding erythrocyte population or arose during generation and culture of the hiPSCs. It seems that the concept of clonal haematopoiesis here is not that helpful, the real subsidiary question is whether the BCOR

mutations were present in the founding population of erythroblasts, regardless of whether the proportion of founding cells with the mutation reaches some level to be termed clonal haematopoiesis.

Other comments:

On page 7, the manuscript draws attention to the interesting finding of high diversity in mutation profile and burden among fibroblast derived hiPSC lines from the same "reprogramming experiment", including 2 lines from donor HPSI03141-bubh. I assume the authors checked germline variants to ensure there were no sample mix ups and that both derived lines were indeed from the same donor, but this should be stated.

On page 8, the manuscript notes "it is essential to compare the F-hiPSC genome to a matched germline sample". Reading methods, pages 21 through 23, it seems this was not done; the relationship between the methods on pages 21 and 22 to those on page 23 is unclear, and it is not clear which methods were used at which places in the Results section.

Minor comments / typos:

Page 4 "pre-screened and believed to be" -> "pre-screened and are believed to be"

Page 6, "p value < 0.0001": what was the comparison and test?

Page 8, the use of the term "oligoclonal" seems a bit odd in some places, for example, rather than writing "F-hiPSCs derived from the same oligoclonal population will be more similar to each other", why not write "F-hiPSCs derived from the same subclone will be more similar to each other"? Is there a semantic difference that I am missing?

Page 9, references 41,42 – there are several other papers reporting "cancer-associated mutations in normal cells", which probably also should be cited here.

Figure S7 legend, "and mostly composed" -> "and are mostly composed"

File 60902_0_data_set_662547_qsmx2f.xlsx, tab "germline" does not seem to have a sample id.

Signed:

Steven G. Rozen

Reviewer #4:

Remarks to the Author:

Rouhani et al. conduct a large-scale investigation of the mutational burden of iPSCs. They observe that skin fibroblast-derived samples retain many UV-related mutations, as was previously described on a smaller number of samples. The authors show that the heterogeneity in the fibroblast donor cells within a single sample may result in different mutational backgrounds for iPSC clones from an individual. They also observe that multiple cancer genes are recurrently mutated, including BCOR. Overall, there is limited novel findings; however, given the large number of iPSC samples with WGS

data this study could have the potential to discover novel and interesting findings.

The investigators initially conduct a comparison between the mutational burden and profiles between 12 skin fibroblast-derived and 6 endothelial-derived iPSCs. They find that the fibroblast derived iPSCs have a greater overall mutation largely driven by the presence of a UV damage signature. The fact that fibroblast derived iPSCs carry UV damage mutations has previously been described.

The investigators next investigate more than 452 fibroblast-derived iPSCs confirming that most carry detectable UV damage. They also show that there is no association between the number of mutations and age (which has been shown using fewer iPSCs) or gender of the donor.

Next, the authors show that genomic heterogeneity across iPSCs are largely derived from the clonal heterogeneity of the fibroblast donor cells. It has already been shown that heterogeneity exists across fibroblast cells within the same sample and that reprogramming occurs from a single cell. And hence the concepts underlying the schematic in Figure 3D are not surprising or novel.

The main advantage of this study is the number of samples (>450, compared to previous studies, which analyzed <20 samples each). The large number of samples is comparable with recent cancer genomics studies and should be exploited to investigate the presence of mutational hotspots, i.e. genomic locations (or genes) that are more likely to accumulate mutations during reprogramming or that may confer some selective advantage if they harbor mutations in the original fibroblasts. While the observation that many iPSC samples carry mutations in cancer genes is useful from a regenerative medicine standpoint, understanding if certain loci undergo selection may impact how iPSCs and iPSC-derived tissues are used as model systems to study human biology. A genome-wide analysis to identify mutational hotspots could greatly strengthen the novelty of this study, which in general confirms several previous studies that were performed on a smaller scale.

Identification of recurrent BCOR mutations is an interesting observation. It is important that the authors show that the mutational burden in this gene is significantly higher than expected by chance.

The observation that the oxidative stress mutational signature increases over culture time is an interesting observation. It would be interesting to explore when these mutations arise. Are they present in a minor fraction of the fibroblast population which have a selective advantage during reprogramming, do they occur during reprogramming or during passaging? In the iPSCs are these mutations continually arising or do subclones that carry the mutations have a selective advantage?

The major conceptual issue with this study stems from the fact that the authors identified mutations by comparing iPSCs to their matched donor fibroblasts (using Fisher's exact test). And hence in vivo mutations with a high VAF in the fibroblasts will not be identified as mutations in the iPSCs. The authors are therefore restricted to identifying mutations that are subclonal in the fibroblast sample or that occur during reprogramming or subsequent culturing. Additionally, it is impossible to distinguish between mutations that were present in a minor fibroblast population from which the donor cell was derived (and are therefore not observed in the fibroblast WGS data) and those that occurred during reprogramming or shortly thereafter (and hence have a VAF $\sim .5$ in iPSCs). The authors need to address this point.

Minor comments

While the authors acknowledge previous work addressing somatic mutations in iPSCs (Introduction: “systematic large-scale, whole genome assessments of mutagenesis at single-nucleotide resolution have been limited”), they do not discuss the findings of these studies, but focus on recent studies on embryonic stem cells.

While this study and others show that there is no correlation between mutation burden of F-hiPSC and donor age or gender, as a previous study (ref. 24) is in contrast with this statement it would be good to discuss the discrepancy.

Figure 3B shows the “number of mutations in fibroblasts”, but it is unclear from the Methods how this number was calculated.

In the abstract “In contrast, B-hiPSCs reprogrammed from erythroblasts show lower levels of genome-wide mutations” is not supported by any statistical analysis. The primary analysis is on page 5 – starting on line 4.

Author Rebuttal to Initial comments

Response to Reviewers

We would like to thank the reviewers for the time and effort in appraising our manuscript. We are grateful that the reviewers can see the value in the analysis. Based on the questions and with editorial guidance, we identified two major areas that required work that has resulted in substantial revision of the manuscript. The two areas are i) an agnostic dN/dS (selection) analysis on all acquired mutations to provide a statistically robust framework for presenting putatively selected for variants, and ii) Experimental work to explore the functional impact of *BCOR* mutations on hiPSCs. A detailed point-by-point response to individual comments of the reviewers is also provided below. First, we provide an overarching report of these two areas of work which has caused considerable changes to the manuscript. We hope that the reviewers agree that we have enhanced our understanding and improved the manuscript.

- i) **Statistical work to provide backing for reporting putatively selected variants.** This addresses points 3.3, 4.1

Aim: To provide a robust, agnostic, statistical framework of assessing selection on coding variants of all genes in hiPSCs.

Methods: Evaluate selection across all genes by calculating the dN/dS ratio, which is the ratio between the rate of non-synonymous substitutions per non-synonymous site and the rate of synonymous substitutions per synonymous site using dNdScv¹.

Three analyses were performed on coding variants derived from 452 HipSci F-hiPSCs and 78 Insignia B-hiPSCs:

- (1) Default dNdScv: an agnostic search looking for selection across *all* genes (or the exome).
- (2) Restricted hypothesis testing: To increase statistical power to detect selection, we calculated restricted q-values for *a priori* known cancer genes, (list obtained from Martincorena et al 2017¹).
- (3) Detection of mutational hotspots (using sitednds from the dNdScv package)¹.

Results from F-hiPSCs:

- Exome-wide dN/dS ratio is ~1
- dNdScv analysis of all genes in the genome found only *BCOR* as being under significant positive selection (qval=3.64e-08).
- Restricted hypothesis testing on known cancer genes also found only *BCOR* as significant. A few known cancer genes appear towards the top of the list (although without reaching significance (qvals~0.25)).
- Mutational hotspots searches revealed 4 in *TP53* and 1 in *APC*, but all were non-significant.

Results from B-hiPSCs:

- Exome-wide dN/dS ratios ~1
- dNdScv analysis of all genes in the genome found only *BCOR* as being under significant positive selection (qval=0).
- Restricted hypothesis testing also only revealed *BCOR* as significant (qval=0).
- No mutational hotspots were found.

Conclusion:

In this revision, through both a systematic statistical search across all genes and a more directed approach restricted to cancer genes, we find statistically significant evidence of selection for *BCOR*, in both B-hiPSCs and F-hiPSCs, and in no other genes in the coding sequence.

- ii) Experimental work and functional analyses to understand the impact of BCOR in hiPSCs. This addresses points 1.4 and 1.5.

Aim: To understand whether *BCOR* mutations produce any functional consequence in affected B-hiPSCs.

Method 1: We performed RNA-sequencing on the B-hiPSCs from 78 Insignia donors comprising healthy controls and patients.

Result 1: Global transcriptomic analysis revealed two principal components (PC) driving most of the variance in the dataset. The first PC distinguished two groups (Fig. A below and the new Fig. 5a in the manuscript), with almost all the *BCOR*-mutated B-hiPSCs (orange and red dots) restricted to one group. The second PC distinguished the donors by gender. Seven B-hiPSCs (highlighted in grey, Fig. A) did not show any *BCOR* variants in the parental population. However, following a series of propagation experiments, *BCOR* mutations were found in some (not all) daughter subclones of these seven B-hiPSCs, suggesting that these were subclonal and/or had arisen during propagation. The rest of the 50 B-hiPSCs (highlighted in blue, Figure A) had no *BCOR* mutations in either parental clone or daughter subclones. **This suggests that *BCOR* mutations are associated with significant global transcriptional changes in B-hiPSCs.**

Figure A: Principal component analysis of transcriptomic data on 78 B-hiPSCs derived from Insignia donors. The first principal component distinguishes cases with *BCOR* mutations (orange and red; dots and triangles have high variant allele fractions of *BCOR* mutations) and those with no *BCOR* mutations (blue and grey dots and triangles). The second principal component distinguishes by gender. Grey dots/triangles represent B-hiPSC lines which do not demonstrate *BCOR* mutations in parental B-hiPSCs but which subsequently were found to have *BCOR* mutations in daughter sub-clones.

Method 2: To understand whether there was an impact on differentiation potential, we performed directed differentiation towards a neuronal lineage, contrasting a *BCOR*-mutated B-hiPSC, MSH40i2 (*BCOR*-mut), and a wild-type B-hiPSC, MSH34i2 (*BCOR*-wt). Three independent replicates were used for MSH40i2 and MSH34i2 (six lines altogether). To capture dynamic shifts through the directed differentiation process, we took samples in a time course, characterizing these lines morphologically and transcriptionally, on day 0 (D0, as hiPSC), day 6 (D6) and day 12

(D12), representative of early and late neural stem cell (NSC) induction stages, and on day 27 (D27) as neurons.

Result 2: There were no overt morphological differences between *BCOR*-mut and *BCOR*-wt lines at the hiPSC stage (Day 0, Fig. B below, new Fig. 5d in manuscript, Fig. S10 and Fig. S11a in supplementary figures), with near identical Brightfield images and immunofluorescence (IF) characterization of pluripotent markers SSEA4 (green) and OCT4 (red). However, upon NSC induction (D12), *BCOR*-mut replicates showed inefficiency in differentiation confirmed by patchy expression of NSC marker PAX6 (Day 12, Fig. B, new Fig. 5d and Fig. S11b in manuscript). At D27, neuronal generation in *BCOR*-mut lines was markedly affected as seen as a reduction in TUBB3 (green) positive neurons (Day 27, Fig. B, new Fig. 5d and Fig. S11c in manuscript).

Figure B: Brightfield images and IF characterisation of hiPSCs (Day 0), neural stem cells (Day 12) and neurons (Day 27) of *BCOR* mutant (top panel, *BCOR* MT, MSH40i2 highlighted in orange) and *BCOR* intact (lower panel, *BCOR* WT, MSH34i2 highlighted in blue) B-hiPSCs.

In keeping with the morphological characterization, global transcriptomics of each independent line at each stage of differentiation (D0, D6, D12 and D27) revealed that *BCOR*-mut replicates exhibited a modest reduction in *BCOR* expression but elevated levels of *NANOG*, *KLF4* and *NODAL* compared to *BCOR*-wt replicates (Fig. C), similar to reports in other pluripotent models with *BCOR*-PRC1.1 defects². As shown in Fig. C, *UTF1* (implicated in maintenance of pluripotency through chromatin regulation), *VENTX*, *IRX4*, *PITX2* and *MIXL1* (all homeobox-related proteins with various roles in embryonic patterning) and *FOXC1* (important in the development of organs derived from the mesodermal-lineage) were strongly upregulated in *BCOR*-mut replicates while *NPTX1* (member of the neuronal pentraxin family) was strongly downregulated. At D6 and D12,

transcriptomic dynamics evolved to show that mesodermal markers such as *PAX7*, *TBX1* and *PAX3* were substantially elevated in the *BCOR*-mut replicates compared to *BCOR*-wt replicates, implicating a drive towards mesodermal lineages in *BCOR* compromised lines. On D27, neuronal markers such as *MAP2*, *DCX* and *FOXP1* were well-established in *BCOR*-wt but not in *BCOR*-mut cells underscoring the failure of neuronal differentiation in the latter.

Figure C: Heatmap of relative expression levels of members of non-canonical polycomb repressive complex 1.1 (PRC1.1), pluripotency markers of hiPSCs and markers of the three germ layers for *BCOR*-mut MSH40i2 (orange) and *BCOR*-wt MSH34i2 (blue) B-hiPSCs. *UTF1*, *VENTX* and the early mesodermal commitment marker *NODAL* are upregulated in *BCOR* mutants while the early neural commitment marker *NPTX1* is significantly downregulated. As the hiPSC lines are differentiated into the neural lineage, markers of the mesodermal lineage such as *MSX1*, *PAX3*, *PAX7*, *TBXT* and *TBX15* are upregulated in *BCOR* mutants and neural markers *SOX1*, *FOXP1*, *TBR1*, *TUBB3* are downregulated, compared to *BCOR* intact cells.

Conclusion:

In this substantial revision, we have conducted additional experiments and comprehensive analyses to demonstrate that:

- *BCOR* variants, that have been statistically shown to be selected for, produce defining global transcriptomic changes on affected B-hiPSCs.
- B-hiPSC lines that acquire *BCOR* variants do not show marked phenotypic differences at the iPSC stage when compared to *BCOR* wild-type lines.
- However, time-course directed differentiation experiments reveal that *BCOR*-mutated lines have compromised differentiation potential; they are less efficient at differentiation towards neuronal lineages and may be prone to differentiation into alternative lineages such as the mesoderm. This is demonstrated both morphologically and through transcriptomics.
- These results have far reaching consequences for researchers using hiPSCs for disease modelling or potentially for clinical therapies, particularly at the prevalence seen across a range of hiPSCs.

Reviewer #1 and #2:

Remarks to the Author:

The authors analyze large cohorts of dermal fibroblast-derived “F-hiPSCs” as well as blood (erythroblast or EPC)-derived “B-hiPSCs” by WGS and WES to catalogue differences in mutational frequencies and types. They show that many F-hiPSC contain more mutations than B-hiPSC and in particular, more mutations that are consistent with UV exposure. They also show that F-hiPSCs made from the same individual can be highly heterogeneous due to the oligoclonal nature of the skin biopsy fibroblasts. B-hiPSCs, in contrast, show lower levels of mutations but show *BCOR* mutations at rates that are much higher than those of F-hiPSCs. The *BCOR* mutations, detected in multiple independent cohorts of hiPSCs, were not detected in somatic cells and therefore were likely selected for during reprogramming. Finally, the authors describe increased levels of oxidative stress mutational pattern after long-term culture.

Point 1.1

Several of the main findings are not particularly novel: prior studies have shown that skin biopsy fibroblasts are polyclonal and contain numerous mutations (many of which are consistent with

UV exposure, e.g. Refs #8,32), resulting in genetically heterogeneous F-hiPSC clones [e.g., Refs #8,23). Similarly, it has already been shown that blood-derived B-hiPSCs tend to contain few mutations compared to F-hiPSCs (e.g., PMID 23573220, 30840883) and hiPSC cultured under normoxic conditions accumulate mutations linked to oxidative stress (Ref #15), as is the case with many other long-term cultured cells. Nevertheless, the catalogued mutations and mutational patterns constitute a valuable resource, and the finding that BCOR mutations may be very common in otherwise mutation-sparse erythroblast-hiPSCs is, if validated, novel and potentially significant. Overall the study is quite descriptive and would benefit from some further experimental validation.

We agree that some of our research findings provide support for prior smaller studies, which the reviewer highlights. However, while there are similarities to those previous efforts, there are some important differences. This work builds on those excellent studies, it addresses a different type of “heterogeneity” *and* explains why those anecdotal observations were observed in a substantial cohort comprehensively. To dissect it down:

- Ref #8 (PMID:30044985) and Ref #32 (PMID:32203388) indeed showed UV damage in 18 lines, and in limited numbers of individuals but using single cell analyses, respectively. Thus, the magnitude of UV mutagenesis in a large population of hiPSCs *and* the extent of variation between donors was not communicated. Also, Ref #8 did not show heterogeneity *between hiPSCs derived from the same person*. They emphasised the occurrence of subclonal heterogeneity *within* hiPSCs, an interesting but different type of heterogeneity.
- Ref #23 (PMID:23160490) on the other hand, suggested heterogeneity in starting tissue (fibroblasts) by means of copy number (CN) profiling. This was an interesting paper that highlighted how some CN variants observed in iPSCs were likely present in starting fibroblasts but did not perform this assessment at scale. In our paper, although we do argue that substantial mutagenesis in iPSCs has been inherited from starting fibroblasts, we demonstrate this in a substantially larger cohort of 118 pairs of iPSCs. Additionally, we do this for *substitutions/indels, i.e. small nucleotide variation*, something that cannot be “seen” at chromosomal scale and would have passed the customary CN-based QC. Moreover, we use agnostic statistics on mutational profiles and mutational burdens (Fig. 3), something that has never been systematically demonstrated previously, to display the similarities or differences between these 118 pairs, in an unbiased way.
- PMID 30840883 was also an interesting study, however, their comparisons were different to our experiments:
 - They compared “sib-pairs” of peripheral blood vs fibroblasts and peripheral blood vs bone marrow directly in Figure 4 (not iPSCs). They also only sequenced ~10% of

- the genomes, hence requiring PCR-based validation of their calls given their approach. Their approach was thus hugely different to our experiments.
- Subsequently (relevant to Figure 5), they compared *the acquisition of mutations following reprogramming* and argued that there were more mutations in F-hiPSCs when compared to their own fibroblasts, while there were not more mutations in B-hiPSCs compared to peripheral blood. Indeed, the difference in total number of SNVs reported between F-hiPSC and B-hiPSC was minimal (their Figure 5B), they did not report UV signatures at all and argued “The lack of detection of a UV signature in the skin fibroblasts that we analyzed likely reflects the fact that we collected them from the lower back, a body site that is not generally exposed to the sun”.
 - PMID 23573220 used cord-blood derived iPSCs and performed exome, rather than genome, sequencing on 5 clones, reporting under 15 SNVs per sample (i.e. very low numbers) with no direct comparisons. Importantly, this work has limited clinical and research relevance since it focuses exclusively on cord blood whereas our manuscript instead uses B-hiPSCs derived from a large number of adults.

We therefore feel that our work has novelty which builds on these previous publications and characterises hiPSCs to a hitherto unprecedented level.

We thank the reviewer for highlighting our work as an important resource. Furthermore, we hope that the additional work performed both analytically (dN/dS statistical analysis to demonstrate positive selection of *BCOR* mutations) and experimentally (systematic transcriptomics and morphological assessments over the time-course of directed differentiation), to highlight the relevance of *BCOR* mutations, will address their remaining concerns and lead to a recommendation for publication.

Point 1.2

UV exposure as the main cause of the high frequency of apparently UV-related mutational signatures in fibroblasts and F-hiPSCs:

Many of the mutational signatures identified specifically in fibroblast and F-hiPSC are consistent with ‘COSMIC Signature 7’ (UV-damage). It is reasonable to assume that they are indeed at least partially if not predominantly caused by prior UV-exposure of the parental skin fibroblast. However, several other mutational signatures show similarities to Signature 7 but are linked to other processes such as DNA replication (10B) and repair (6,30) or aging and epigenetic

remodelling (1,2). To what degree do these or other processes contribute to the occurrence of the observed mutational spectra?

On the face of it, the Signatures highlighted by the Reviewer are indeed similar in that they share a C>T emphasis. However, more detailed analysis of the mutational signatures associated with UV-light and other signatures reveal that they are significantly different. Specifically, it is accepted that other Signatures with some C>T mutations occur at different trinucleotide contexts and have other clear causes. We have highlighted this below in Table 1 and showed the different signatures for comparison (from COSMIC).

Signature	Aetiology and characteristic of signature	Important notes of validation and/or features that distinguish the signature from UV
	UV-light associated. C>T occurs at $\underline{C}C_N$ and \underline{TC}_N	Proven in human cells (naïve human cells treated with simulated sunlight 1.25 J recapitulates this signature). Kucab et al Cell 2019
	Deamination of methyl-C. Associated with age of diagnosis. C>T occurs at \underline{NCG}	Well documented by many others over the last 3 decades.
	Deamination of cytosine to uracil by APOBECs C>T at \underline{TC}_N	Documented extensively to be caused by APOBECs in yeast and humans
	Not just C>T but C>A and T>G mutations. Recent split into 10a-10d but has yet to be validated.	Caused by mutations in polymerase genes. 10a is still believed to be due to POLE. 10d believed to be due to POLD. 10b has not been validated.
	C>T at \underline{AC}_N and \underline{CC}_N	Has a hypermutator phenotype and an

	Due to NTHL1 loss which is a Base Excision Repair glycosylase.	associated mutation. The hiPSCs do not carry NTHL1 mutations
	C>T at GCN and C>A at CCA.	This is a mismatch repair signature with a massive hypermutator phenotype and indel pattern

Table 1. Mutational signatures that have dominant C>T mutations.

The mutational signature *assignment* step is the step that assigns the presence of any signature to individual samples (independently). All signatures that have ever been reported in skin³ including deamination is permitted to be assigned. The fitting algorithm produces the following outcome (Fig. D below, as Fig. 2c in the manuscript). The vast majority of signatures are either assigned to Sig 7 (pink) and some to Sig 18 (black). Other signatures including deamination are indeed present, as suggested by the Reviewer, but it is possible to see that they make up a much smaller proportion per sample, as shown in yellow.

Figure D. Distribution of mutational signatures in 324 F-hiPSC lines. The inset figure shows the relative exposures of mutational signature types.

Nevertheless, to ensure a rigorous analysis and to further reinforce that the signatures assigned are truly due to UV, we did seek out other corroborating information:

- CC>TT double subs mutations are strongly observed in F-hiPSCs (Figure 2d in the manuscript)⁴
- transcriptional strand bias which is also a strong indicator of the activity of transcription coupled nucleotide excision repair on UV damage (Figure 2e and 2f in the manuscript)⁵

Point 1.3

If a history of more intense UV exposure was indeed the predominant reason for the observed patterns then one would perhaps also predict a correlation with age, but the authors did not see that (Fig 2G). Analysis of hiPSCs derived from UV-unexposed fetal HDFs before and after controlled in vitro exposure to UV light might clarify this point. Furthermore, the amount of UV damage and accumulation of mutations in vivo depends on the location of the skin. Do the authors know which part of the skin was used in each donor? If they do and if the locations vary: does the burden of Signature 7-like mutations correlate with a more exposed location of the biopsied skin? Analysis of hiPSCs derived from exposed and unexposed fibroblast obtained from the same individual would be quite informative.

Whilst we agree that if the degree of UV exposure throughout different individuals' lives were relatively constant, then there ought to be a correlation with age. However, in reality there exists a huge variation in the amount of UV exposure any given person receives, depending on factors such as occupation, travel, behaviours in protecting themselves against UV radiation and frequency of sunburn. Indeed, it has been shown that there is no correlation between UV radiation exposure and age or sex⁶. Therefore, our results showing the lack of correlation of the hallmarks of UV damage with age are consistent with previous studies.

Instead, and in agreement with the reviewer, it is also our view that the location of the fibroblasts within the skin (i.e., superficial vs deep) is likely to be an important determinant on how much UV damage is seen. We also agree that an interesting and informative extension to our work would be to derive hiPSCs from exposed and unexposed skin as well as from deep and superficial fibroblasts. Unfortunately, this is beyond the scope of our manuscript, would be costly and labour intensive, but would serve as an interesting follow up study.

The information that we obtained from HipSci regarding the skin biopsy was that each was taken from approximately the same site for each individual and indeed was taken in a sun exposed area of the upper arm in each case.

Point 1.4

Frequent BCOR mutations in erythroblast hiPSCs:

Erythroblasts are among the most frequent donor cell types used for hiPSC production, and indeed peripheral blood is increasingly the predominant source of tissue for reprogramming given the relative ease of access over skin biopsy. The very high frequency of BCOR mutations in erythroblast-derived hiPSCs derived from multiple individuals and independent cohorts is

therefore potentially quite alarming. However, the cohort of erythroblast donors and hiPSCs used in this study was very small compared to the size of the F-hiPSC cohort: BCOR mutations were found in 3 out of 17 erythroblast hiPSCs (HipSci cohort, 9 donors) and 12/21 erythroblast hiPSCs (Insignia patient cohort, 21 donors). It is not clear how the BCOR mutations might affect the function and properties of differentiated cells derived from mutant hiPSCs. A table showing all of the identified BCOR mutations should be included.

We previously provided Supplementary Table 5 and 7 to show all the identified *BCOR* mutations in HipSci and Insignia data sets, respectively, as requested. We sincerely apologise if this was not available to the reviewer. In this revision, we have increased the number of Insignia cases from 21 donors to 78 donors. The list of *BCOR* mutations of HipSci F-hiPSCs and Insignia B-hiPSCs are now provided in new Supplementary Table 5 and 10, respectively, and new Fig. 4b and 4c in the manuscript report the hiPSC lines that carry *BCOR* mutations.

We have addressed the point regarding the functional consequences of *BCOR* mutations in Figure 5, and in more detail in our response to Point 1.5.

Point 1.5

BCOR is highly expressed by pluripotent stem cell derived neural precursors and linked to neuroepithelial cell tumors and AML. Do the observed BCOR mutations affect neuroectodermal lineage cells or myeloid cell differentiation and function? Can these phenotypes (if present) be rescued by targeted gene repair? Given that BCOR is X-linked, the gender and allele state of the affected lines should also be disclosed in the main text. Given that BCOR function intersects with apoptosis and TP53 pathways it would be interesting to know if episomal reprogramming of PBMCs (which often employs BCL-XL overexpression or TP53 pathway inhibition) results in lower BCOR mutation frequencies compared to Sendai viral reprogramming. The hiPSC derivation, expansion, and subcloning methods should be described in much more detail: differences (e.g., in media composition or addition of small molecule inhibitors) could easily alter the selective pressure landscapes during somatic cell derivation/expansion, reprogramming, and hiPSC expansion.

We thank the reviewer for the insightful question on the impact of *BCOR* mutations on cell differentiation. We have attempted to address this by 1) investigating the relationship between global transcriptional changes of B-hiPSCs and *BCOR* mutations, and 2) performing neuroectoderm differentiation experiments on BCOR mutant and BCOR wildtype hiPSC lines. The results are shown in our new Figure 5 and summarized below:

- As shown in new Figure 5a which is the PCA plot of gene expression of 78 insignia B-hiPSCs, there is segregation of *BCOR* mutants and *BCOR* wildtype samples as well as female and male donors.
- In Figures 5b-d, *BCOR* mutant (MSH40i2) and *BCOR* wildtype (MSH34i2) B-hiPSCs show distinct expression patterns during time-course differentiation at different stages (hiPSC, neural stem cells and neurons).
- Immunofluorescence characterization was performed at the hiPSC, neural stem cells (NSC), and neuron stages, on Day 0, Day 12, and Day 27 respectively (Fig. 5d). This revealed that whilst *BCOR* mutants and wild type show similar pluripotency markers, *BCOR* mutants have impaired differentiation capabilities into NSCs and neurones.

Additionally, we have also added further details in the Methods covering the hiPSC derivation, culture, expansion, and sub-cloning. Table S11 provides the gender and allele state of the affected lines.

We hope that our major revision addresses the Reviewer's concerns. The effects of *BCOR* turns out to be a little more complex but thoroughly interesting. Given our findings and the direction of our work, we have not performed targeted gene repair nor episomal reprogramming. These additional bodies of work could be additionally informative, but which are enormous in scope and unfortunately that we have neither the capabilities of performing them to include in this manuscript, nor do we have the material to do so any longer. However, we hope that the reviewer and editors see the significant value of our manuscript with the additional experimental work that has been performed.

Minor issues:

Point 1.6

That claim that "hiPSCs reprogrammed from erythroblasts show ... *BCOR* mutations in ~57% of lines" appears to correct for the cohort showing 12/21 (57%) but the authors also observed *BCOR* mutations in 2/9 (22%) in the other cohort; i.e., in total, in 14/30 (47%) of such lines.

We thank the Reviewer for highlighting this to us. In this revision, we increased our Insignia B-hiPSC lines from 21 to 78 (the numbers have thus changed). We heed the reviewer's point and have now corrected how this is presented in the abstract and the main text:

"(18% of HipSci and 27% of Insignia, therefore ~25% overall)" – page 15.

Point 1.7

Suppl. Figure 4B, bottom: the legend is shown in reverse order.

Thank you. Corrected.

Point 1.8

The authors should separate the description and analysis of PBMC-erythroblast derived “B-hiPSCs” from that of PBMC-EPC derived ‘B-hiPSCs’: EPCs and erythroblasts are different cell types derived from different lineages and their expansion and reprogramming involves different medias and culture conditions that likely result in significantly different selection pressure landscapes.

Thank you for highlighting this point. Of the blood derived hiPSCs, a small minority are EPC-derived and we have made this clearer now in the text.

Point 1.9

Another note of interest is the frequent occurrence of clonal hematopoiesis, especially in older individuals. One DNMT3A mutation was found in the B-hiPSC, but it was not mentioned whether the nature of the mutation was similar to those found in individuals with clonal hematopoiesis.

Thank you for this suggestion. The mutation is p.G543V and this has indeed been reported in both leukaemias and as a variant seen in CH. We have added this in the main text, acknowledging that the *DNMT3A* mutation may have derived from a CH clone.

Point 1.10

The presence of mutations should be confirmed by PCR/Sanger or ddPCR.

We apologise if we have misunderstood the question here, as we are not clear what the Reviewer is suggesting.

- Whole genome sequencing (WGS) data analysis and Quality Control (QC) have developed to such an extent over the last decade such that it is not customary to perform PCR/Sanger or ddPCR routinely across all passenger variants of WGS data. It would be prohibitively expensive and hugely labour intensive.
- The advantages of doing so are not clear to us either. This is a customary high-depth WGS experiment which has been performed thousands of times in many other studies and confirmation by PCR/Sanger sequencing is not the standard in the field.
 - o All somatic whole cancer genomics projects no longer perform PCR/Sanger or ddPCR sequencing, although this was something we did do when WGS first started (see PMID 22608083 and PMID 22608084 where additional PCRs were carried out for thousands of variants and it became clear that this was an exorbitant exercise and nearly all high-quality variants were true).
 - o However, if in a particular paper, when new techniques that are less reliable are brought into play, for example shallow WGS/sc-seq or new methodologies like in PMID 30840883 that performed unusual experiments where they were looking at unique SNVs in polyclonal peripheral blood/fibroblasts and taking a special reduced (~10%) sequencing strategy, then validation would be reasonably expected there. This is however not the case in our manuscript which utilised standard WGS/WES sequencing, performed at the Wellcome Sanger Institute, with standardised protocols, QC steps, etc.
- However, it may be that the Reviewer does not refer to validating all mutations in this WGS study, but only putatively selected variants. Again, this is not something that is customarily performed because the coding sequences for whole genome (WGS) and whole exome sequencing (WES) tend to be precisely where most variants are true positive findings. Nevertheless, if it helps to reassure the Reviewer, for the B-hiPSCs, we had identified *BCOR* mutations in some parental lines. We then propagated the parental line and then sequenced single-cell derived subclones. The same *BCOR* mutations present in the parental clone were also detected in corresponding subclones (Figure 4c in the manuscript). Each subclone is thus effectively an independent sequencing validation exercise of the parental line. So, there is already an internal validation, and further validation on these variants would not be good use of our limited resources and ultimately would not increase the value of the manuscript.

Reviewer #3:

Remarks to the Author:

Main Findings

The manuscript by Rouhani and colleagues reports substantial mutagenesis in blood and skin derived human induced pluripotent stem cells (hiPSCs) from one individual, especially in hiPSCs derived from skin fibroblasts, which have many ultraviolet-radiation (UV) induced mutations.

The manuscript follows up on this finding with an analysis of whole genome sequence from 324 fibroblast derived hiPSCs from the Human Induced Pluripotent Stem Cell Initiative, "HipSci", and finds UV mutations in 72%, with mutation burdens ranging from 0.25 to 15 single base mutations per megabase.

After adding some additional exome sequence data from HipSci fibroblast derived hiPSCs, the manuscript reports 272 predicted pathogenic mutations in 77 out of 452 hiPSC lines.

The manuscript also analysed mutations in 44 erythroblast-derived hiPSCs (of which 17 were HipSci), which show much less mutagenesis: 0.28 to 1.4 single base mutations per megabase. However, over half of these hiPSCs have mutations in the BCOR (BCL6 corepressor) gene, which the manuscript proposes is due to selection pressure.

The manuscript also notes that the examined hiPSC lines were karyotypically stable, indicating that use of hiPSCs for cell-based therapy would require sequencing in addition to karyotypic screening.

I do not have extensive background in the hiPSC area, but a quick literature search indicates that these sorts of high mutational burdens have not been reported in hiPSCs. In the context of literature on mutagenesis in normal skin fibroblasts, the results regarding the fibroblast derived hiPSCs are not surprising but well worth reporting. Below, I have some further comments on missed opportunities regarding this theme.

Major issues:

Point 3.1

1. It was not possible to fully assess this manuscript given that variant call sets, summary files,

and code (bespoke software) were not available for review (page 26 of the manuscript). These should be provided to allow for a full review.

We have now included the following in the Methods on page 37.

In Data availability:

“The variant call sets and summary files are deposited at Mendeley:

<https://data.mendeley.com/datasets/6rfc2xrnyd/draft?a=7b8e2ce7-c61f-47b9-95f7-049a5c598487>”

In Code availability:

“The code of bespoke software is on github: <https://github.com/dg13/ips-seq>. The code of statistical analysis and figures is on github: https://github.com/Nik-Zainal-Group/hiPSCs_BCOR.git”

Point 3.2

2. Results and interpretation regarding UV mutations in fibroblast derived hiPSCs. There are two substantial publications regarding UV mutagenesis in normal skin; one is reference 41 in the current manuscript and the other is Saini et al., <https://doi.org/10.1371/journal.pgen.1006385>, which should be referenced. One would assume that there is minimal UV exposure during cell culture, and that all UV mutagenesis observed in the hiPSC lines was inherited from the founding fibroblasts. Therefore, the analysis on page 8 showing positive correlation between mutation burdens in founding fibroblasts and derived hiPSC lines seems confirmatory and could be in supplementary information. There is a surprisingly weak statement on page 9: “It is therefore possible that some driver mutations identified in hiPSCs were acquired in vivo and not during cell culture”. The mutational spectra are completely dominated by UV (Figure 2C). Does the manuscript really propose that UV mutations were acquired during cell culture?

Thank you for these points. We have addressed them in turn:

First, we apologise for omission of the reference and have now cited it (ref. 36 in manuscript) as the Reviewer suggested.

Second, the figures showing the positive correlation between mutation burdens in founding fibroblasts and derived hiPSC lines were indeed in Figure 3 and in supplementary information

(Fig. S5). According to the journal format, any supplementary figures should be referred to in the main text, so we used two sentences to describe the analysis in the main manuscript.

Third and in agreement with the Reviewer we do not propose that “the UV mutations were acquired during cell culture” in the present manuscript. Through mutational signature analysis of shared mutations between fibroblasts and derived iPSCs, we provided direct evidence that UV mutations in F-iPSCs were inherited from founding fibroblasts and not from cell culture. However in addition, we also showed that some mutations only seen in F-iPSCs, but not fibroblasts, could be due to the limited sequencing depth: through high coverage WES (271X) of fibroblasts, we discovered more coding mutations that were not seen in normal WES (72X), as shown in Figure S9. Within this context, we wrote the sentence “It is therefore possible that some driver mutations identified in hiPSCs were acquired *in vivo* and not during cell culture”. We therefore apologize that this sentence caused this confusion. We have now rephrased the sentence on Page 11:

“..., it is therefore probable that some mutations identified in hiPSCs but not detected in corresponding fibroblast were still acquired *in vivo* and not during cell culture.”

Lastly, although the vast majority of drivers were acquired *in vivo*, some drivers were indeed acquired *in vitro* (e.g., reprogramming and culture) like *BCOR* mutations. Hence, we suggested reducing culture time to reduce the likelihood of producing additional drivers in hiPSCs.

Point 3.3

The manuscript should computationally attribute (assign) the driver mutations to likely causal mutational processes and report how many are likely due to UV and therefore likely acquired prior to cell culture.

In order to address this point, in this major revision, we have performed three new statistical analyses, including a dN/dS approach across all genes, to detect possible driver mutations (please see Page 1 above). The only gene found to be statistically significant for potential selection is *BCOR*. There are only 11 *BCOR* mutations found in F-hiPSCs (Figure 4b in manuscript) with a spectrum of frameshifting or nonsense mutations (1 C>A, 2 C>T and 8 indels). The vast majority are indels, leaving 3 variants to perform such an analysis. We are hugely underpowered to draw any conclusions.

One could take a maximum likelihood approach to assign substitutions to likely causal mutational processes. However, C>T mutations will almost all be assigned to UV taking such an approach and would simply be because C>T transitions are such a strong phenotype in the signature. Indeed, one C>A mutation in *BCOR* was assigned to the culture-associated signature (Signature 18) and two C>T mutations in *BCOR* were assigned to the UV-associated signature (Signature 7), respectively (Table S5). However, as mentioned previously with such small numbers, it would be a weak inference.

For more details about dN/dS analysis, please see “**i). Statistical work to provide backing for reporting putatively selected variants**” on page 1 of this Response to Reviewers).

Point 3.4

The manuscript misses the opportunity to compare mutation burdens in the founder fibroblast lines and in the derived hiPSCs to those reported in reference 41 and in Saini et al. The manuscript also misses the opportunity to compare the driver mutation profile in fibroblast derived hiPSCs to that reported in reference 41.

Mutation burdens of the HipSci data with the studies mentioned (Ref 41, now Ref 37 in present version of manuscript, and Saini et al) are outlined in Table 2 below.

	HipSci Fibroblasts	Martincorena et al.	Saini et al.
Sample source	288 biopsies of upper arm of 288 donors	234 biopsies of sun-exposed eyelid epidermis from 4 individuals	10 biopsies of left and right lateral forearms and hips from 2 Caucasian male donors
SNV burden	692 – 37,120	2-6 mutations/megabase/cell (6,000-18,000)	581 – 12,743
CC>TT DNV burden	0 – 7,864		-
Indel burden	17 – 641		-

Table 2. Comparisons of mutation burden between our study, reference 37 (41) and Saini et al. Note that Ref 37 estimates are not based on WGS - they are based on targeted sequences.

The position of biopsies and sample size are likely to affect the detected mutation burdens in fibroblasts. Nevertheless, the range of mutation loads are similar across these three studies. We thus revised the following sentence in our manuscript (Page 7):

“These findings are consistent with previous reports of UV-related mutagenesis including similar mutation classes and burdens observed in fibroblasts [refs. Martincorena et al. and Saini et al.] and the more efficient activity of transcription-coupled nucleotide excision repair (TC-NER) particularly in gene-rich early replication timing regions”

Regarding comparing driver mutation profiles in F-hiPSCs in our study to that in reference 41 (now 37) (Martincorena et al.):

- 1) We have now applied the Martincorena approach (dNdScv) on all the coding variants and it transpires that very few of these are likely to be true drivers based on the statistical analyses.
- 2) Reference 37 (41) did not study drivers in F-hiPSCs, but the drivers in normal epidermis (keratinocytes). Drivers found in normal skin are acquired *in vivo*, whilst drivers found in F-hiPSCs include those acquired *in vivo* and *in vitro*, so there are key differences in the two studies.
- 3) Also it is important to note: Reference 37 (41) performed ultra-deep sequencing (500X) of 74 cancer genes in normal skin samples. Hence, genes that were not included in the panel will not be detected, e.g. *BCOR* – so we feel that a direct comparison is not appropriate in this case.

Point 3.5

The Discussion, page 15, states “Whilst 17% of lines have variants in cancer genes that are recurrently mutated in hiPSCs, nearly half of these variants are also present in the starting fibroblast material and represent clonal mutations acquired *in vivo*.”. Does this seem to imply that half were acquired during hiPSC generation and culture as opposed to the UV mutations pre-existing but not being detected? The same issue arises in the sentence on page 10 “in summary, nearly half of cancer-related variants in hiPSC lines that were seen recurrently, had been acquired *in vivo*.”

We thank the reviewer for this point. We did not intend to communicate it in that way, nevertheless, because of the major revision and robust analysis of putative drivers, this has been changed substantially in the manuscript and is now no longer applicable.

Point 3.6

The question of how UV mutations could be acquired during cell culture is not touched on, so these sentences, absent further interpretation, seem fundamentally misleading. In fact, the null hypothesis would be that almost all cancer gene mutations were present in the founding fibroblast population because the overwhelming proportion of mutations were UV mutations (Figure 2C), but these mutations were not detected in the cultured fibroblasts because they were not present in a large-enough proportion of the cells to be detected by sequencing or were present when the hiPSCs were derived but we subsequently lost during culture of the founding fibroblasts.

We do apologise, as we must not have been adequately clear. We did not intend to communicate that the UV mutations were acquired in culture, and at no point do we say this. We agree with Reviewer 3's argument that many mutations were likely to have been in the founding fibroblast population and were not detected because of sequencing depth limitations. We reinforce this on page 10 where we show that deeper sequencing resulted in finding more UV mutations in fibroblasts (Fig. S9).

We have thus adapted the text on page 10:

Second, some F-hiPSC mutations may be present in the fibroblasts but not detected through lack of sequencing depth. Comparing WES with high coverage WES (hcWES) data of originating fibroblasts, we found that an increased sequencing depth **uncovered additional coding mutations that had been** acquired *in vivo*: WES data showed 47% of coding mutations **detected** in hiPSCs were shared with matched fibroblasts. **By contrast, 64% of coding mutations were shared with fibroblasts when using hcWES data in the comparisons** (Fig. S9). The additional 17% of mutations identified only in hcWES exhibited a strong UV substitution signature (Fig. S9) **suggesting that they may have been acquired *in vivo* and have been present within the fibroblast population, but undetected at standard sequence coverage.** Given recent sequencing studies that have demonstrated a high level of cancer-associated mutations in normal cells^{7,8}, it is therefore **probable** that some mutations identified in hiPSCs **but not detected in corresponding fibroblasts were still acquired *in vivo* and not during cell culture.**

On page 19:

"We examined mutations shared between hiPSCs and their matched fibroblasts, representing *in vivo* and/or early *in vitro* mutations, and private mutations that are only present in hiPSCs, ~~purely~~ most (but not all) of which are likely representative ~~representing~~ of mutations acquired *in vitro*."

Point 3.7

3. The BCOR findings are intriguing but there seem to be many loose ends. First, 3 / 17 BCOR mutated hiPSC lines is not very impressive after considering the implicit multiple hypothesis testing across many candidate driver genes. That statistics regarding which driver mutations occur in which lines and which lines are from the same founding population are not clear in the top partial paragraph on page 10 or in the second paragraph on that page. We need a table and more details on how the data were used in the statistical analysis.

We thank the Reviewer for raising the concern of limited B-hiPSC lines in HipSci. Indeed, we also recognized that in HipSci 3 out of 17 B-hiPSCs having BCOR mutations was not very impressive. Therefore, we sought additional B-hiPSC lines from another cohort, the Insignia project. In this major revision, we have increased the number of B-hiPSCs from previously 21 cases to 78. In this exercise, we found $21/78=27\%$ of Insignia B-hiPSC lines have BCOR mutations. The list of mutations in COSMIC cancer genes detected in HipSci and Insignia B-hiPSCs was provided in Tables S7 and S10, respectively.

Furthermore, in this revision, we conducted additional genome-wide selection analysis for F-hiPSCs and B-hiPSCs by calculating dN/dS ratios of mutations for all genes in an agnostic manner, in a restricted manner and also sought out hotspots. This work reveals that only *BCOR* mutations are under significant positive selection in both F-hiPSCs and B-hiPSCs. The detailed dN/dS analysis is described in Methods and the results formed a new subsection “Genome-wide selection analysis of hiPSCs reveals strong selection for BCOR mutations” including a new Figure 4.

We hope that the increased sample size of B-hiPSCs and newly added dN/dS driver analysis assures the Reviewer that our finding of BCOR mutations being positively selected in hiPSCs is both statistically significant and reliable.

Point 3.8

The follow on analysis of the Insignia project data also does not provide enough information. Many of the hiPSC lines were from patients with DNA repair defects, and some may have had high somatic mutation burdens in the founding populations. Of the 9 donors with BCOR mutations, how many were in patients with DNA repair defects and high mutation burdens? I could not locate Tables S7 and S8. Overall, I think the BCOR finding may well be correct, but the supporting data and analysis needs to be provided in detail.

We sincerely apologise if the previous Tables S7 and S8 were not available to the Reviewer. In this revision, we provided a new Fig 4c, Fig 4e and new Table S8 to clearly show the mutation burden and genetic background of all 78 Insignia donors and Table S10 to show all pathogenic mutations including *BCOR* mutations identified in 78 Insignia B-hiPSCs and 141 subclones.

Of note, *BCOR* mutations were found in B-hiPSCs derived from normal donors (control) and patients with DNA repair defects that were not necessarily associated with high levels of mutagenesis in founding populations (new Figure 4c in the manuscript). Hence, there is no correlation between the likelihood of having *BCOR* mutations and the genetic background of the donor. In addition, as shown in Figure 4e in the manuscript, there is no correlation between the likelihood of having *BCOR* mutations and mutation burden.

We hope that Table S8 and S10 are now available to the Reviewer.

Point 3.9

I also think that the discussion on pages 12 and 13 should be clarified. It seems that the key question is whether there is selection for the *BCOR* mutations, for which there seems reasonable evidence. The question of whether there is selection is somewhat independent of whether the mutations were inherited from the founding erythrocyte population or arose during generation and culture of the hiPSCs. It seems that the concept of clonal haematopoiesis here is not that helpful, the real subsidiary question is whether the *BCOR* mutations were present in the founding population of erythroblasts, regardless of whether the proportion of founding cells with the mutation reaches some level to be termed clonal haematopoiesis.

We thank the Reviewer for this important point which we address in the manuscript. Based on the evidence we provide, we believe the *BCOR* variants have arisen post-reprogramming for the following reasons:

- *BCOR* variants were not observed in the founding erythroblasts at standard sequencing depths
- Some *BCOR* variants in B-hiPSCs were present at a lower variant allele fraction (VAF < 0.3), indicating that they may be subclonal.
- Some parental B-hiPSCs did not appear to have a *BCOR* variant, but after cell culture, new *BCOR* variants were identified in daughter subclones and not in the parent, again supporting the mutations to be occurring post reprogramming.

Other comments:

Point 3.10

On page 7, the manuscript draws attention to the interesting finding of high diversity in mutation profile and burden among fibroblast derived hiPSC lines from the same “reprogramming experiment”, including 2 lines from donor HPSI03141-bubh. I assume the authors checked germline variants to ensure there were no sample mix ups and that both derived lines were indeed from the same donor, but this should be stated.

Checking for sample mix-ups, using SNP genotyping, is a standardised part of routine QC at the Sanger Institute and in many other pipelines, and therefore we have confirmed this was checked for all samples in HipSci. We have added a statement to this effect in the methods section (Page 26) and in Fig S14

Point 3.11

One page 8, the manuscript notes “it is essential to compare the F-hiPSC genome to a matched germline sample”. Reading methods, pages 21 through 23, it seems this was not done; the relationship between the methods on pages 21 and 22 to those on page 23 is unclear, and it is not clear which methods were used at which places in the Results section.

Our apologies for causing confusion in that sentence and we thank the Reviewer for this point. In the context of that sentence, our intention was to point out that one should not call mutations in F-hiPSC genomes directly against the founding fibroblasts, as many mutations may be detectable in both the fibroblast and F-hiPSC populations and so will be dismissed by mutation-calling software (Fig. S6). We have now clarified the text to say “preferable” rather than “essential”. Indeed, in our exercise, although matched germline samples are not available for HipSci (but were for Insignia), we still did not use fibroblasts as “normal” in mutation calling, to avoid this problem. Instead, we adopted a bespoke approach for mutation calling to avoid missing mutations of interest that are also present in fibroblasts and verify this approach by identifying mutations which would otherwise have been missed (Fig. S6 - 8). Please see Methods “HipSci HiPSC sequence alignment, QC and variant calling” for more details.

Again, our apology that the previous Method sections were not arranged clearly. Now we have modified the titles, added the details of the differentiation of hiPSCs, RNA sequencing and characterisations and rearranged the subsections in Methods in the following order:

1. Samples

2. HipSci hiPSC line generation and growth
3. HipSci hiPSC line whole exome and genome library preparation and sequencing
4. HipSci hiPSCs sequence alignment, QC, and variant calling
5. Mutation calling of hiPSCs derived from S2, S7 and 10 HipSci lines by using fibroblasts as germline controls
6. Mutational signature analysis
7. Analysis of C>T/CC>>TT transcriptional strand bias in replication timing regions
8. Identification of fibroblast-shared mutations and private mutations in HipSci F-hiPSCs
9. Clonality of samples
10. Variant consequence annotation
11. dNdScv analysis
12. Insignia hiPSC line generation, growth, QC and sequencing
13. Insignia hiPSC variant calling by using blood as germline controls
14. Differentiation of Insignia B-hiPSCs (BCOR-mutant and BCOR-wildtype) and RNA sequencing
15. Processing RNA sequencing data
16. Immunofluorescence staining
17. Other statistical analysis

We hope that the current Methods section is now much clearer and if the Reviewer has any further suggestions/requests, we are happy to improve the Methods accordingly.

Minor comments / typos:

Point 3.12

Page 4 “pre-screened and believed to be” -> “pre-screened and are believed to be”

This has been changed – thank you.

Point 3.13

Page 6, “p value < 0.0001”: what was the comparison and test?

Our apology for not providing the details before. We conducted a correlation test to assess possible linear associations between the mutation burden of substitution signatures and indel types (Fig. S3). We have now added that information in the manuscript (Page 7).

Point 3.14

Page 8, the use of the term “oligoclonal” seems a bit odd in some places, for example, rather than writing “F-hiPSCs derived from the same oligoclonal population will be more similar to each other”, why not write “F-hiPSCs derived from the same subclone will be more similar to each other”? Is there a semantic difference that I am missing?

Thank you for the suggestion which we have adopted and therefore the text has been adapted accordingly.

Point 3.15

Page 9, references 41,42 – there are several other papers reporting “cancer-associated mutations in normal cells”, which probably also should be cited here.

We have now included additional key papers on somatic mutations in normal tissues in Page 11 of our revised manuscript including PMID: 31645730, PMID: 32350471 and PMID: 34594041.

Point 3.16

Figure S7 legend, “and mostly composed” -> “and are mostly composed”

Done. We thank the reviewer for the correction.

Point 3.17

File 60902_0_data_set_662547_qsmx2f.xlsx, tab “germline” does not seem to have a sample id.

We think that the reviewer refers to the “n” input in the “germline” column in current Table S10 (previous Table S8). Table S10 shows all the pathogenic mutations found in cancer genes in Insignia B-hiPSCs and their subclones. To ensure that these mutations were not germline variants, we manually checked the BAM files of all germline samples and recorded the results in the “germline” column. “n” indicates that the mutation was not observed in the matched germline samples. It was not meant to show the germline sample id. We apologise for this confusion and have removed the “germline” column in the current Table S10.

Signed:

Steven G. Rozen

Reviewer #4:

Remarks to the Author:

Rouhani et al. conduct a large-scale investigation of the mutational burden of iPSCs. They observe that skin fibroblast-derived samples retain many UV-related mutations, as was previously described on a smaller number of samples. The authors show that the heterogeneity in the fibroblast donor cells within a single sample may result in different mutational backgrounds for iPSC clones from an individual. They also observe that multiple cancer genes are recurrently mutated, including BCOR. Overall, there is limited novel findings; however, given the large number of iPSC samples with WGS data this study could have the potential to discover novel and interesting findings.

The investigators initially conduct a comparison between the mutational burden and profiles between 12 skin fibroblast-derived and 6 endothelial-derived iPSCs. They find that the fibroblast derived iPSCs have a greater overall mutation largely driven by the presence of a UV damage signature. The fact that fibroblast derived iPSCs carry UV damage mutations has previously been described.

The investigators next investigate more than 452 fibroblast-derived iPSCs confirming that most carry detectable UV damage. They also show that there is no association between the number of mutations and age (which has been shown using fewer iPSCs) or gender of the donor.

Next, the authors show that genomic heterogeneity across iPSCs are largely derived from the clonal heterogeneity of the fibroblast donor cells. It has already been shown that heterogeneity exists across fibroblast cells within the same sample and that reprogramming occurs from a single cell. And hence the concepts underlying the schematic in Figure 3D are not surprising or novel.

Point 4.1

The main advantage of this study is the number of samples (>450, compared to previous studies, which analyzed <20 samples each). The large number of samples is comparable with recent cancer genomics studies and should be exploited to investigate the presence of mutational

hotspots, i.e. genomic locations (or genes) that are more likely to accumulate mutations during reprogramming or that may confer some selective advantage if they harbor mutations in the original fibroblasts. While the observation that many iPSC samples carry mutations in cancer genes is useful from a regenerative medicine standpoint, understanding if certain loci undergo selection may impact how iPSCs and iPSC-derived tissues are used as model systems to study human biology. A genome-wide analysis to identify mutational hotspots could greatly strengthen the novelty of this study, which in general confirms several previous studies that were performed on a smaller scale.

Identification of recurrent BCOR mutations is an interesting observation. It is important that the authors show that the mutational burden in this gene is significantly higher than expected by chance.

Thank you for raising this point which we have now addressed in our significantly revised manuscript. We have added a robust statistical analysis to demonstrate the strength of the effect of recurrent BCOR mutations. Briefly, we conducted a genome-wide selection analysis on 452 Hipsci F-hiPSCs and 78 Insignia B-hiPSCs using dNdScv R package (Martincorena et al., 2017). The results showed that *BCOR* is under significant positive selection with a q-value of 3.64e-08 in F-hiPSCs and 0 in B-hiPSCs. Please see the first section of the response-to-reviewers document for the full details.

Point 4.2

The observation that the oxidative stress mutational signature increases over culture time is an interesting observation. It would be interesting to explore when these mutations arise. Are they present in a minor fraction of the fibroblast population which have a selective advantage during reprogramming, do they occur during reprogramming or during passaging? In the iPSCs are these mutations continually arising or do subclones that carry the mutations have a selective advantage?

Thank you. Indeed, the culture-related signature (caused by an excess of 8-oxo-dG damage) is not specific to iPSCs. It has been demonstrated by multiple groups that this signature is also seen in Embryonic Stem Cells (ESCs)⁹, organoids¹⁰ and in cancer cell lines such as HAP1¹¹. It is thus a universal form of DNA damage that is causing this culture-associated oxidative damage.

Furthermore, we can show directly from our experimental work in this manuscript that the signature arises over progressive passages: When iPSCs from Insignia patients were propagated

over multiple passages and daughter subclones subsequently derived, we found new culture-related mutations in all daughter subclones as well (Figure E).

Figure E. Mutational profiles of *de novo* mutations in (A) four daughter subclones of iPSCs MSH13i1, (B) MSH20i1 from the Insignia project, (C) four subclones from a variety of different gene knockouts from another study¹² and (D) four control subclones from a study on environmental mutagens¹³. Novel mutations arising in daughter subclones from different studies all showed typical signatures of oxidative damage (dominated by C>A mutations). In the present study, we showed that oxidative stress mutational signature increases over culture time, indicating that these mutations were continually arising in all cultured cells.

Point 4.3

The major conceptual issue with this study stems from the fact that the authors identified mutations by comparing iPSCs to their matched donor fibroblasts (using Fisher's exact test). And hence *in vivo* mutations with a high VAF in the fibroblasts will not be identified as mutations in the iPSCs. The authors are therefore restricted to identifying mutations that are subclonal in the fibroblast sample or that occur during reprogramming or subsequent culturing. Additionally, it is impossible to distinguish between mutations that were present in a minor fibroblast population from which the donor cell was derived (and are therefore not observed in the fibroblast WGS data) and those that occurred during reprogramming or shortly thereafter (and hence have a VAF \sim .5 in iPSCs). The authors need to address this point.

We thank the Reviewer for raising these points.

First, we agree with the Reviewer that "in vivo mutations with a high VAF in the fibroblasts will not be identified as mutations in the iPSCs", as these mutations are unlikely to pass the Fisher's exact test. However, no method is perfect. For example, if the HipSci resource sequenced matched blood samples instead of fibroblast, and called mutations in hiPSCs against blood samples, we would indeed remove all germline variation in theory. However, we would not be able to distinguish whether the mutations had been acquired whilst a fibroblast or acquired *in vitro*. In Table 3 below, we listed the pros and cons of three different mutation calling approaches.

Mutation calling approach for hiPSCs	Pros	Cons
Blood as "normal"	Remove all germline mutations.	Cannot distinguish the source of mutations (i.e. acquired in fibroblast or in culture)
Fibroblast as "normal"	Removes germline mutations	Computationally removes a large number of somatic mutations acquired in vivo even though the mutations are still present in the cell line
Our bespoke approach	Removes germline mutations	May miss a few somatic mutations that are present in fibroblast at high VAF (although this may be helpful to

		remove in vivo acquired driver events)
--	--	---

Table 3. Comparison of different mutation calling approaches.

Ideally, we could do 1) whole genome sequencing of all 288 blood samples, 288 fibroblast, 452 hiPSCs, 2) calling mutations in fibroblasts and hiPSCs separately against matched blood samples and 3) comparing mutations found in fibroblasts and corresponding hiPSCs to identify the sources of mutations. We understand that the Reviewer certainly was not asking us to sequence blood samples but checking if we were aware of the shortcomings of our mutation calling approach. We can ensure the Reviewer that we are fully aware of them, but we don't think these shortcomings will change the findings of this paper for the following two reasons:

1. The mutation burden found of F-hiPSCs in this study is consistent with recent studies [D'Antonio, M. et al, 2018; McCarthy, D.J. et al., 2020], indicating that we did not lose a significant number of somatic mutations.
2. When we searched for driver mutations, we first used high sensitivity call set, e.g. without Fisher's exact test, to make sure no potential somatic driver mutations was missed. Second, if a driver mutation was seen in a fibroblast, we would check whether it had a dbSNP ID. For all *BCOR* mutations, we don't see them in fibroblasts (Figure 4b in manuscript) and confirmed that they had no dbSNP IDs. Hence importantly, we did not miss any potential driver mutations.

Second, we also agree with the Reviewer that "it is impossible to distinguish between mutations that were present in a minor fibroblast population from which the donor cell was derived (but not observed in the fibroblast WGS data i.e. missed at the current sequencing depth) and those that occurred during reprogramming or shortly thereafter (and hence have a VAF \sim .5 in iPSCs)". This is the reason that it is possible to observe UV signatures amongst mutations observed only in iPSCs (and that were not detected in the matched fibroblasts) as shown in Fig. 6a-c. Indeed, through comparing mutations obtained from WES and high-coverage WES (hcWES), we verified this by identifying more mutations in hcWES that were not seen in WES (Fig. S9). In the manuscript (Page 10) we have written:

"Comparing WES with high coverage WES (hcWES) data of originating fibroblasts, we found that an increased sequencing depth uncovered additional coding mutations that had been acquired *in vivo*: WES data showed 47% of coding mutations detected in hiPSCs were shared with matched fibroblasts. By contrast, 64% of coding mutations were shared with fibroblasts when using hcWES data in the comparisons (Fig. S9). The additional 17% of mutations identified only in hcWES exhibited a strong UV substitution signature (Fig. S9) suggesting they had been acquired *in vivo*

and have been present within the fibroblast population, but undetected at standard sequence coverage.”

Third, in the present study, we do not aim to precisely classify each mutation as *in vivo* or *in vitro* mutations, but to use reasonable criteria to qualitatively separate mutations according to their possible causal sources and understand how different mutational processes shape the mutational landscape in hiPSCs.

In summary, we feel that we have shown and acknowledged that mutations not seen in fibroblasts could be still present in a minor fibroblast population and we are cautious of it when we interpret our results.

Minor comments

Point 4.4

While the authors acknowledge previous work addressing somatic mutations in iPSCs (Introduction: “systematic large-scale, whole genome assessments of mutagenesis at single-nucleotide resolution have been limited”), they do not discuss the findings of these studies, but focus on recent studies on embryonic stem cells.

We have modified the text (cognizant of not extending the length of the paper too much) in the manuscript on page 3. Our introduction to somatic mutations in human pluripotent stem cells is skewed towards hESCs because to date there has been no comprehensive study on large numbers of hiPSCs (e.g. current studies report on 18 or 22 hiPSC lines only). This further emphasises the importance of our manuscript and the context in which it fits in with previous work published.

Point 4.5

While this study and others show that there is no correlation between mutation burden of F-hiPSC and donor age or gender, as a previous study (ref. 24) is in contrast with this statement it would be good to discuss the discrepancy.

PMID 27941802 reported a relationship with age when they examined 16 iPSC lines derived from blood, not skin. Furthermore, these had been whole exome sequenced, not genome sequenced. The very small number of lines together with the fact that they focused on 1% of a genome meant

that the study was rather under-powered. Additionally, the total numbers of mutations are very low in an exome of blood-derived iPSCs and we note that the absolute numbers were rarely referred to. In fact, in that paper, the counts were less than 30 per line, more than half had less than 20 mutations and some were in single digits. Interestingly, the relationship with age was not consistent and in very elderly donors over 90, the authors reported fewer mutations which they argued was due to a “contracted progenitor pool”.

When we reported that we did not see a relationship with age, that was in skin-derived lines not blood-derived lines and that is simply because of the extent of UV damage in the skin-derived lines, which is in the thousands or tens of thousands. The degree of UV damage to skin is related to the amount of sun exposure and not necessarily age, which has been noted previously⁶.

We had added the following to the manuscript in page 7 to clarify this point:

“Of note, similar to findings of UV damage in skin⁴², there was no correlation between mutation burden of F-hiPSC and donor age or gender (Fig. 2g and 2h).”

Point 4.6

Figure 3B shows the “number of mutations in fibroblasts”, but it is unclear from the Methods how this number was calculated.

Our apology for not explaining it clearly previously. The variants in each fibroblast were first called using BCFtools/mpileup and BCFtools/call version 1.4.25. Then all the germline variants that were above 0.1% minor allele frequency in 1000 Genomes phase 3 or ExAC 0.3.1 were excluded. The number of remaining variants is the number of mutations in fibroblasts. We have provided all the mutation calls used in our analysis on Mendeley Data website. We have also included the following in our updated Methods (Page 37):

Code availability

The code of bespoke software is on github: <https://github.com/dg13/ips-seq>. The code of statistical analysis and figures is on github: https://github.com/Nik-Zainal-Group/hiPSCs_BCOR.git. The variant call sets and summary files are deposited at Mendeley: <https://data.mendeley.com/datasets/6rfc2xrnyd/draft?a=7b8e2ce7-c61f-47b9-95f7-049a5c598487>

Point 4.7

In the abstract “In contrast, B-hiPSCs reprogrammed from erythroblasts show lower levels of genome-wide mutations” is not supported by any statistical analysis. The primary analysis is on page 5 – starting on line 4.

The sentence in the abstract previously referred to the comparison of mutation burdens of a cohort of B-hiPSCs reprogrammed from erythroblasts (range 0.28-1.4 per Mb) with that of a large cohort of F-hiPSCs reprogrammed from fibroblasts (range 0.25-15 per Mb). We provided a range of mutations per megabase and did not perform any statistical testing since these are sourced from different donors.

By contrast, there is a different analysis on page 5 where we did compare the mutation burden of blood-derived-hiPSCs and F-hiPSCs obtained from the same donor (S2), showing that there is a greater number of mutations (~4.4 increase) in F-hiPSCs, as compared to B-hiPSCs (Fig. 1b). This is a fair comparison, but was not the context of the sentence in the abstract.

Regardless, the abstract has been changed substantially and the context of that sentence has been made clearer.

References

- 1 Martincorena, I. *et al.* Universal Patterns of Selection in Cancer and Somatic Tissues. *Cell* **171**, 1029-1041 e1021, doi:10.1016/j.cell.2017.09.042 (2017).
- 2 Wang, Z. *et al.* A Non-canonical BCOR-PRC1.1 Complex Represses Differentiation Programs in Human ESCs. *Cell Stem Cell* **22**, 235-251 e239, doi:10.1016/j.stem.2017.12.002 (2018).
- 3 Degasperi, A. *et al.* A practical framework and online tool for mutational signature analyses show inter-tissue variation and driver dependencies. *Nat Cancer* **1**, 249-263, doi:10.1038/s43018-020-0027-5 (2020).
- 4 Brash, D. E. *et al.* A role for sunlight in skin cancer: UV-induced p53 mutations in squamous cell carcinoma. *Proc Natl Acad Sci U S A* **88**, 10124-10128, doi:10.1073/pnas.88.22.10124 (1991).
- 5 Fousteri, M. & Mullenders, L. H. Transcription-coupled nucleotide excision repair in mammalian cells: molecular mechanisms and biological effects. *Cell Res* **18**, 73-84, doi:10.1038/cr.2008.6 (2008).
- 6 Thieden, E., Philipsen, P. A., Heydenreich, J. & Wulf, H. C. UV radiation exposure related to age, sex, occupation, and sun behavior based on time-stamped personal dosimeter readings. *Arch Dermatol* **140**, 197-203, doi:10.1001/archderm.140.2.197 (2004).
- 7 Martincorena, I. *et al.* Tumor evolution. High burden and pervasive positive selection of somatic mutations in normal human skin. *Science* **348**, 880-886, doi:10.1126/science.aaa6806 (2015).
- 8 Martincorena, I. *et al.* Somatic mutant clones colonize the human esophagus with age. *Science* **362**, 911-917, doi:10.1126/science.aau3879 (2018).
- 9 Thompson, O. *et al.* Low rates of mutation in clinical grade human pluripotent stem cells under different culture conditions. *Nat Commun* **11**, 1528, doi:10.1038/s41467-020-15271-3 (2020).
- 10 Kuijk, E. *et al.* The mutational impact of culturing human pluripotent and adult stem cells. *Nat Commun* **11**, 2493, doi:10.1038/s41467-020-16323-4 (2020).

- 11 Zou, X. *et al.* Validating the concept of mutational signatures with isogenic cell models. *Nat Commun* **9**, 1744, doi:10.1038/s41467-018-04052-8 (2018).
- 12 Zou, X. *et al.* A systematic CRISPR screen defines mutational mechanisms underpinning signatures caused by replication errors and endogenous DNA damage. *Nat Cancer* **2**, 643-657, doi:10.1038/s43018-021-00200-0 (2021).
- 13 Kucab, J. E. *et al.* A Compendium of Mutational Signatures of Environmental Agents. *Cell* **177**, 821-836 e816, doi:10.1016/j.cell.2019.03.001 (2019).

Decision Letter, first revision:

2nd Feb 2022

Dear Dr Nik-Zainal,

First of all, please accept my apologies for the delay in returning this decision to you.

Your Article, "Substantial somatic genomic variation and selection for BCOR mutations in human induced pluripotent stem cells" has now been seen by 4 referees. As in the previous round, please note that Reviewers #1 and #2 reviewed the paper together and have uploaded the same report. You will see from their comments below that while they find your work of interest, some important points are raised. We are interested in the possibility of publishing your study in Nature Genetics, but would like to consider your response to these concerns in the form of a revised manuscript before we make a final decision on publication.

Thank you for your email of 30 January, which I discussed with my colleagues in the context of this decision. We agree that providing robust evidence for the role of BCOR mutations in the observed phenotype is importance, given the novelty of the finding. We agree that adding data from additional mutant versus wild-type cell lines, as you suggest, would be valuable (providing that that they are independent biological and not technical replicates). We believe that the CRISPR experiments suggested by Reviewers #1 and #2 would add further useful orthogonal evidence, but the absence of these data will not preclude our interest in the paper. Regarding the other concerns, please address them, particularly those raised by Reviewer #3 pertaining to data availability. Your comments regarding the use of dNdScv to identify signals of selection are well taken, and we think that you can address this textually in your point-by-point letter. At this stage, we would likely return the manuscript to Reviewers #1 and #2 but depending on your response, we might have to go back to others. Rest assured that we'll only do this if absolutely necessary.

We therefore invite you to revise your manuscript taking into account all reviewer and editor comments. Please highlight all changes in the manuscript text file. At this stage we will need you to upload a copy of the manuscript in MS Word .docx or similar editable format.

We are committed to providing a fair and constructive peer-review process. Do not hesitate to contact us if there are specific requests from the reviewers that you believe are technically impossible

or unlikely to yield a meaningful outcome.

*2) If you have not done so already please begin to revise your manuscript so that it conforms to our Article format instructions, available

[here](http://www.nature.com/ng/authors/article_types/index.html).

*3) Include a revised version of any required Reporting Summary:

[REDACTED]

We hope to receive your revised manuscript within four to eight weeks. If you cannot send it within this time, please let us know.

Sincerely,

Safia Danovi

Editor
Nature Genetics

Reviewers' Comments:

Reviewer #1 (reviewed with Reviewer #2):

Remarks to the Author:

The authors have earnestly attempted to respond to the several issues raised during the prior round of reviews, and have improved the manuscript. The authors rebut the prior concern raised that their work replicates and extends numerous prior reports of iPSC heterogeneity by pointing out fairly minor differences that support the novelty of the current study. While much of the current study adds incrementally to an already extensive literature, the observation of the role of BCOR is potentially more important and publication-worthy, subject to the limitation of the interpretation of significance for iPSC differentiation noted below.

Major point:

The authors demonstrate marked differences in the differentiation of BCOR mutant vs non-mutant lines in neural differentiation, with BCOR mutant status correlating with inefficient differentiation as reflected in lower PAX6 marker expression and subsequently reduced numbers of TUBB3+ neurons. The authors are to be commended for conducting a comparison of BCOR mutant and WT hiPSCs that suggests functional significance of the BCOR mutations. However, the authors only compared one BCOR-mutant (albeit in 3 replicates) to one BCOR-WT hiPSC line. Any two hPSC lines are likely to differ somewhat in their IVD performance, especially if they came from different donors (as is the case here – e.g, <https://www.ncbi.nlm.nih.gov/pubmed/22802639>, <https://pubmed.ncbi.nlm.nih.gov/18278034/>). Consequently, several journals have adopted a policy that requires these types of comparisons to include ≥ 2 hiPSC lines from ≥ 2 donors per group (i.e., a minimum of 8 independent lines), especially if the lines being compared are genetically diverse. The request to meet this standard seems appropriate (given that the outcome could affect one of the key claims) as well as feasible (given that many BCOR-mutant and WT B-hiPSC lines are available and that the authors already conducted six such experiments [3 replicate experiments for each of the two lines]). While adding additional lines is one means of strengthening the conclusion, the better experiment is CRISPR gene correction in iPSCs, which is widely practiced. The data would be considerably more compelling if the differences were reverted by CRISPR-correction of the BCOR mutation, thereby confirming that the BCOR mutation itself and not some other of the myriad differences between the small number of clones compared truly account for the phenotype.

Minor points:

The authors expanded the number of Insignia B-hiPSCs from 21 to 78. The original cohort contained 12 instances of BCOR mutant lines (12/21=57%), while the additional 57 Insignia B-hiPSC lines only included 9 BCOR-mutant lines (15.7%). This difference is quite substantial and statistically significant ($p < 0.0001$; Barnard test). The independent HipSci cohort also included a much lower frequency of BCOR mutant lines (3/17=17.6%). Are there any systemic differences in donor populations or experimental procedures between the high-frequency cohort and the lower-frequency cohorts that could explain these discordant frequencies?

The authors use Fisher's exact test. This test should practically never be used (see <https://pubmed.ncbi.nlm.nih.gov/19170020/>); the calculations should be repeated with a more

appropriate test (I am not a statistician, but Barnard or Boschloo might be better options).

The erythroblast expansion medium details are still missing ('cultured in media favouring expansion into erythroblasts for 9 days' is too vague)

The hiPSC culture medium and passaging method details are still missing ('onto ... MEF-CF1 feeder plates and cultured in iPS cell medium with daily medium change until ready to passage.' Is too vague)

Reviewer #2 (reviewed with Reviewer #1):

Remarks to the Author:

The authors have earnestly attempted to respond to the several issues raised during the prior round of reviews, and have improved the manuscript. The authors rebut the prior concern raised that their work replicates and extends numerous prior reports of iPSC heterogeneity by pointing out fairly minor differences that support the novelty of the current study. While much of the current study adds incrementally to an already extensive literature, the observation of the role of BCOR is potentially more important and publication-worthy, subject to the limitation of the interpretation of significance for iPSC differentiation noted below.

Major point:

The authors demonstrate marked differences in the differentiation of BCOR mutant vs non-mutant lines in neural differentiation, with BCOR mutant status correlating with inefficient differentiation as reflected in lower PAX6 marker expression and subsequently reduced numbers of TUBB3+ neurons. The authors are to be commended for conducting a comparison of BCOR mutant and WT hiPSCs that suggests functional significance of the BCOR mutations. However, the authors only compared one BCOR-mutant (albeit in 3 replicates) to one BCOR-WT hiPSC line. Any two hPSC lines are likely to differ somewhat in their IVD performance, especially if they came from different donors (as is the case here – e.g, <https://www.ncbi.nlm.nih.gov/pubmed/22802639>, <https://pubmed.ncbi.nlm.nih.gov/18278034/>). Consequently, several journals have adopted a policy that requires these types of comparisons to include ≥ 2 hiPSC lines from ≥ 2 donors per group (i.e., a minimum of 8 independent lines), especially if the lines being compared are genetically diverse. The request to meet this standard seems appropriate (given that the outcome could affect one of the key claims) as well as feasible (given that many BCOR-mutant and WT B-hiPSC lines are available and that the authors already conducted six such experiments [3 replicate experiments for each of the two lines]). While adding additional lines is one means of strengthening the conclusion, the better experiment is CRISPR gene correction in iPSCs, which is widely practiced. The data would be considerably more compelling if the differences were reverted by CRISPR-correction of the BCOR mutation, thereby confirming that the BCOR mutation itself and not some other of the myriad differences between the small number of clones compared truly account for the phenotype.

Minor points:

The authors expanded the number of Insignia B-hiPSCs from 21 to 78. The original cohort contained 12 instances of BCOR mutant lines (12/21=57%), while the additional 57 Insignia B-hiPSC lines only included 9 BCOR-mutant lines (15.7%). This difference is quite substantial and statistically significant ($p < 0.0001$; Barnard test). The independent HipSci cohort also included a much lower frequency of BCOR mutant lines (3/17=17.6%). Are there any systemic differences in donor populations or

experimental procedures between the high-frequency cohort and the lower-frequency cohorts that could explain these discordant frequencies?

The authors use Fisher's exact test. This test should practically never be used (see <https://pubmed.ncbi.nlm.nih.gov/19170020/>); the calculations should be repeated with a more appropriate test (I am not a statistician, but Barnard or Boschloo might be better options).

The erythroblast expansion medium details are still missing ('cultured in media favouring expansion into erythroblasts for 9 days' is too vague)

The hiPSC culture medium and passaging method details are still missing ('onto ... MEF-CF1 feeder plates and cultured in iPS cell medium with daily medium change until ready to passage.' Is too vague)

Reviewer #3:

Remarks to the Author:

The paper is substantially improved, and my comments from the previous review have been addressed. In particular, selection for the BCOR mutations seems solid, which is a quite interesting result.

Minor comments:

The description of dN/dS analysis is clearer in the rebuttal than in the manuscript. In addition, the rebuttal and the manuscript's Methods section refers to the hotspot analyses, but I did not see these results presented in the paper.

On Mendeley, the folder with the Insignia data seems empty. Also under HipSci, the Drivers folder seems empty. Please see the uploaded word document for screenshots. I will leave it to the editorial staff to follow up.

Signed, Steven G. Rozen

Reviewer #4:

Remarks to the Author:

1. As discussed in the first review much the findings described in the beginning Results sections are not novel. Rather the findings confirm several previous studies that were performed on a smaller scale.

- High prevalence of UV-associated DNA damage in F-hiPSCs.
- Substantial genomic heterogeneity between F-hiPSCs is due to clonal populations present in the starting material

2. Identification of BCOR mutations higher than expected by chance in hiPSCs would be of interest. The authors attempted to show that the mutational burden in this gene is significantly higher than expected by chance by analyzing all genes in genome using dNdScv and found only BCOR as being under significant positive selection ($qval=3.64e-08$). I am not sure the method they used is the most appropriate. There are standard methods in the cancer field for conducting this type of analysis (such as MuTect). Why did the authors choose to identify genes that have a dn/ds that deviates from the

expected (which is a method used in population genetics to look for selection) rather than employ a standard method from the cancer field that for identifying genes with mutation rates higher than expected by chance?

3. The experimental work and functional analyses to understand the impact of BCOR in hiPSCs significantly adds to the paper.

Author Rebuttal, first revision:

Substantial somatic genomic variation and selection for *BCOR* mutations in human induced pluripotent stem cells

Foad J Rouhani^{1,2,†}, Xueqing Zou^{3,4,†}, Petr Danecek^{1,†}, Cherif Badja^{3,4}, Tauanne Dias Amarante³, Gene Koh³, Qianxin Wu¹, Yasin Memari³, Richard Durbin¹, Inigo Martincorena¹, Andrew R Bassett¹, Daniel Gaffney^{1*}, Serena Nik-Zainal^{3*}

Supplementary Table

	Donors	Parental lines	Subclone lines	Sequencing method	“normal” used to remove germline mutations
F-hiPSCs	S2	2	0	WGS	Fibroblast
	9 from HipSci	10	0	WGS	Fibroblast
	HipSci 288 healthy donors	452	0	324 WGS 381 WES 106hcWES	Bespoke approach
B-hiPSCs	S2	2	0	WGS	Fibroblast
	S7	4	0	WGS	Fibroblast
	HipSci 9 donors (2	17	0	WES	-

	normal + 7 patients)				
	Insignia 78 patients	78	141	WGS RNA-seq for 78 parental lines	Germline control

Table S1. Summary of hiPSC samples.

Supplementary Figures

Figure S1. Mutational profiles of 18 blood-derived iPSCs and skin-derived iPSCs featured in Figure 1.

Figure S2. Distribution of passage number of HipSci F-hiPSCs. The median is 18. Passage numbers of eight non-HipSci hiPSCs are: P2 for: S7_RE11, S7_RE14, S7_RE2, S7_RE17, S2_RE19, S2_SF3 and S2_SF2; P1 for S2_RE5.

Figure S3. (A) Distribution of mutational signatures in 324 fibroblast-derived iPSC lines. The inset figure shows the relative exposures of mutational signatures/indel types. (B) Correlation between substitution signatures and indel types.

Figure S4. Mutation burden and mutational signatures in fibroblasts. (A) Mutation burden of substitutions, CC>TT double substitutions and indels in fibroblasts. (B) The amount of each mutational signature and indel type (exposure) in fibroblasts.

Figure S5. Analysis of variant allele frequency in fibroblasts and iPSCs. Distribution of variant allele frequency distribution of five fibroblasts and five hiPSCs are shown in (A) and (B), respectively. Kernel density estimation was used to smooth the distribution. Local maximums and minimums were calculated to identify subclonal clusters. (C) Summary of subclonal clusters in fibroblasts and hiPSCs. Each dot represents a cluster which has at least 10% of total mutations in the sample. Most of the fibroblasts are polyclonal with VAF of a cluster of nearly 0.25, whilst hiPSCs are mostly clonal with VAF of nearly 0.5.

Figure S6. Number of substitutions that were removed from iPSCs by using fibroblast as “normal” for ten HipSci samples from Figure 1.

Figure S7. Shared mutations between hiPSCs and the matched fibroblasts.

Figure S8. Mutation profile of substitutions that were removed from iPSCs by using fibroblast as “normal” for ten HipSci samples from Figure 1. These removed mutations are not germline SNPs, and are mostly composed of mutations that are typical of UV exposure.

Figure S9. Comparison of WES (72X) and high coverage WES (hcWES, 271X) of fibroblasts. More mutations in hiPSCs were discovered in fibroblasts (shared mutations) through hcWES than through WES, resulting in the percentage of shared-mutations increased in hcWES data. Interestingly, the mutational profile of these increased shared-mutations is very similar to the UV signature, indicating that increasing sequencing depth enables more UV-caused somatic mutations that were found in hiPSCs to also be detected in the corresponding fibroblasts.

Figure S10. Brightfield images showing cell morphology changes during neural differentiation stages. BCOR-mut samples have normal hiPSC colony morphology (Day 0). The differentiation of cells into neurons (Day 27) showed fewer differentiated cells in BCOR-mut compared to BCOR-wt.

Figure S11. Immunofluorescence characterization of BCOR-mut and BCOR-wt at different neural differentiation stages. (a) At the hiPSC stage, BCOR-mut colonies are indistinguishable from BCOR-wt, both expressing SSEA4 (green) and OCT4/POU5F1 (red). (b) and (c) BCOR-mut cells have impaired neural differentiation resulting in fewer Pax 6 positive cells at Day 12 (b) and TUBB3 positive cells at Day 27 (c).

Figure S12. Principal Component (PC) Analysis of RNA sequencing data. PC analysis of RNA-seq data shows transcriptomic differences in both BCOR-mut lines compared to both BCOR-wt samples, across the neural differentiation stages.

Figure S13. Histogram of shared and private (de novo) mutations for signature 7 (UV), signature 18 (oxidative damage), [-]Mh and [+]T.

Figure S14. There is no evidence of contamination except for one cell line and there is no correlation between the number of mutations and the FREEMIX score ($R^2=0.1$). The dashed line at 0.03 is the threshold suggested by VerifyBamID to accept or potentially flag the sample as contaminated. The outlier cell line (HPSI0913pf-coyi) was removed from analysis.

Figure S15. *De novo* extraction on 324 skin-derived WGS hiPSCs from the HipSci project. (A) Metrics for selecting the optimal number of signatures. (B) Four mutational signatures extracted from this data set. Profiles of similar skin cancer derived signatures are shown. (C) Cosine similarities between F-iPSCs signatures and skin cancer derived signatures. S2 and S4 are most similar (cossim: 0.94-0.98) to UV-associated mutational signatures, Skin_J and Skin_D (signature 7), respectively. S1 is most similar to Skin_A (signature 18), the culture signature (cossim: 0.94). S3 does not show high similarity to any skin-specific signatures (cossim <0.8), but also has very low probabilities for all 96 channels (note y-axis values are very small), and the relatively featureless profile would suggest that it is likely to be “noise”. This is not uncommon in signature extractions.

Decision Letter, second revision:

Our ref: NG-A57534R1

23rd Feb 2022

Dear Dr. Nik-Zainal,

Thank you for submitting your revised manuscript "Substantial somatic genomic variation and selection for BCOR mutations in human induced pluripotent stem cells" (NG-A57534R1). It has now been seen by Reviewer #1 and their comments are below. The reviewers find that the paper has improved in revision, and therefore we'll be happy in principle to publish it in Nature Genetics, pending minor revisions to comply with our editorial and formatting guidelines.

Sincerely,

Safia Danovi
Editor
Nature Genetics

Reviewer #1 (Remarks to the Author):

To respond to the request for additional corroboration of the effect of BCOR mutation on the differentiation of mutant iPSCs, the authors have added analysis of an additional wild type and an additional BCOR mutant line, with comparable results. The authors assert that CRISPR repair is beyond the scope of the publication. The added data supports the conclusions that BCOR mutation alters the transcriptome and the differentiation along the neural lineage. The report of BCOR mutation during iPSC derivation and culture is potentially highly important and worthy to be reported to the community. Understanding why truncating BCOR mutations are selected for during iPSC culture in future studies will be critical to developing reprogramming and iPSC expansion strategies that minimize this risk.

Final Decision Letter:

In reply please quote: NG-A57534R2 Nik-Zainal

24th Jun 2022

Dear Dr. Nik-Zainal,

I am delighted to say that your manuscript "Substantial somatic genomic variation and selection for BCOR mutations in human induced pluripotent stem cells" has been accepted for publication in an upcoming issue of Nature Genetics.

Your paper will be published online after we receive your corrections and will appear in print in the next available issue. You can find out your date of online publication by contacting the Nature Press Office (press@nature.com) after sending your e-proof corrections. Now is the time to inform your Public Relations or Press Office about your paper, as they might be interested in promoting its publication. This will allow them time to prepare an accurate and satisfactory press release. Include your manuscript tracking number (NG-A57534R2) and the name of the journal, which they will need when they contact our Press Office.

Please note that *Nature Genetics* is a Transformative Journal (TJ). Authors may publish their research with us through the traditional subscription access route or make their paper immediately open access through payment of an article-processing charge (APC). Authors will not be required to make a final decision about access to their article until it has been accepted. [Find out more about Transformative Journals](https://www.springernature.com/gp/open-research/transformative-journals)

Authors may need to take specific actions to achieve [compliance with funder and institutional open access mandates](https://www.springernature.com/gp/open-research/funding/policy-compliance-faqs). If your research is supported by a funder that requires immediate open access (e.g. according to [Plan S principles](https://www.springernature.com/gp/open-research/plan-s-compliance)) then you should select the gold OA route, and we will direct you to the compliant route where possible. For authors selecting the subscription publication route, the journal's standard licensing terms will need to be accepted, including <https://www.nature.com/nature-portfolio/editorial-policies/self-archiving-and-license-to-publish>. Those licensing terms will supersede any other terms that the author or any third party may assert apply to any version of the manuscript.

Please note that Nature Portfolio offers an immediate open access option only for papers that were first submitted after 1 January, 2021.

An online order form for reprints of your paper is available at https://www.springernature.com/gp/open-research/reprints

href="https://www.nature.com/reprints/author-reprints.html">https://www.nature.com/reprints/author-reprints.html. Please let your coauthors and your institutions' public affairs office know that they are also welcome to order reprints by this method.

If you have not already done so, we invite you to upload the step-by-step protocols used in this manuscript to the Protocols Exchange, part of our on-line web resource, natureprotocols.com. If you complete the upload by the time you receive your manuscript proofs, we can insert links in your article that lead directly to the protocol details. Your protocol will be made freely available upon publication of your paper. By participating in natureprotocols.com, you are enabling researchers to more readily reproduce or adapt the methodology you use. Natureprotocols.com is fully searchable, providing your protocols and paper with increased utility and visibility. Please submit your protocol to <https://protocolexchange.researchsquare.com/>. After entering your nature.com username and password you will need to enter your manuscript number (NG-A57534R2). Further information can be found at <https://www.nature.com/nature-portfolio/editorial-policies/reporting-standards#protocols>

Sincerely,

Safia Danovi
Editor
Nature Genetics